# GLP-1R–GIPR–PPARα/γ/δ quintuple agonism corrects obesity and diabetes in mice

Daniela Liskiewicz[1,2,3,34], Aaron Novikoff[1,2,34], Ahmed Khalil[1,2], Seun Akindehin[1,2], Jonathan E. Campbell[4,5], Pietra Candela[6], Russell L. Castelino[1,2], Callum Coupland[1,2], Maxime Culot[6], W. Scott Dodson[7], Jonathan D. Douros[7,8], Hannes Embring[9], Annette Feuchtinger[10], Brian Finan[8], Cristina Garcia-Caceres[1,2,11], Xiao-Bing Gao[12], Fabien Gosselet[6], Gerald Grandl[1,2], Robert M. Gutgesell[1,2,13], Daniel T. Haas[1,2], Martin Jastroch[14], Ezgi Karaoglu[1,2,15], Pamela Kakimoto[1,2], Anna Cristina Kaltenbach[16], Michaela Keuper[14], Christine M. Kusminski[17], Danielle C. Leander[5], Arkadiusz Liskiewicz[1,2,18], Xue Liu[1,2], Gandhari Maity-Kumar[1,2], Sara Martinez Martinez[19], Stephanie A. Mowery[7,8], Ruben Nogueiras[19], Marshall Paisley[5], Diego Perez-Tilve[20], Patricia S. S. Petersen[9], Paul T. Pfluger[1,2,21,22], Sneha Prakash[1,2], Sabine Steffens[16,23,24], Alberto Cebrian-Serrano[1,2], Monica Tost[10], Jordan Wean[25], Christian Weber[16,24,26,27], Junichi Yoshida[12], Zachary Gerhart-Hines[9], Tamas L. Horvath[12,28], Philipp E. Scherer[17], Randy J. Seeley[25], Richard D. DiMarchi[29], Matthias H. Tschöp[30,31], Natalie Krahmer[1,2,32], Patrick J. Knerr[7,8] & Timo D. Müller[1,2,33 ✉]

There are increasing numbers of effective drugs to improve obesity-linked metabolic dysfunction; GLP-1R–GIPR co-agonism is effective in the management of obesity and type 2 diabetes[1,2], and lanifibranor—a nuclear-acting small-molecule triple agonist of PPARα, PPARγ and PPARδ—is in clinical phase 3 trials for the treatment of metabolic dysfunction-associated steatohepatitis[3]. Here, seeking to further improve the metabolic efficacy of GLP-1R–GIPR co-agonism, we report the development of a unimolecular quintuple agonist that combines the body weight-reducing and blood glucose-lowering effects of GLP-1R–GIPR co-agonism with the insulin-sensitizing and anti-inflammatory effects of lanifibranor via its targeted delivery into GLP-1R- and GIPR-expressing cells. In vitro, GLP-1–GIP–lanifibranor is indistinguishable from GLP-1–GIP in relation to incretin receptor signalling and shows equal stimulation of insulin secretion in isolated mouse islets. In vivo, however, GLP-1–GIP–lanifibranor outperforms GLP-1R–GIPR co-agonism and semaglutide, further decreasing body weight, food intake and hyperglycaemia in obese and insulin-resistant mice through synergistic incretin and PPAR action. The metabolic action of GLP-1–GIP–lanifibranor is blunted in mice with genetic or pharmacological inhibition of GLP-1R, GIPR or PPARδ and is absent in DIO double incretin receptor-knockout mice, collectively suggesting that GLP-1–GIP–lanifibranor has substantial therapeutic value in the treatment of obesity and diabetes.

In recent years, there has been a remarkable resurgence in drugs to treat obesity and its related co-morbidities. A standout example is tirzepatide, a co-agonist at the receptors for GLP-1 and GIP. In phase 3 clinical trials, tirzepatide improved liver fibrosis in individuals with metabolic dysfunction-associated steatohepatitis (MASH)[4], and outperformed semaglutide in the management of obesity[1] and type 2 diabetes[2]. Along with preclinical data indicating that long-acting GIPR agonists act in the brain to decrease body weight and food intake via GIPR signalling in inhibitory γ-aminobutyric acid-producing (GABAergic) neurons[5–7], and by mitigating the emetic effects of GLP-1R agonism[8,9], GIPR–GLP-1R co-agonism has been established as a highly effective strategy for the management of obesity, type 2 diabetes and MASH.

Progress has also been made using small molecules that target the peroxisome proliferator-activated receptors (PPARα, PPARγ and PPARδ (PPARα/γ/δ)), a family of nuclear-acting receptors, which upon activation, improve systemic glucose and lipid metabolism[10,11]. PPARα is expressed in many glucoregulatory organs[12–17], where its activation improves hepatic lipid and cholesterol metabolism while decreasing liver fibrosis and expression of proinflammatory cytokines[18]. Selective PPARα agonists ameliorate body weight gain and adiposity in high-fat-diet (HFD)-fed mice[19], although with only limited ability to decrease body weight in rodents with already established obesity[20–22]. Activation of PPARγ improves insulin sensitivity in the adipose tissue, liver and skeletal muscle[10,11,23]. Although the mechanisms underlying these effects remain largely unknown, they are assumed to at least in part rely on the

ability of PPARγ to decrease ectopic lipid deposition by stimulating adipocyte differentiation and fatty acid uptake[10,23]. Expression of PPARγ is also found in hypothalamic nuclei that govern energy metabolism[24], and viral-mediated overexpression of hypothalamic PPARγ decreases food intake in diet-induced obese (DIO) mice[25]. In agreement with this, we recently showed that GLP-1-mediated delivery of the PPARα/γ co-agonist tesaglitazar (Tesa) decreases body weight and food intake with increased efficacy compared with GLP-1R agonism in DIO mice[26]. Nonetheless, the role of PPARγ in regulating food intake remains controversial, since studies have shown discrepant results depending on the models used and experimental conditions[10,11,23]. PPARδ shows high expression in the brain, where its activation has neuroprotective and anti-apoptotic effects in animal models of cerebral ischemia and Parkinson's Disease[27]. Germline deletion of PPARδ is embryonically lethal[28,29], but its overexpression in the adipose tissue improves lipid metabolism and leads to resistance to diet-induced obesity[30]. Although PPARα/γ co-agonists have shown metabolic benefits in clinical trials, many have been discontinued owing to adverse cardiovascular and/or renal effects[10,11,23]. The PPARα/γ/δ triple agonist lanifibranor (Lani) is currently in clinical phase 3 trials for the treatment of MASH. In phase 2b, Lani decreased liver fibrosis and MASH, but led to body weight gain, anaemia and fluid retention (peripheral oedema)[3].

Here we report the design and preclinical evaluation of a unimolecular quintuple agonist, which via covalent binding of Lani to a GLP-1R–GIPR co-agonist allows for its targeted delivery into cells that express the receptor for GLP-1 or GIP. This approach enabled the use of Lani at doses 6,898-fold lower than the dose (30 mg kg⁻¹) required preclinically to improve liver metabolism[31]. In vitro, GLP-1–GIP–Lani is indistinguishable from its GLP-1R–GIPR co-agonist backbone in relation to incretin receptor signalling and glucose-stimulated insulin secretion, and is equally effective as Lani for inducing PPARα/γ/δ target gene expression in the presence of the incretin receptors. In vivo, however, GLP-1–GIP–Lani outperforms GLP-1R–GIPR co-agonism and semaglutide to further decrease body weight, food intake and hyperglycaemia in mice with diet, or genetically-induced obesity. Consistent with the incretin-mediated delivery of the PPARα/γ/δ agonist, the metabolic effects of GLP-1–GIP–Lani were blunted in DIO mice with pharmacological or genetic inhibition of GLP-1R, GIPR or PPARδ, and vanished in double-incretin receptor knockout (DIR-KO) mice. Together, these results indicate that GLP-1–GIP–Lani has value for the treatment for obesity and type 2 diabetes.

## Development of GLP-1–GIP–Lani

We previously showed that covalent attachment of the PPARα/γ co-agonist Tesa to a pharmacokinetically optimized GLP-1R agonist enabled targeted delivery of Tesa into GLP-1R-expressing cells, leading to greater weight loss and further improvement of glucose control relative to treatment with pharmacokinetically-matched GLP-1R agonist backbone[26] (Extended Data Fig. 1a,b). Building on these data, here we assess whether the metabolic effects of such an incretin-conjugated PPAR agonist could further be enhanced by covalent tethering of Tesa to the peptide backbone of the DPP4-protected GLP-1R–GIPR co-agonist MAR709[32] (Extended Data Fig. 1c). In DIO mice, daily treatment with GLP-1–GIP–Tesa (50 nmol kg⁻¹) only moderately outperformed GLP-1–Tesa to yield greater weight loss and further suppression of food intake (Fig. 1a,b), which was paralleled by slightly greater loss of fat and lean tissue mass (Fig. 1c), but solidly enhanced glucose tolerance (Fig. 1d) and further decreased blood glucose (Fig. 1e).

We next exchanged the PPARα/γ co-agonist Tesa with Lani, a PPARα/γ/δ triple agonist that has previously been shown to ameliorate HFD-induced body weight gain[33,34] (Extended Data Fig. 1d,e). GLP-1–GIP–Lani and GLP-1R–GIPR co-agonism comparably induced Gαs recruitment (Extended Data Fig. 2a,b) and cAMP production (Fig. 1f,g) via the incretin receptors, and equally potentiated glucose-stimulated insulin secretion

in isolated mouse islets (Fig. 1h). But in contrast to GLP-1–GIP, GLP-1–GIP–Lani exhibited similar effectiveness to Lani in promoting the expression of the PPAR target gene *PDK4* in GLP-1R and GIPR-expressing HEK293T cells transfected to co-express PPARα, PPARγ or PPARδ (Fig. 1i,j). Confirming the targeting nature of the molecule, the ability of GLP-1–GIP–Lani to induce *PDK4* expression was observed only in the presence of the incretin receptors (Fig. 1i,j) and was absent in double-incretin receptor negative cells that were transfected to express PPARα, PPARγ or PPARδ (Fig. 1k). When given at a daily dose of 50 nmol kg⁻¹, GLP-1–GIP–Lani demonstrated exceptional potency in decreasing body weight of DIO mice, with placebo-corrected weight loss 2.63-fold greater relative to GLP-1–Lani after 14 days of treatment (Fig. 1l). Compared with GLP-1–Lani, GLP-1–GIP–Lani yielded greater reduction in body fat mass and food intake (Fig. 1m,n), greater improvement of glucose tolerance (Fig. 1o) and insulin sensitivity (Extended Data Fig. 2c) and further decreased blood glucose and insulin (Extended Data Fig. 2d,e). Placebo-corrected drug effects showed that GLP-1–GIP–Lani solidly outperformed GLP-1–Lani, GLP-1–Tesa and GLP-1–GIP–Tesa to further decrease body weight, fat mass and blood glucose (Extended Data Fig. 2f–h) identifying GLP-1–GIP–Lani as the lead candidate for subsequent studies.

## Comparison with GLP-1–GIP and semaglutide

In DIO mice, GLP-1–GIP–Lani dose-dependently decreased body mass, fat and lean tissue mass and food intake (Extended Data Fig. 3a–c). Reduction in fat mass was comparable at daily doses of 10 and 50 nmol kg⁻¹ (Extended Data Fig. 3b), which identified the 10 nmol kg⁻¹ dose as the lead concentration for further studies. In DIO mice, this daily dose of GLP-1–GIP–Lani solidly outperformed semaglutide to further decrease body weight, food intake and fat mass, without difference in lean body mass, but with further improved glucose tolerance, greater decrease in blood glucose and similar reduction in plasma insulin (Extended Data Fig. 3d–i). GLP-1–GIP–Lani (10 nmol kg⁻¹) also decreased body weight with superior effects compared with Lani or GLP-1R–GIPR co-agonism (Fig. 2a), and this was paralleled by greater decreased fat mass, slightly greater loss of lean mass (Fig. 2b) and further inhibition of food intake (Fig. 2c). GLP-1–GIP–Lani did not affect energy expenditure, substrate utilization or locomotor activity (Extended Data Fig. 3j–l), but outperformed GLP-1–GIP to further decrease blood glucose and glucose-stimulated insulin secretion (Fig. 2d,e). Compared with GLP-1R–GIPR co-agonism, GLP-1–GIP–Lani further improved oral glucose tolerance (Fig. 2f) and insulin sensitivity, as assessed using hyperinsulinaemic-euglycaemic clamps (Fig. 2g–i). GLP-1–GIP–Lani also suppressed endogenous glucose production with greater efficacy than GLP-1–GIP (Fig. 2j), and this was confirmed by reduced pyruvate-induced glucose production (Fig. 2k) and decreased hepatic expression of *Pcx* and *Pepck1*, the master regulators of gluconeogenesis (Fig. 2l,m). In summary, GLP-1–GIP–Lani outperforms GLP-1R–GIPR co-agonism and semaglutide to yield greater body weight loss and further improvement of glucose metabolism, with the latter being mediated by improved insulin sensitivity and enhanced suppression of endogenous glucose production.

## Transcriptomics in peripheral tissues

Bulk RNA sequencing showed that GLP-1–GIP–Lani robustly induced anti-inflammatory gene programmes in the liver and skeletal muscle (Fig. 2n,o). Compared with vehicle controls, GLP-1–GIP–Lani yielded more than 5,411 differentially expressed genes (DEGs) in the liver, compared to only 913 and 57 DEGs induced by GLP-1–GIP and Lani, respectively (Extended Data Fig. 4a,b). The transcriptional hepatic effects of GLP-1–GIP–Lani separated clearly from those of Lani and GLP-1–GIP, which was also apparent by a robust shift in principal component analysis (PCA) (Extended Data Fig. 4c). The most enriched gene sets induced by GLP-1–GIP–Lani corresponded to cell cycling and cholesterol metabolism (Extended Data Fig. 4d). Large systemic

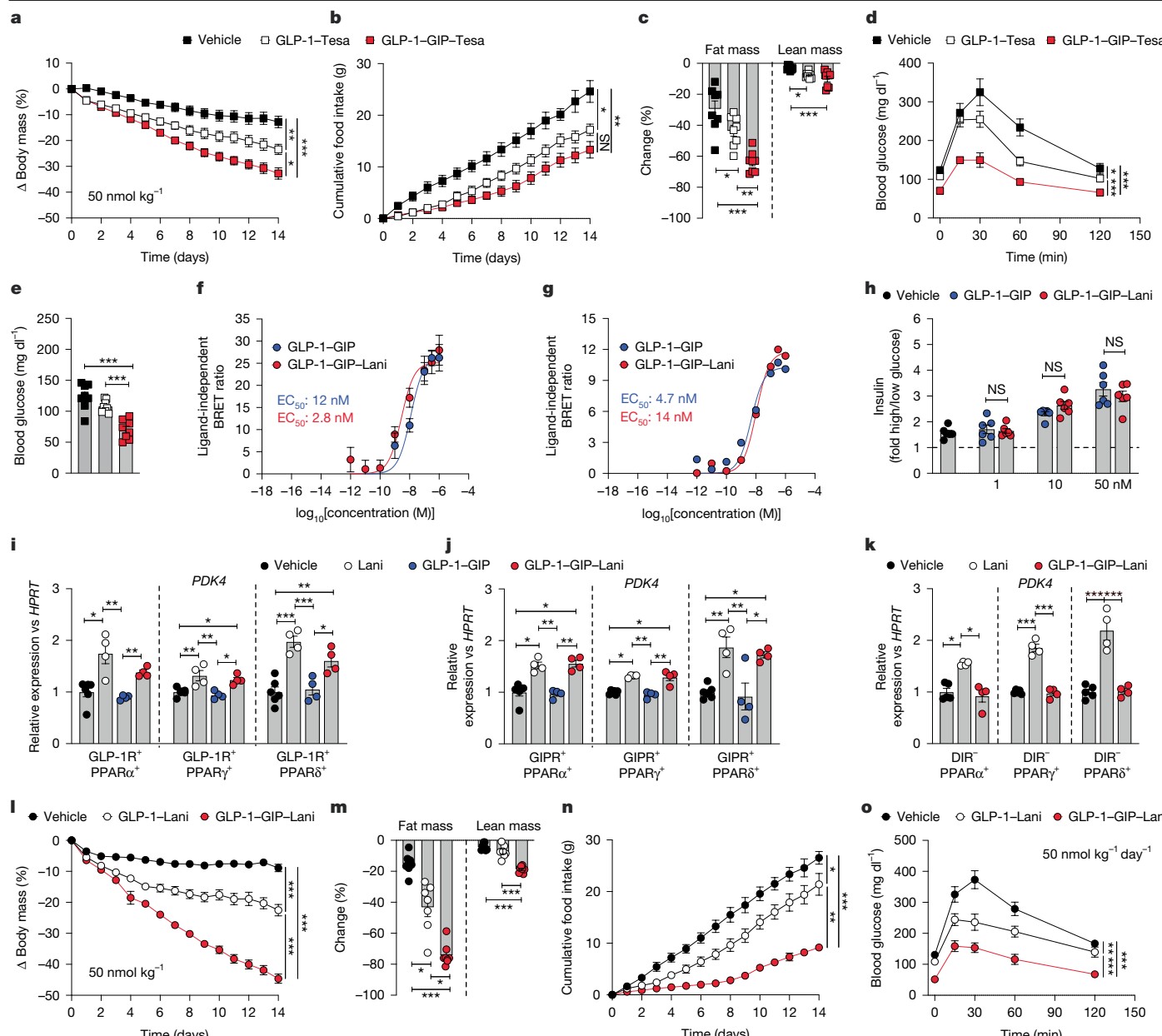

**Fig. 1 | Metabolic effects of PPAR conjugates. a–e**, Body mass (**a**), change in fat or lean mass (**b**), cumulative food intake (**c**), glucose tolerance on day 14 (**d**) and corresponding fasting blood glucose (**e**) in DIO mice treated once daily subcutaneously with vehicle or 50 nmol kg⁻¹ GLP-1–Tesa or GLP-1–GIP–Tesa for 14 days. *n* = 6–8 mice per group. **f,g**, Ligand-induced cAMP production in HEK293T cells expressing GLP-1R (**f**) or GIPR (**g**). *n* = 5 or 6 biological replicates per group. BRET, bioluminescence resonance energy transfer. **h**, Insulin stimulation index in isolated mouse pancreatic islets maintained in at 2, 8 or 20 mM glucose. *n* = 6 mice per group. **i–k**, *HPRT*-corrected expression of *PDK4* after 24 h stimulation in HEK293T cells expressing PPARα/γ/δ in cells expressing GLP-1R (**i**) or GIPR (**j**), or in double-incretin receptor negative (DIR⁻) cells. *n* = 4–6 biological replicates per group. **l–o**, Body mass (**l**), change in fat or lean mass (**m**), cumulative food intake (**n**) and glucose tolerance on day 14 (**o**) in mice treated once daily subcutaneously with vehicle or 50 nmol kg⁻¹ GLP-1–Lani or GLP-1–GIP–Lani for 14 days. *n* = 7–8 mice per group. Data were analysed using one-way ANOVA with Bonferroni post hoc multiple-comparison test (**c,e**), two-way repeated-measures ANOVA (**a,b,d,l,n,o**) or Mann–Whitney test (**h**). Data in **i–k,m** were analysed using one-way ANOVA with Bonferroni post hoc multiple-comparison test in the case of normal distribution or the Kruskal–Wallis test with uncorrected Dunns's test in the case of non-normal distribution. In **b,n**, cumulative food intake was assessed per cage in single- or double-housed mice. Data are mean ± s.e.m. *\*P* < 0.05, *\*\*P* < 0.01, *\*\*\*P* < 0.001; NS, not significant (*P* > 0.05). Exact *P* values, *n* values and detailed statistics are provided in Supplementary Tables 1 and 3 and Source Data.

effects were also observed in the epididymal white adipose tissue (eWAT), with 8,060 DEGs after treatment with GLP-1–GIP–Lani, compared to only 264 and 108 DEGs with GLP-1–GIP and Lani, respectively (Extended Data Fig. 4e–g). DEGs induced by GLP-1–GIP–Lani were mostly involved in cell cycle regulation, adipogenesis and Notch signalling (Extended Data Fig. 4h), suggesting effects on adipose tissue remodelling and inflammation. Strong effects of GLP-1–GIP–Lani were also observed in the skeletal muscle, with a distinct expression

profile and 1,715 DEGs, particularly corresponding to enhanced oxidative phosphorylation, indicating a shift towards enhanced substrate utilization and fatty acid metabolism (Extended Data Fig. 4i–l).

## Effects on adipose tissue

PPARγ agonists act not only on mature adipocytes to improve insulin sensitivity, but also on adipocyte precursors to increase body weight

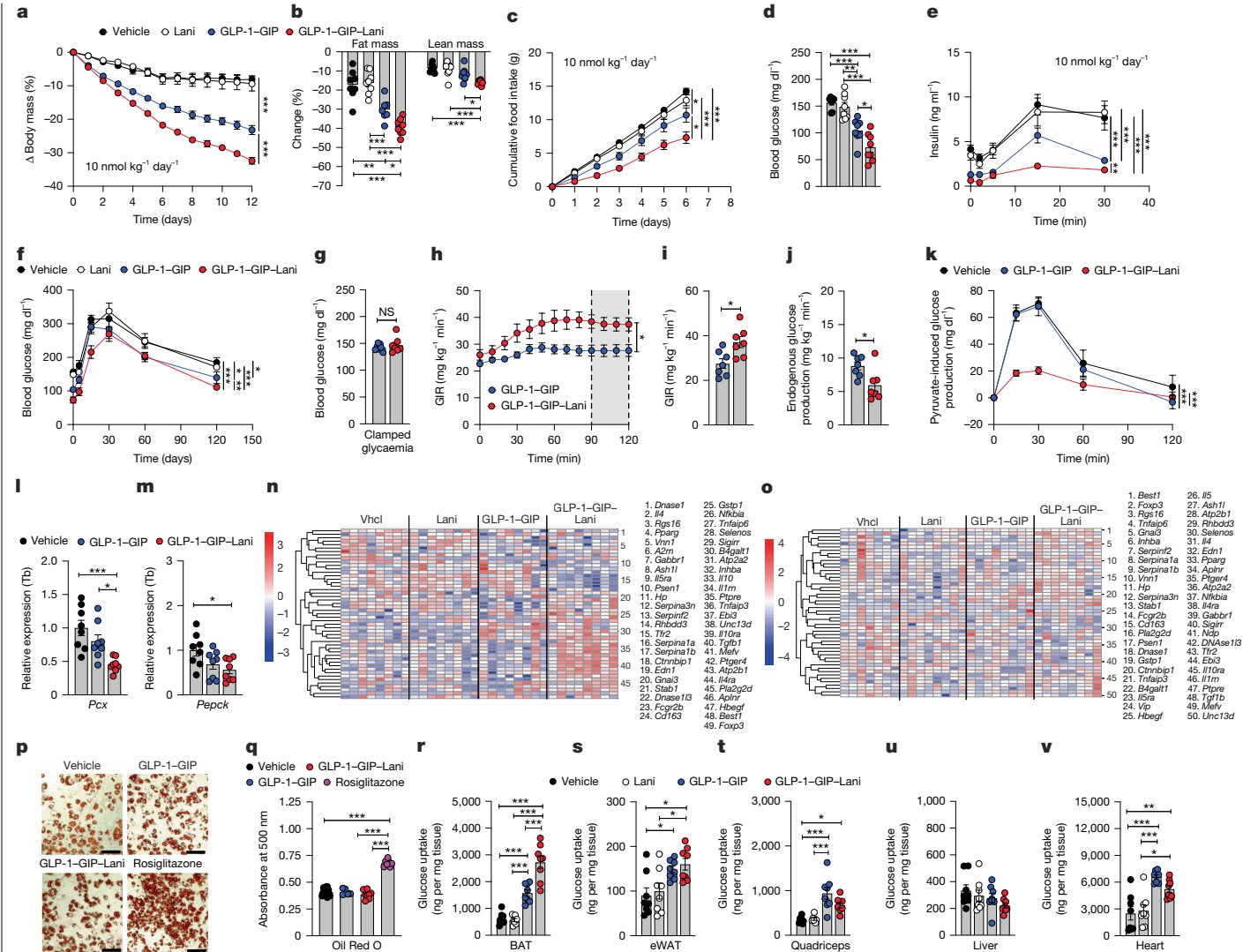

**Fig. 2 | Comparison with GLP-1–GIP. a–f**, DIO mice were treated once daily subcutaneously for 12 days. $n$ = 5–8 mice per group. 10 nmol kg$^{-1}$. **a**, Body weight. **b**, Change in fat and lean mass. **c**, Cumulative food intake. **d**, Fasting blood glucose. **e**, Glucose-stimulated insulin secretion. **f**, Oral glucose tolerance. **g–j**, Hyperinsulinaemic–euglycaemic clamps in DIO mice pretreated for 6 days via once daily subcutaneous injections of 10 nmol kg$^{-1}$ of indicated drug. $n$ = 7 per group. **g**, Clamped glycaemia. **h**, Glucose infusion rate over time. **i**, Mean glucose infusion rate at steady state (90–120 min). **j**, Endogenous glucose production. **k–m**, DIO mice were treated once daily with subcutaneous injections of vehicle or 10 nmol kg$^{-1}$ of indicated drug for 5 days. $n$ = 7–8 mice per group. **k**, Pyruvate tolerance test. **l**, Hepatic expression of *Pcx*. **m**, Hepatic expression of *Pepck1*. **n,o**, Expression of anti-inflammatory genes in liver (**n**) and skeletal muscle (**o**) of DIO mice treated once daily with subcutaneous injections of vehicle

or 10 nmol kg$^{-1}$ of indicated drug for 12 days. $n$ = 7–8 mice per group. **p,q**, Oil Red O accumulation in mouse inguinal white adipocytes differentiated for 8 days in the presence of vehicle or 50 μM GLP-1–GIP, GLP-1–GIP–Lani or rosiglitazone. $n$ = 5–12 biological replicates per group. **p**, Oil Red O staining. Scale bars, 100 μm. **q**, Absorbance at 500 nm. **r–v**, Tissue-selective glucose uptake in BAT (**r**), eWAT (**s**), quadriceps (**t**), liver (**u**) and heart (**v**) of DIO mice treated once daily with subcutaneous injections of vehicle or 10 nmol kg$^{-1}$ of indicated drug for 6 days. $n$ = 7–8 mice per group. Data were analysed using one-way ANOVA with Bonferroni post hoc multiple-comparison test (**b,d,l,m,q–v**), two-way repeated-measures ANOVA (**a,c,e,f,k**), two-sided Wald test (**n,o**) or unpaired two-sided t test (**g,i,j**). In **c**, cumulative food intake was assessed per cage in single- or double-house mice. Data are mean ± s.e.m. Exact $P$ values, $n$ values and detailed statistics are provided in Supplementary Tables 1 and 3 and Source Data.

via stimulation of adipocyte differentiation[10,11,23]. Consistent with these activities, GIPR is expressed in mature adipocytes but not in preadipocytes[35]. We found that GLP-1–GIP–Lani, in contrast to rosiglitazone, was unable to induce adipocyte differentiation (Fig. 2p,q), but resulted in greater glucose uptake relative to GLP-1–GIP into the brown adipose tissue (BAT) (Fig. 2r) and similar glucose uptake into eWAT, skeletal muscle, liver and heart (Fig. 2s–v). Consistent with the notion that glucose uptake by BAT reflects insulin sensitivity rather than energy expenditure[36,37], GLP-1–GIP–Lani and GLP-1–GIP did not affect expression of thermogenic genes in the BAT (Extended Data Fig. 5a–d) and had no acute or chronic effect on mitochondrial functions in differentiated BAT primary cells, including mitochondrial respiration, proton production rate, proton leak respiration, coupling

efficiency, maximal substrate oxidation, cellular respiration, non-mitochondrial and mitochondrial respiration, ATP-linked respiration, glycolysis, glycolytic proton production rate and glycolytic and oxidative phosphorylation-linked ATP production (Extended Data Fig. 5e–s). Collectively, these data indicate that the enhanced glycaemic effects of GLP-1–GIP–Lani originate from its ability to improve insulin sensitivity through decreased inflammation in key glucometabolic tissues, leading to enhanced suppression of endogenous glucose production and increased glucose uptake into these tissues. Moreover, GLP-1–GIP–Lani protects from the adverse effects of PPAR that promote body weight gain via stimulation of adipocyte differentiation, while preserving its activity on mature adipocytes to facilitate glucose uptake.

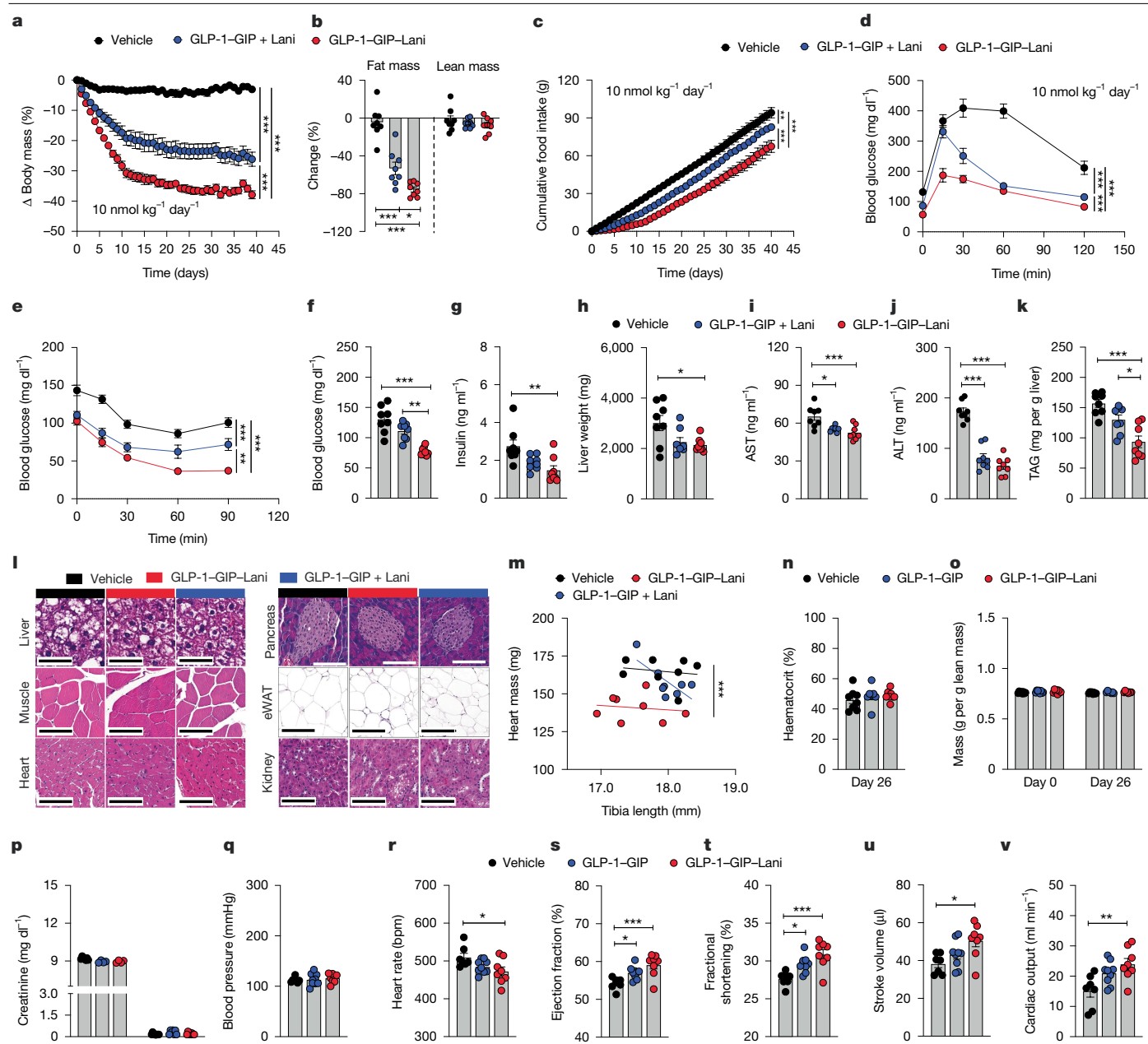

**Fig. 3 | Chronic effects in DIO mice. a–m**, DIO mice were treated once daily with subcutaneous injections of vehicle or 10 nmol kg⁻¹ GLP-1–GIP–Lani or GLP-1–GIP plus Lani for 39 days. *n* = 7–8 mice per group. **a**, Body mass. **b**, Change in fat and lean mass **c**, Cumulative food intake. **d**, Glucose tolerance test on day 9. **e**, Insulin tolerance test on day 40. **f**, Fasting blood glucose. **g**, Fasting plasma insulin. **h**, Liver weight. **i**, Plasma aspartate aminotransferase (AST). **j**, Plasma alanine aminotransferase (ALT). **k**, Liver triglycerides (TAG). **l**, Representative histology of liver, muscle, heart, pancreas, eWAT and kidney. **m**, Heart mass against tibia lengths. **n–p**, Renal effects were assessed in DIO mice treated once daily with subcutaneous injections of vehicle or 10 nmol kg⁻¹ GLP-1–GIP or GLP-1–GIP–Lani for 26 days. *n* = 4–8 mice per group. **n**, Haematocrit. **o**, Total body fluid. **p**, Plasma and urinary creatinine. **q–v**, Cardiovascular end-points were assessed in DIO mice treated once daily with subcutaneous injections of vehicle

or 10 nmol kg⁻¹ GLP-1–GIP or GLP-1–GIP–Lani for 14 days. *n* = 7–9 mice per group. **q**, Blood pressure. **r**, Heart rate. **s**, Ejection fraction. **t**, Fractional shortening. **u**, Stroke volume. **v**, Cardiac output. Data were analysed using one-way ANOVA with Bonferroni post hoc multiple-comparison test (**b**,**f**,**i**–**k**,**n**,**o**,**r**–**v**), two-way repeated-measures ANOVA (**a**,**c**–**e**), Kruskal–Wallis test with uncorrected Dunns's test (**g**,**h**,**q**) or analysis of covariance (ANCOVA) (**m**). Data in **p** were analysed using one-way ANOVA with Bonferroni post hoc multiple-comparison test in the case of normal distribution or with the Kruskal–Wallis test with uncorrected Dunns's test in case of non-normal distribution. In **c**, cumulative food intake was assessed per cage in single- or double-house mice. Data are mean ± s.e.m. Exact *P* values, *n* values and detailed statistics are provided in Supplementary Tables 1 and 3 and Source Data.

## Assessment of drug safety

GLP-1–GIP–Lani also decreased body weight, fat mass and food intake with increased effectiveness compared with co-treatment with GLP-1–GIP plus Lani, without affecting lean tissue mass (Fig. 3a–c), but with further improved glucose tolerance and insulin sensitivity and further

decreased blood glucose and insulin (Fig. 3d–g). Relative to GLP-1R–GIPR co-agonism, GLP-1–GIP–Lani similarly decreased liver mass (Fig. 3h) and plasma levels of aspartate aminotransferase and alanine aminotransferase (Fig. 3i,j) and further decreased liver triglycerides (Fig. 3k). No histological alterations were observed in liver, muscle, heart, eWAT or kidney (Fig. 3l). GLP-1–GIP–Lani nonetheless decreased

HFD-induced heart hypertrophy without causing anaemia, fluid retention or changes in urinary or plasma creatinine (Fig. 3m–p). GLP-1–GIP–Lani further improved cardiac performance with increased effectiveness versus GLP-1–GIP, with no changes on blood pressure, but decreased heart rate and increased ejection fraction, fractional shortening, stroke volume and cardiac output relative to vehicle controls (Fig. 3q–v). In contrast to co-treatment with GLP-1–GIP plus Lani, the GLP-1–GIP–Lani conjugate also fully prevented body weight gain in obesity-prone leptin receptor-deficient db/db mice (Extended Data Fig. 6a), a model in which tirzepatide, semaglutide and retatrutide showed only modest ability to prevent the establishment of obesity[38,39]. GLP-1–GIP–Lani decreased fat mass and food intake in db/db mice with greater effectiveness than co-treatment with GLP-1–GIP plus Lani, without notable changes in lean tissue mass (Extended Data Fig. 6b), but further decreased food intake and blood glucose and further improved glucose tolerance and insulin sensitivity (Extended Data Fig. 6d–h). Collectively, these data show that GLP-1–GIP–Lani decreases body weight, food intake and hyperglycaemia with greater effectiveness than GLP-1–GIP plus Lani, without detrimental effects on the renal system, and enhanced efficacy compared with GLP-1–GIP alone for improving liver and cardiovascular health.

## Drug effects in lean mice

Single peripheral bolus administration (10 nmol kg$^{-1}$) of GLP-1–GIP and GLP-1–GIP–Lani both induced conditioned taste avoidance (CTA) in lean wild-type mice, with GLP-1–GIP–Lani having a slightly greater effect compared with GLP-1–GIP, and without inducing CTA in glutamatergic (*Vglut2-cre*) *Glp1r*-knockout mice (*Vglut2* is also known as *Slc17a6*; Extended Data Fig. 6i). Nonetheless, GLP-1–GIP–Lani did not affect body weight, body composition, food intake or plasma levels of glucose, insulin, triglycerides or cholesterol in lean mice (Extended Data Fig. 6j–p). Collectively, GLP-1–GIP–Lani outperforms GLP-1R–GIPR co-agonism to improve glucometabolic health in DIO mice, with similar tolerability and no risk of hypoglycaemia or weight loss in lean mice. The absence of weight loss despite induction of CTA in lean mice further indicates that enhanced nausea is unlikely to account for its body weight-reducing effects in DIO mice.

## Effects in target receptor-knockout mice

Weight loss and food intake suppression effects of GLP-1–GIP–Lani were impaired in DIO *Vglut2/Glp1r*-knockout mice, whereas the effects of GLP-1–GIP remained largely preserved (Fig. 4a,b). Weight loss induced by GLP-1–GIP–Lani was also diminished in DIO *Gipr*-knockout mice (Fig. 4c), confirming the contribution of both incretin receptors. Corroborating a functional role of GIPR, GLP-1–GIP–Lani decreased body weight and food intake with increased efficacy compared with GLP-1–Lani, but with equal efficacy to co-therapy of GLP-1–Lani and a long-acting GIPR agonist (acyl-GIP) (Fig. 4d,e). However, despite similar decrease in body weight, GLP-1–GIP–Lani outperformed GLP-1–GIP plus Lani co-therapy to yield greater improvement of glucose tolerance (Fig. 4f), suggesting that GLP-1–GIP–Lani also improves glycaemia via the GIPR-targeted activity of the PPAR agonist. Consistent with this idea, we found that the blood glucose-lowering effect of GLP-1–GIP–Lani was abolished after selective antagonization of PPARδ using GSK3787 (Fig. 4g). Of note, although inhibition of PPARδ diminished the glucose-lowering effect of GLP-1–GIP–Lani, antagonization of PPARδ did not ameliorate weight loss induced by either GLP-1–GIP–Lani or the selective PPARδ agonist GW501516 (Fig. 4h), suggesting that GW501516 and the PPAR agonist moiety of GLP-1–GIP–Lani also decrease body weight independently of PPARδ. In summary, these data show that GLP-1–GIP–Lani decreases body weight via both incretin receptors, while further improving glucose control via PPARδ. Functionally verifying the incretin receptor-dependent nature of the molecule, GLP-1–GIP–Lani decreased

body weight, food intake and blood glucose in DIO wild-type mice, and these effects were absent in DIO DIR-knockout mice (Fig. 4i–m).

## Comparison with weight-matched controls

To further confirm whether GLP-1–GIP–Lani improves energy and glucose metabolism beyond GLP-1R–GIPR co-agonism, we treated DIO mice for 12 days with either vehicle, GLP-1–GIP or GLP-1–GIP–Lani. As an additional control, we further included a group of GLP-1–GIP-treated mice that were additionally food-restricted to match the body weight of the GLP-1–GIP–Lani-treated mice. These two groups thus differed only in the targeted delivery of Lani, but were otherwise matched by body weight and treated equally with GLP-1R–GIPR co-agonist. Although mice treated with GLP-1–GIP–Lani had the same weight loss and body composition as their food-restricted body weight-matched GLP-1–GIP-treated controls (Extended Data Fig. 7a,b), GLP-1–GIP–Lani yielded further improved glucose tolerance and greater decreased blood glucose with similar decreases of fasting insulin (Extended Data Fig. 7c–e). Of note, starting from day 6 onwards, the food-restricted (weight-matched) GLP-1–GIP-treated mice required a smaller amount of daily food intake relative to the GLP-1–GIP–Lani-treated mice to match their body weight (Extended Data Fig. 7f), indicating that GLP-1–GIP–Lani decreases body weight via food intake-dependent and independent mechanisms. In summary, these data reiterate that at least some of the glycaemic benefit of GLP-1–GIP–Lani is body weight-independent and is not observed with GLP-1R–GIPR co-agonism, even after correction for body weight.

## Effects in GIPR transgenic mice

Since GIPR agonism[40] and GLP-1–GIP–Lani (Fig. 2r,s) promote glucose uptake into adipose tissue, we next assessed the metabolic effects of low-dose GLP-1–GIP–Lani (5 nmol kg$^{-1}$) in DIO transgenic mice with adipose-specific overexpression of GIPR. Consistent with previous reports[41], such adipose GIPR transgenic mice show substantially reduced body weight upon doxycycline-induced overexpression of GIPR in the adipose tissue relative to wild-type controls (Extended Data Fig. 7g). Notably, however, whereas weight loss induced by GLP-1–GIP–Lani and GLP-1R–GIPR co-agonism was much stronger in GIPR transgenic mice relative to wild-type controls, they both equally induced weight loss in GIPR transgenic mice (Extended Data Fig. 7g), but GLP-1–GIP–Lani was slightly more effective at improving control of glucose levels (Extended Data Fig. 7h,i). Whether and how enhanced GIPR activity in adipocytes contributes to glycaemic effects of the GLP-1–GIP–Lani conjugate warrants clarification; however, it is clear that GLP-1–GIP–Lani produces greater absolute weight loss in mice with adipocyte-specific overexpression of GIPR.

## Acute peripheral proteomic effects

After 7 h of single subcutaneous administration, GLP-1–GIP–Lani induced robust proteomic changes in the pancreas, with only minor effects in the eWAT, liver and skeletal muscle. PCA of the pancreatic proteome showed that GLP-1–GIP–Lani induced a clear shift relative to GLP-1–GIP (Extended Data Fig. 8a), with induction of 778 proteins compared to 304 proteins induced by GLP-1–GIP (Extended Data Fig. 8b,c). Most of the proteins were uniquely affected by GLP-1–GIP–Lani, suggesting a dominant effect of the Lani moiety via incretin receptor-dependent transport (Extended Data Fig. 8b–g). Proteins that were upregulated by GLP-1–GIP–Lani (cluster 2) were enriched in functions related to cell division, DNA binding, transcriptional regulation and mRNA metabolism, whereas proteins that were downregulated by GLP-1–GIP–Lani (cluster 4) were associated with mitochondrial inner membrane components, oxidative phosphorylation, tricarboxylic acid cycle, NAD-related processes and protein biosynthesis (Extended Data

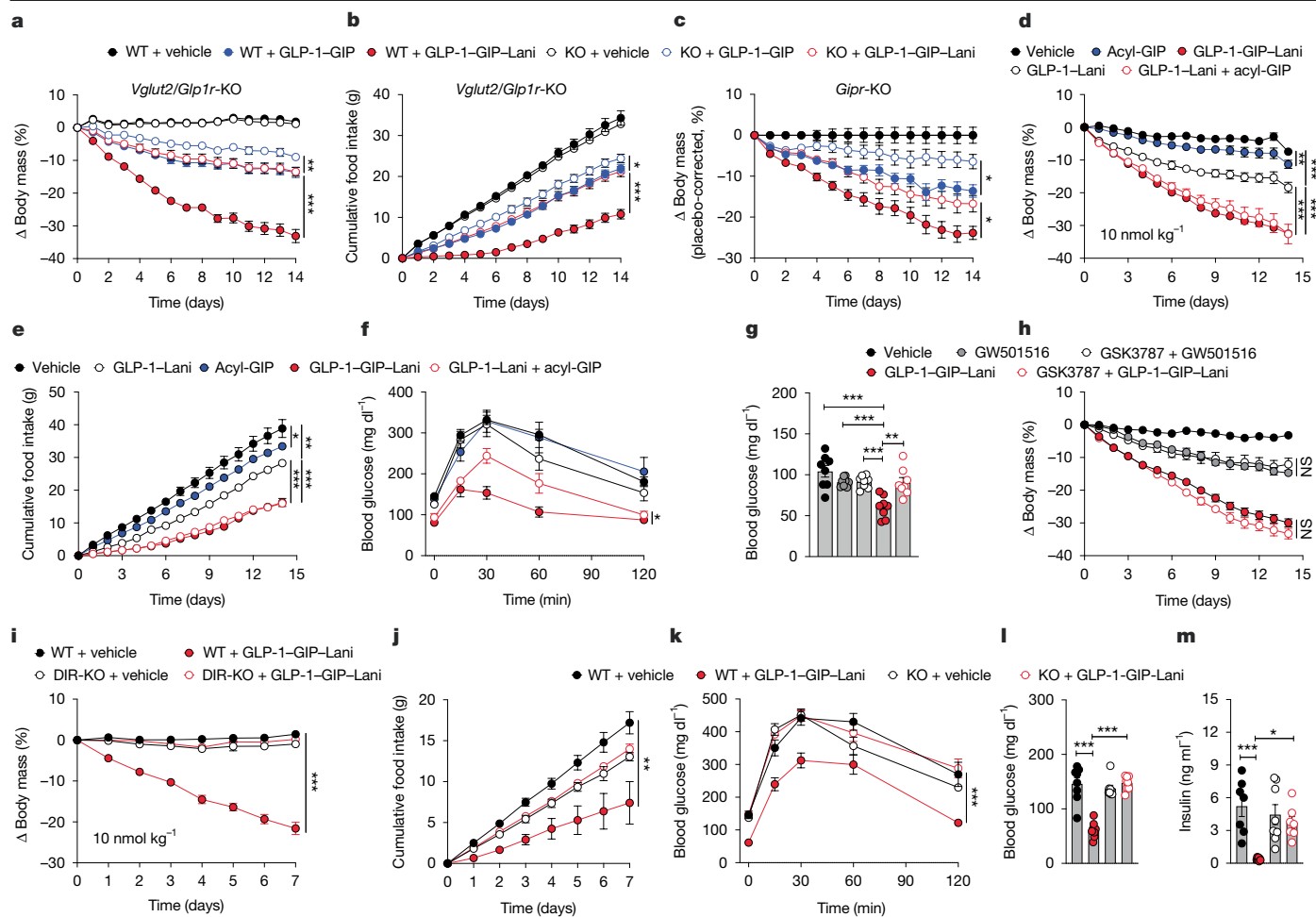

**Fig. 4 | Effects of inhibition of GLP-1R, GIPR or PPARδ in mice. a,b**, DIO wild-type (WT) and *Vglut2*/*Glp1r*-knockout (KO) mice were treated once daily with subcutaneous injections of vehicle or 10 nmol kg⁻¹ GLP-1-GIP or GLP-1-GIP-Lani for 14 days. $n = 8$–9 mice per group. **a**, Body weight. **b**, Cumulative food intake. **c**, Change in body weight of DIO wild-type and *Gipr*-knockout mice treated once daily with subcutaneous injections of vehicle or 10 nmol kg⁻¹ GLP-1-GIP or GLP-1-GIP-Lani for 14 days. $n = 7$–8 mice per group. **d**–**f**, DIO mice were treated once daily with subcutaneous injections of vehicle or 10 nmol kg⁻¹ acyl-GIP, GLP-1-Lani, GLP-1-GIP-Lani or GLP-1-Lani plus acyl-GIP for 14 days. $n = 7$–8 mice per group. **d**, Body weight. **e**, Cumulative food intake. **f**, Glucose tolerance test at day 14. **g,h**, DIO mice were treated once daily for 14 days with intraperitoneal injections of the PPARδ agonist GW501516 (11 μmol kg⁻¹), subcutaneous injections of GLP-1-GIP-Lani (10 nmol kg⁻¹) or vehicle, or pretreated intraperitoneally with the PPARδ

antagonist GSK3787 (25.5 μmol kg⁻¹) 4 h before administration of GW501516 or GLP-1-GIP-Lani. $n = 6$–8 mice per group. **g**, Ad libitum blood glucose at day 11. **h**, Body weight. **i**–**m**, DIO wild-type and DIR-knockout mice were treated once daily with subcutaneous injections of vehicle or 10 nmol kg⁻¹ GLP-1-GIP-Lani for 7 days. $n = 7$–8 mice per group. **i**, Body weight. **j**, Cumulative food intake. **k**, Glucose tolerance test at day 7. **l**, Fasting blood glucose. **m**, Fasting plasma insulin. Data were analysed using one-way ANOVA with Bonferroni post hoc multiple-comparison test (**g**,**m**), two-way repeated-measures ANOVA (**a**–**f**,**h**–**k**) or Kruskal–Wallis test with uncorrected Dunns's test (**l**). Cumulative food intake (**e**,**j**) was assessed per cage in single- or double-house mice. Data are mean ± s.e.m. Exact *P* values, *n* values and detailed statistics are provided in Supplementary Tables 1 and 3 and Source Data.

Fig. 8c–g). In the eWAT, no group separation was observed in the PCA, and only five and three proteins were upregulated by GLP-1-GIP-Lani and GLP-1-GIP, respectively (Extended Data Fig. 8h–j). Similarly, no group separation was found in the liver, with only 20 proteins induced by GLP-1-GIP-Lani and 17 induced by GLP-1-GIP (out of which 16 overlapped), and no clear pathway enrichment (Extended Data Fig. 8k–m). No differences were seen in the PCA of the skeletal muscle proteome, with only seven and six DRPs induced by GLP-1-GIP-Lani and GLP-1-GIP, respectively (Extended Data Fig. 8n–p).

## Acute drug effects in the CNS

After single subcutaneous administration, GLP-1-GIP-Lani and GLP-1-GIP induced equal amounts of neuronal FOS activity in the arcuate nucleus, area postrema and nucleus tractus solitarius (Fig. 5a–f). Similar to liraglutide[42], semaglutide[43] and acyl-GIP[5], GLP-1-GIP-Lani and GLP-1-GIP showed no ability to cross the blood–brain barrier,

as assessed in vitro using an established human blood–brain barrier model[44] (Fig. 5g). Nonetheless, 7 h after single subcutaneous administration, GLP-1-GIP-Lani induced 350 proteins in the brainstem relative to vehicle compared with only 94 proteins induced by GLP-1-GIP, and this was accompanied by a marked shift in the PCA of the proteome (Fig. 5h–j). Consistent with the nuclear action of Lani, most proteins downregulated by GLP-1-GIP-Lani (cluster 1) were associated with nuclear processes such as RNA processing, neurotransmitter receptor internalization and chromosome organization, whereas proteins upregulated by GLP-1-GIP-Lani were mainly linked to neurotransmitter receptor internalization (Fig. 5j and Extended Data Fig. 9a). In the hypothalamus, GLP-1-GIP-Lani induced 530 proteins, whereas GLP-1-GIP induced 606 proteins, with no clear separation in the PCA (Fig. 5k–m). Proteins downregulated by GLP-1-GIP-Lani (cluster 1) were mainly associated with calcium and tubulin binding, ion channel and phosphatase regulation and protein folding, whereas proteins upregulated by GLP-1-GIP-Lani (cluster 3) were linked to oxidative

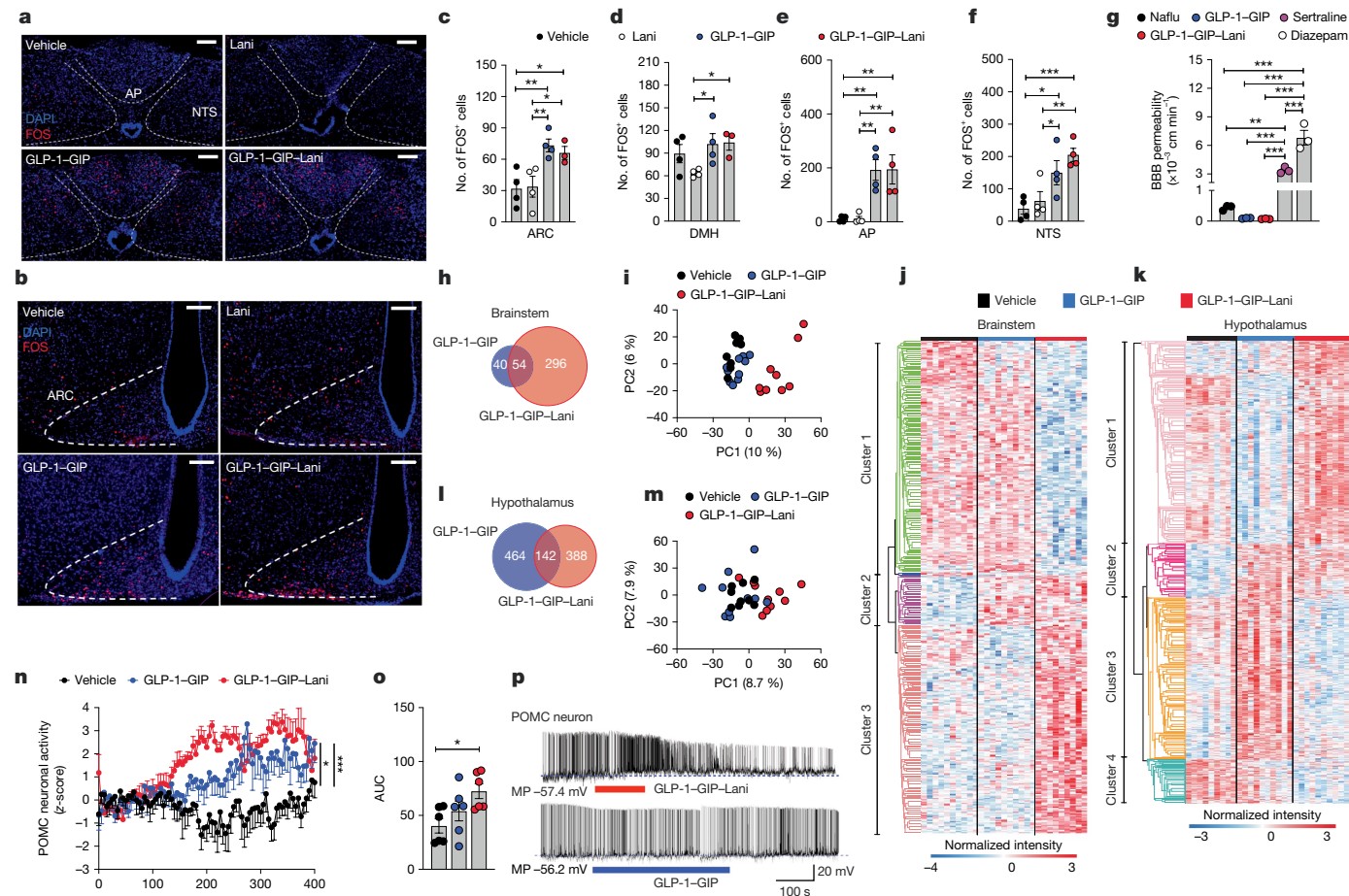

**Fig. 5 | Effects on the central nervous system. a–f**, DIO mice were treated with a single subcutaneous injection of vehicle or 50 nmol kg⁻¹ GLP-1–GIP, GLP-1–GIP–Lani or Lani. $n$ = 3–4 per group. **a,b**, Representative image of FOS in brainstem (**a**) and hypothalamus (**b**). Scale bars, 100 μm. **c–f**, Number of FOS-positive neurons in arcuate nucleus (ARC) (**c**), dorsomedial hypothalamus (DMH) (**d**), area postrema (AP) (**e**) and nucleus tractus solitarius (NTS) (**f**). **g**, In vitro assessment of blood-brain barrier (BBB) permeability in CD34⁺ endothelial cells derived from haematopoietic stem cells co-cultured with brain pericytes. $n$ = 3 biological replicates per group. **h–m**, DIO mice were treated with a single subcutaneous injection of vehicle or 100 nmol kg⁻¹ GLP-1–GIP or GLP-1–GIP–Lani, and brainstem (**h–j**) and hypothalamus (**k–m**) were collected 7 h post-injection. $n$ = 9–10 per group. **h,l**, Venn diagram showing the number of differentially regulated proteins in the brainstem (**h**) and hypothalamus (**l**). **i,m**, PCA of the proteome in brainstem (**i**) and hypothalamus (**m**). **j,k**, Supervised hierarchical clustering of $z$-scored intensities of significantly changed proteins in the brainstem (**j**) and hypothalamus (**k**). **n,o**, Fibre-photometric assessment of POMC activity in *Pomc-cre* mice treated with a single subcutaneous injection of vehicle or 10 nmol kg⁻¹ GLP-1–GIP or GLP-1–GIP–Lani. $n$ = 6 per group. **n**, POMC neuronal activity over time. **o**, Corresponding area under curve. **p**, Electrophysiological recording of POMC neuronal activity in *Pomc*-GFP mice stimulated with 2 nM GLP-1–GIP or GLP-1–GIP–Lani. $n$ = 6 mice per group. Data were analysed using one-way ANOVA with uncorrected Fisher's least significant difference test (**c–f**), one-way ANOVA with Bonferroni's multiple comparisons test (**g,o**), two-way repeated-measures ANOVA (**n**), one-way ANOVA (false discovery rate (FDR) < 0.05) with Tukey's honest significant difference post hoc test (**h,l**) or one-way ANOVA with FDR < 0.05 and unsupervised hierarchical clustering of $z$-scored log₂ label-free quantification intensities (**j,k**). Data are mean ± s.e.m. Exact $P$ values, $n$ values and detailed statistics are provided in Supplementary Tables 1 and 3 and Source Data.

and lipid metabolism, ribosomal activity and amino acid and carbohydrate metabolism (Fig. 5k and Extended Data Fig. 9b). Although proteomic changes induced by GLP-1–GIP–Lani were less robust in the hypothalamus relative to hindbrain, GLP-1–GIP–Lani more robustly induced POMC neuronal activity compared with GLP-1–GIP, as assessed in vivo using fibre-photometric assessment of POMC neuronal activity in *Pomc-cre* mice, and ex vivo using whole-cell patch recordings in *Pomc*-GFP mice (Fig. 5n–p).

## Discussion

Here we report the development of a unimolecular quintuple agonist that combines the body weight and blood glucose-lowering effects of GLP-1–GIP co-agonism with the insulin-sensitizing and anti-inflammatory properties of the nuclear-acting PPARα/γ/δ triple agonist Lani through targeted delivery into cells expressing GLP-1R and/or GIPR.

In vitro, GLP-1–GIP–Lani had indistinguishable effects on incretin receptor signalling compared with its GLP-1–GIP co-agonist backbone, and this was verified by equal potentiation of glucose-stimulated insulin secretion in isolated mouse islets (Fig. 1f–h), and similar induction of FOS neuronal activity in the brainstem and hypothalamus (Fig. 5a–f). Collectively, these data indicate that GLP-1–GIP–Lani is neither inferior nor superior to its GLP-1R–GIPR co-agonist backbone in relation to GLP-1R or GIPR signalling and action. However, in contrast to GLP-1–GIP, GLP-1–GIP–Lani enhanced PPAR target gene expression in the presence of the incretin receptors (Fig. 1i,j), without notable effects on PPAR-expressing cells that lack incretin receptors (Fig. 1k).

In DIO mice, GLP-1–GIP–Lani decreased body weight, food intake and hyperglycaemia with improved efficacy compared with GLP-1–GIP, semaglutide or Lani (Fig. 2a–i). Improvement of systems metabolism by GLP-1–GIP–Lani was mechanistically linked to enhanced insulin sensitivity (Fig. 2g–i) and improved liver health, leading to decreased

hepatic inflammation (Fig. 2n), suppression of endogenous glucose production (Fig. 2j–m) and enhanced glucose uptake into key glucoregulatory tissues (Fig. 2r). Although the liver and the skeletal muscle do not express GIPR or GLP-1R[40,45], and are thus not directly targeted by GLP-1–GIP–Lani, hepatic and skeletal muscle effects are nonetheless expected, given that the molecule decreases body weight and fat mass and improve glucose metabolism, which indirectly also improve metabolism in liver and skeletal muscle. Consistent with such indirect effects is the observation that GLP-1–GIP–Lani exhibits minimal acute effects on the liver and skeletal muscle proteome (Extended Data Fig. 8k–p) but profoundly changes the liver and skeletal muscle transcriptome after chronic treatment (Extended Data Fig. 4a–d,i–l). Our data are thus in line with reports showing that semaglutide and tirzepatide ameliorate liver fibrosis in individuals with MASH[4,46], and suggest the potential for similar beneficial effects of GLP-1–GIP–Lani.

Of note, Lani is currently in phase 3 clinical development for the treatment of liver fibrosis and MASH. However, to improve liver metabolism[31,47,48], Lani requires daily doses of 30 mg kg$^{-1}$ (68.98 μmol kg$^{-1}$)—approximately 6,900-fold higher than the 10 nmol kg$^{-1}$ dose used here. This not only emphasizes the potential therapeutic advancement of GLP-1–GIP–Lani relative to Lani as a stand-alone therapy but also implies that such a low dose can be expected to be received with favourable tolerability. In line with this assumption, we demonstrated that GLP-1–GIP–Lani does not cause pathological alterations in peripheral tissues and does not cause heart hypertrophy, fluid retention or renal impairment (Fig. 3l–p), but rather improves liver and cardiovascular health with slightly greater efficacy than GLP-1R–GIPR co-agonism (Fig. 3q–v and Extended Data Fig. 4a–l).

Of note, although GLP-1–GIP–Lani solidly outperforms GLP-1–GIP co-agonism or semaglutide to yield greater decreases of body weight, food intake and hyperglycaemia, it induces equal amounts of FOS neuronal activation in the brainstem and hypothalamus (Fig. 5a–f) despite substantial differences in regulation of hypothalamic/brainstem gene programmes (Fig. 5j,k). Assessment of drug effects in mice with genetic or pharmacological inhibition of GLP-1R, GIPR or PPARδ showed that GLP-1–GIP–Lani decreases body weight via both incretin receptors, while further improving glucose metabolism via PPARδ (Fig. 4a–c,g). Along with reports showing that GLP-1R agonists induce hypothalamic POMC activity[49] and our observation that weight loss induced by GLP-1–GIP–Lani is markedly diminished in *Vglut2/Glp1r*-knockout mice (Fig. 4a), these data collectively suggest that GLP-1–GIP–Lani enhances body weight loss relative to GLP-1–GIP by further accelerating POMC neuronal activity via glutamatergic GLP-1R neurons.

Limitations of our study include that delineation of the mechanisms underlying improvement of systems metabolism by GLP-1–GIP–Lani is challenging, not only because mice with germline deletion of PPARγ or PPARδ are embryonically lethal[28,29,50], but also because targeting of both incretin receptors along with PPARα, PPARγ or PPARδ using conditional triple, quadruple or quintuple-knockout mice is beyond the possibilities of most scientific laboratories. Furthermore, there remains great uncertainty related to how and where PPAR agonists act to regulate energy and glucose metabolism[51]. Accordingly, whereas PPARα agonists (fibrates) are classically assumed to improve lipid and cholesterol metabolism via their action on the liver, agonists at PPARγ (thiazolidinediones) are assumed to improve insulin sensitivity by enhancing adipose tissue differentiation and fatty acid uptake[10,11,23]. Nonetheless, this classical view has been challenged by the development of thiazolidinediones, which possess potent insulin-sensitizing properties despite having very low to absent PPAR-binding affinity[51]. Among the most notable are MSDC-0160 (PNU-91325) and MSDC-0602K, which show potent insulin-sensitizing effects in preclinical[52–54] and clinical[53,55] studies despite exhibiting much reduced ability to bind and signal via PPARγ[51]. Collectively, these studies emphasize the difficulties of studying how and where GLP-1–GIP–Lani acts to improve body weight loss and insulin sensitivity. The lack of commonly available antibodies to detect GIPR is another limitation that hampers immunohistochemical analysis of potentially relevant drug targets in the brain. Future studies are warranted to clarify the spatiotemporal mechanisms by which GLP-1–GIP–Lani separates from GLP-1–GIP co-agonism to further decrease body weight and food intake. Although the shown enhancement of POMC neuronal activity is likely to contribute to the observed increased weight loss with GLP-1–GIP–Lani relative to GLP-1–GIP, mechanisms in the hindbrain are also known to account for GLP-1-induced weight loss, and may thus also have a role in the increased weight loss induced by GLP-1–GIP–Lani relative to GLP-1–GIP. Consistent with this assumption is the observation that GLP-1–GIP–Lani has distinct effects on the hindbrain proteome compared with GLP-1–GIP (Fig. 5h,i,k).

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

[1]Institute for Diabetes and Obesity, Helmholtz, Munich, Germany. [2]German Center for Diabetes Research (DZD), Munich, Germany. [3]Institute of Physiotherapy and Health Sciences, Academy of Physical Education, Katowice, Poland. [4]Department of Pharmacology and Cancer Biology, Duke University, Durham, NC, USA. [5]Duke Molecular Physiology Institute, Department of Medicine, Division of Endocrinology, Duke University, Durham, NC, USA. [6]UR 2465, Laboratoire de la Barrière Hémato-Encéphalique (LBHE), Université Artois, Lens, France. [7]Indiana Biosciences Research Institute, Indianapolis, IN, USA. [8]Novo Nordisk Research Center Indianapolis, Indianapolis, IN, USA. [9]Novo Nordisk Foundation Center for Basic Metabolic Research, Faculty of Health and Medical Sciences, University of Copenhagen, Copenhagen, Denmark. [10]Core Facility Pathology and Tissue Analytics, Helmholtz, Munich, Germany. [11]Medizinische Klinik und Poliklinik IV, Klinikum der Universität, Ludwig-Maximilians-Universität München, Munich, Germany. [12]Department of Comparative Medicine, Yale University School of Medicine, New Haven, CT, USA. [13]Institute of Computational Biology, Helmholtz Munich, Munich, Germany. [14]Department of Molecular Biosciences, The Wenner-Gren Institute, Stockholm University, Stockholm, Sweden. [15]Department of Pharmacology, Experimental Therapy and Toxicology, Institute for Experimental and Clinical Pharmacology and Pharmacogenomic, Eberhard Karls University, Interfaculty Center of Pharmacogenomic and Drug Research, Tübingen, Germany. [16]Institute for Cardiovascular Prevention (IPEK), University Hospital, LMU Munich, Munich, Germany. [17]Touchstone Diabetes Center, The University of Texas Southwestern Medical Center, Dallas, TX, USA. [18]Department of Physiology, Faculty of Medical Sciences in Katowice, Medical University of Silesia, Katowice, Poland. [19]CIMUS, University of Santiago de Compostela, Instituto de Investigación Sanitaria, Santiago de Compostela, Spain. [20]Department of Pharmacology and Systems Physiology, University of Cincinnati College of Medicine, Cincinnati, OH, USA. [21]Research Unit Neurobiology of Diabetes, Helmholtz Munich, Neuherberg, Germany. [22]Division of NeuroBiology of Diabetes, TUM School of Medicine and Health, Technical University Munich, Munich, Germany. [23]Institute for Diabetes and Cancer (IDC), Helmholtz Diabetes Center (HDC), Helmholtz Zentrum München (Helmholtz Munich), Munich, Germany. [24]DZHK (German Centre for Cardiovascular Research) Partner Site, Munich Heart Alliance, Munich, Germany. [25]Department of Surgery, University of Michigan, Ann Arbor, MI, USA. [26]Department of Biochemistry, Cardiovascular Research Institute Maastricht (CARIM), Maastricht University, Maastricht, The Netherlands. [27]Munich Cluster for Systems Neurology (SyNergy), Munich, Germany. [28]Department of Anatomy and Histology, University of Veterinary Medicine, Budapest, Hungary. [29]Department of Chemistry, Indiana University, Bloomington, IN, USA. [30]Helmholtz Munich, Munich, Germany. [31]Ludwig-Maximilians-University (LMU) Munich, Munich, Germany. [32]Metabolic Cell Architecture, Department of Molecular Life Sciences, TUM School of Life Sciences, Technical University of Munich, Munich, Germany. [33]Walther-Straub Institute of Pharmacology and Toxicology, Ludwig-Maximilians-University (LMU) Munich, Munich, Germany. [34]These authors contributed equally: Daniela Liskiewicz, Aaron Novikoff. ✉e-mail: timodirk.mueller@helmholtz-munich.de

## Methods

### Animals and housing conditions

Experiments were performed in accordance with the Animal Protection Law of the European Union after permission by the Governments of Upper Bavaria, Germany, or Copenhagen, Denmark, or by the Institutional Animal Care and Use Committees of the Universities of Texas Southwestern, Michigan, Duke or Yale, USA. Mice were double- or single-housed and unless otherwise indicated fed ad libitum with either a regular chow (1314, Altromin or 5L0D, LabDiet) or HFD (58% fat, D12331, Research Diets) under constant ambient conditions of $22 \pm 2\,°C$ with constant humidity (45–65%) and a 12 h:12 h light:dark cycle. Leptin receptor-deficient db/db mice were purchased from the Jacksons Laboratory (Strain 000697). Doxycyclin-inducible GIPR-overexpressing mice (TRE-GIPR mice) were generated in-house at The University of Texas Southwestern Medical Center as described previously[41]. C57BL/6J DIR-knockout mice were generated in-house at Helmholtz Munich (Supplementary Information 1).

### Pharmacological studies

Indirect calorimetry and assessment of body composition was performed as described in Supplementary information 1. Drug effects were assessed in age-matched male single-, or double-housed C57BL6/J mice that were randomly assigned in groups matched for genotype, body weight and body composition (fat and lean tissue mass). Mice were treated subcutaneously at the indicated doses with 5 µl per g body weight of either Vehicle, Lani (CAS: 927961-18-0, MedChemExpress), semaglutide, GLP-1–Lani, GLP-1–Tesa, GLP-1–GIP, GLP-1–GIP–Tesa, GLP-1–GIP–Lani, or co-administration of either GLP-1–GIP plus Lani or GLP-1–Lani plus acyl-GIP (Extended Data Fig. 1a–f). All peptides were provided by the Novo Nordisk Research Center Indianapolis, IN, USA or the Indiana Biosciences Research Institute, IN, USA (for drug development see Supplementary Information 1). Assessment of drug effects on the cardiovascular system and on POMC neuronal activity was performed as described in Supplementary Information 1.

### Glucose and lipid metabolism

Glucose and insulin tolerance was assessed in 6 h-fasted mice injected intraperitoneally with either 1.5–2 g kg$^{-1}$ of glucose or 0.6–0.75 U kg$^{-1}$ of insulin (Humalog; Eli Lilly). Glucose-induced insulin secretion was assessed in 6 h-fasted mice orally gavaged with 4 g kg$^{-1}$ glucose. Pyruvate tolerance was assessed in 12 h-fasted mice injected intraperitoneally with 1.25 g kg$^{-1}$ sodium pyruvate (11360070, Thermo Fisher). Commercially available ELISAs were used according to the manufacturer's instruction to measure insulin (90080, Crystal Chem), triglycerides (94501, Fujifilm), cholesterol (293-93601, Fujifilm) and free fatty acids (434-91795, 436-91995, 270-77000, Fujifilm). Hyperinsulinaemic-euglycaemic clamps and assessment of tissue-selective glucose uptake were performed as described in Supplementary Information 1.

### Gene expression analysis

Total RNA was isolated using the RNeasy Kit (QIAGEN) according to the manufacturer's instructions. cDNA synthesis was performed using the QuantiTect Reverse Transcription Kit (QIAGEN) or High-Capacity cDNA Reverse Transcription Kit (Thermo Fisher Scientific) according to manufacturer's instructions. Gene expression was profiled using SYBR green (Thermo Fisher Scientific) and the Quantstudio 7 flex cycler (Applied Biosystems). The relative expression levels of each gene were normalized to the housekeeping gene *HPRT*. Primer sequences are listed in Supplementary Information 1.

### Cell culture studies

HEK293T cells (CRL-3216, ATCC) were cultured in DMEM (11995073, Life Technologies) supplemented with 10% heat-inactivated FBS (10500064, Life Technologies), 100 IU ml$^{-1}$ penicillin and 100 µg ml$^{-1}$ streptomycin solution (Pen/Strep, P4333, Sigma Aldrich). Cells (700,000 per well) were seeded in 6-well plates and incubated to 70% confluency in DMEM (10% FBS, 1% Pen/Strep). Twenty-four hours following seeding, transient transfections were performed using Lipofectamine 2000 (11668019, Invitrogen) according to the manufacturer's instructions without including additional transformation carrier DNA. BRET assays and in vitro quantification of PPAR-responsive genes were performed as described in Supplementary Information 1.

### Proteomics, transcriptomics and histology

Proteomics, bulk RNA sequencing and immunofluorescence was performed as described in Supplementary Information 1. For histological analysis, excised samples were fixed in 4% (w/v) neutral buffered formalin, embedded in paraffin and cut into 3 µm slices for haematoxylin and eosin (H&E) staining or immunohistochemistry. Immunohistochemical detection of alpha and beta cells was performed using rabbit anti-insulin (3014, 1:800, Cell Signaling Technology) and mouse anti glucagon (G2654, 1:1,000, Merck) as primary antibodies and goat anti-rabbit AF750 (A21039, 1:100, Invitrogen) and donkey anti-mouse AF555 (A32773, 1:200, Invitrogen) as secondary antibodies. Nuclei were labelled with Hoechst33342 (H1399, Thermo Fischer). The stained tissue sections were scanned with an AxioScan 7 digital slide scanner (Zeiss) equipped with a 20× objective. Steatosis was graded semiquantitatively by the presence of fat vacuoles in liver cells according to the percentage of affected tissue (0, <5%; 1, 5–33%; 2, 33–66%; 3, >66%) and lobular inflammation was scored by overall assessment of inflammatory foci per 200× field (0, no foci; 1, <2 foci; 2, 2–4 foci; 3, >4 foci)[56]. Automated digital image analysis (Visiopharm) was used for determination of alpha and beta cell mass and mean islet size.

### Glucose-stimulated insulin secretion

Mice were euthanized by cervical dislocation, followed immediately by clamping of the bile duct and perfusion with collagenase P (11249002001, Roche Diagnostics). Tissues were incubated in a 15 ml Falcon tube with 1 ml of collagenase P solution for 15 min at 37 °C, followed by addition of 12 ml of the cold G-solution (Sigma Aldrich) and centrifugation at 1,620 rpm at room temperature. The pellet was subsequently washed with 10 ml of the G-solution, which comprised of 500 ml HBSS (BE10-508F, Life Technologies) with 10% BSA (126615, Sigma Aldrich) and 1% Pen-Strep (15140122, Life Technologies), and re-suspended in 5.5 ml of gradient solution (15% Optiprep (5 ml 10% RPMI, Life Technologies) + 3 ml of 40% Optiprep, which was diluted from 60% Optiprep with G-solution (D1556, Sigma Aldrich)) per sample, and placed on top of 2.5 ml of the gradient solution. To form a 3-layer gradient, 6 ml of the G-solution was added on the top. Samples were then incubated for 10 min at room temperature and centrifuged at 1,700 rpm. The interphase was then collected and filtered through a 70-µm nylon filter (352350, BD Falcon), before washing with G-solution. Islets were handpicked by a micropipette under the microscope and cultured in RPMI 1640 medium (11875093, Life Technologies) overnight.

For assessment of glucose-induced insulin secretion, culture medium was removed and islet microtissues were equilibrated for 1 h with Krebs Ringer Hepes Buffer (KRHB; 131 mM NaCl, 4.8 mM KCl, 1.3 mM CaCl$_2$, 25 mM HEPES, 1.2 mM KH$_2$PO$_4$, 1.2 mM MgSO$_4$, 2% BSA) containing 2.8 mM glucose. The supernatant was collected as a sample under low glucose condition for 45 min incubation, and islets were incubated for another 45 min at 37 °C with KRHB containing 16.7 mM glucose and supplements as above. The supernatant was collected as a sample under high glucose condition and stored at −20 °C. For drug-induced insulin secretion, GLP-1–GIP and GLP-1–GIP–Lani were diluted in 1× KRHB buffer with 20 mM glucose to reach a concentration of 1, 10 or 50 nM. Cells were subsequently treated with either compound for 45 min. Insulin concentrations were determined using a Mouse Insulin ELISA (90082, Crystal Chem).

## Conditioned taste avoidance

CTA experiments used either wild-type C57BL/6J mice purchased from the Jackson Laboratories or a glutamatergic neuron GLP-1R knockout mouse (a cross between a *Glp1r-flox* mouse and a *Vglut2-ires-cre* mouse, 035238 and 016963; Jackson Laboratory). Mice were acclimatized to the experimental conditions, then automated water systems were removed and replaced with two bottles full of water for three days. During this time, mice were handled and injected daily with 0.1 ml saline. On day 4, water bottles were removed for 22 h to induce thirst. On day 5, mice were given a bottle containing 0.15% saccharin in water for 2 h. Saccharin bottles were weighed before and after the 2 h period to ensure that all mice consumed the tastant. Then, mice were injected with either vehicle or the drugs as indicated, and the saccharin bottle was replaced by a water bottle. On day 8, water bottles were again removed for 22 h to induce thirst. On day 9, mice were given both a water bottle and a saccharin bottle, each of which were weighed at 0 and 24 h. The preference ratio was calculated as: saccharin intake/(saccharin + water intake).

## Replicates, randomization and blinding

In vivo studies were performed in male mice that were randomly distributed in groups matched for genotype, age, body weight and body composition (fat and lean tissue mass). The number of independent biological samples per group is indicated in the figure legends and the Source Data files. For in vivo studies, drugs were aliquoted by a lead scientist in number-coded vials and most, but not all, handling investigators were blinded to the treatment condition. Tolerance tests (glucose, insulin, pyruvate and in vivo glucose-stimulated insulin secretion) were performed by experienced research assistants who were blinded to the treatment conditions.

## Statistics and reproducibility

For animal studies, sample sizes were calculated based on a power analysis assuming that a body weight difference of ≥5 g between the treatment groups can be captured with a power of ≥75% when using a two-sided, two-tailed statistical test under the assumption of a s.d. of 3.5 and an alpha level of 0.05. Statistical analyses were performed using the statistical tools implemented in GraphPad Prism10 (v10.0.3), and after testing of data for normal distribution using the Kolmogorov–Smirnov test, D'Agostino and Pearson test, Anderson–Darling test or Shapiro–Wilk test implemented in GraphPad Prism (v10.0.3). Statistical tests and individual *P* values are presented in Supplementary Table 1 or Source Data. All results are given as mean ± s.e.m. *P* < 0.05 was considered statistically significant. Differences in energy expenditure and heart weight were calculated using ANCOVA with body weight or tibia lengths as covariate using SPSS (v31). No animals or data were excluded from the analysis unless for animal welfare reasons (for example, injury due to fighting) or identification of singular outliers using Grubbs test. Outliers are shown in the Source Data. The metabolic effects of GLP-1–GIP–Lani have been reproduced in several in vivo studies in the manuscript, and across several independent laboratories. In vitro studies in Fig. 1f,g and Extended Data Fig. 2a,b represent 3–6 independent biological replicates, each obtained in an independent study and calculated based on the average of 2–6 technical replicates. In vitro studies in Fig. 1i–k represent 4–6 independent biological replicates, each obtained in an independent study and calculated based on the average of 2 technical replicates. Histological data in Fig. 3l are representative examples out of *n* = 8 mice per group. Microscopic images and FOS quantification in Fig. 5a–f are representative examples from 3–4 mice per group. Electrophysiological recordings in Fig. 5p are representative examples from 6–7 mice per group. The original pictures from which representative examples are depicted are displayed in Supplementary Fig. 1. All in vivo data represent independent biological samples as indicated in the figure legends.

## Reporting summary

Further information on research design is available in the Nature Portfolio Reporting Summary linked to this article.

## Data availability

Raw data for the proteomic and transcriptomic analysis are available via PRIDE (PXD062990). Raw data from the bulk RNA sequencing are available in the Gene Expression Omnibus (GEO) under SuperSeries accession number GSE314029. All data used for the statistical analysis are available in the Source Data, along with the GraphPad Prism-derived report on the statistical analysis. The statistical report contains the mean difference between the treatment groups, the 95% confidence intervals, the significance summary, and the exact *P* values (unless *P* < 0.0001). Transcripts were aligned using the GRCm38 reference genome (GenBank accession GCA_000001635.20). Source data are provided with this paper.

56. Finan, B. et al. Chemical hybridization of glucagon and thyroid hormone optimizes therapeutic impact for metabolic disease. *Cell* **167**, 843–857.e814 (2016).

**Acknowledgements** This work was funded by the European Union within the scope of the European Research Council ERC-CoG Trusted 101044445, awarded to T.D.M. P.T.P. received funding from the ERC-CoG Yoyo-LepReSens 101002247. Views and opinions expressed are those of the author(s) only and do not necessarily reflect those of the European Union or the European Research Council. Neither the European Union nor the awarding authority can be held responsible for them. T.D.M. further received funding from the German Research Foundation (DFG TRR296, TRR152, SFB1123 and GRK 2816/1) and the German Center for Diabetes Research (DZD e.V.). R.J.S. received funding for this work from NIH grant R01DK133140. N.K. is supported by DFG Emmy Noether KR5166/2 and the European Foundation for the Study of Diabetes, Future Leader Award NNF20SA0066171. We thank M. Geiger, X. Leonhardt, P. Dörfelt, W. Lu, M. Kilian, I. Karavaeva, D. Tandio, S. Ribicic, D. Brandt and C. K. Kristensen for technical assistance.

**Author contributions** D.L., A.N., A.L., A.K., S.A., J.E.C., P.C., R.L.C., C.C., C.G.-C., X.-B.G., M.C., H.E., G.G., R.M.G., D.T.H., M.J., E.K., P.K., A.C.K., M.K., C.M.K., D.C.L., X.L., G.M.-K., S.M.M., R.N., M.P., D.P.-T., P.S.S.P., S.P., S.S., J.W. and J.Y. designed and performed experiments and analysed and interpreted data. A.C.-S. developed animal models. A.F., F.B., P.T.P. and M.T. analysed and interpreted data. W.S.D., B.F., J.D.D., S.A.M. and P.J.K. participated in drug development and interpretation of data. C.W., T.L.H., Z.G.-H., P.E.S., R.J.S., R.D.D., M.H.T. and N.K. participated in study design, analysis of data and interpretation of results. T.D.M. conceptualized the project, supervised experiments and analysed and interpreted data. T.D.M. wrote the manuscript with support of D.L and A.N.

**Funding** Open access funding provided by Helmholtz Zentrum München - Deutsches Forschungszentrum für Gesundheit und Umwelt (GmbH).

**Competing interests** M.H.T. is a member of the scientific advisory board of ERX Pharmaceuticals, was a member of the Research Cluster Advisory Panel (ReCAP) of the Novo Nordisk Foundation between 2017 and 2019, attended a scientific advisory board meeting of the Novo Nordisk Foundation Center for Basic Metabolic Research, University of Copenhagen, in 2016, received funding for research projects from Novo Nordisk (2016–2020) and Sanofi-Aventis (2012–2019), and was a consultant for Bionorica SE (2013–2017), Menarini Ricerche S.p.A. (2016) and Bayer Pharma AG Berlin (2016). As former director of the Helmholtz Diabetes Center and the Institute for Diabetes and Obesity at Helmholtz Zentrum München (2011–2018), and since 2018, as CEO of Helmholtz Zentrum München, M.H.T. has been responsible for collaborations with a multitude of companies and institutions worldwide. In this capacity, M.H.T. discussed potential projects with and has signed/signs contracts for his institute(s) and for the staff for research funding and/or collaborations with industry and academia worldwide, including but not limited to pharmaceutical corporations such as Boehringer Ingelheim, Eli Lilly, Novo Nordisk, Medigene, Arbormed, BioSyngen and others. In this role, M.H.T. was/is further responsible for commercial technology transfer activities of the institute(s), including diabetes related patent portfolios of Helmholtz Zentrum München such as WO/2016/188932 A2 or WO/2017/194499 A1. M.H.T. confirms that to the best of his knowledge none of the above funding sources were involved in the preparation of this paper. T.D.M. holds stock from Eli Lilly and receives research funding from Novo Nordisk. T.D.M. has further received speaking fees from Novo Nordisk, Eli Lilly, Boehringer Ingelheim, Merck, AstraZeneca and Mercodia. J.E.C. received funding from Novo Nordisk, Eli Lilly, Merck, Structure Therapeutics and Prostasis. J.E.C. has served as an advisor/consultant in the past 12 months to Arrowhead Therapeutics, Boehringer Ingelheim, Neurocrine Biosciences, Prostasis, Protagonist and Structure Therapeutics. R.D.D. is a co-inventor on intellectual property owned by Indiana University and licensed to Novo Nordisk. R.D.D., B.F., J.D.D., S.A.M. and P.J.K. were previously employed by Novo Nordisk. J.D.D., S.A.M. and P.J.K. receive research funding from Eli Lilly and are co-founders and shareholders in Volari Therapeutics,

both unrelated to this work. P.T.P. received speaker honoraria from Novo Nordisk. B.F. is a current employee of Eli Lilly. R.J.S. has received research support from Novo Nordisk, Fractyl, AstraZeneca, Congruence Therapeutics, Eli Lilly, Bullfrog AI, Glyscend Therapeutics and Amgen. R.J.S. has served as a paid consultant for Novo Nordisk, Eli Lilly, CinRx, Fractyl, Structure Therapeutics, Crinetics, Amgen, Congruence Therapeutics and Nuanced Health. R.J.S. has equity in Nuanced Health, Coro Bio. Fractyl and Rewind. The other authors declare no competing interests.

**Additional information**
**Correspondence and requests for materials** should be addressed to Timo D. Müller.

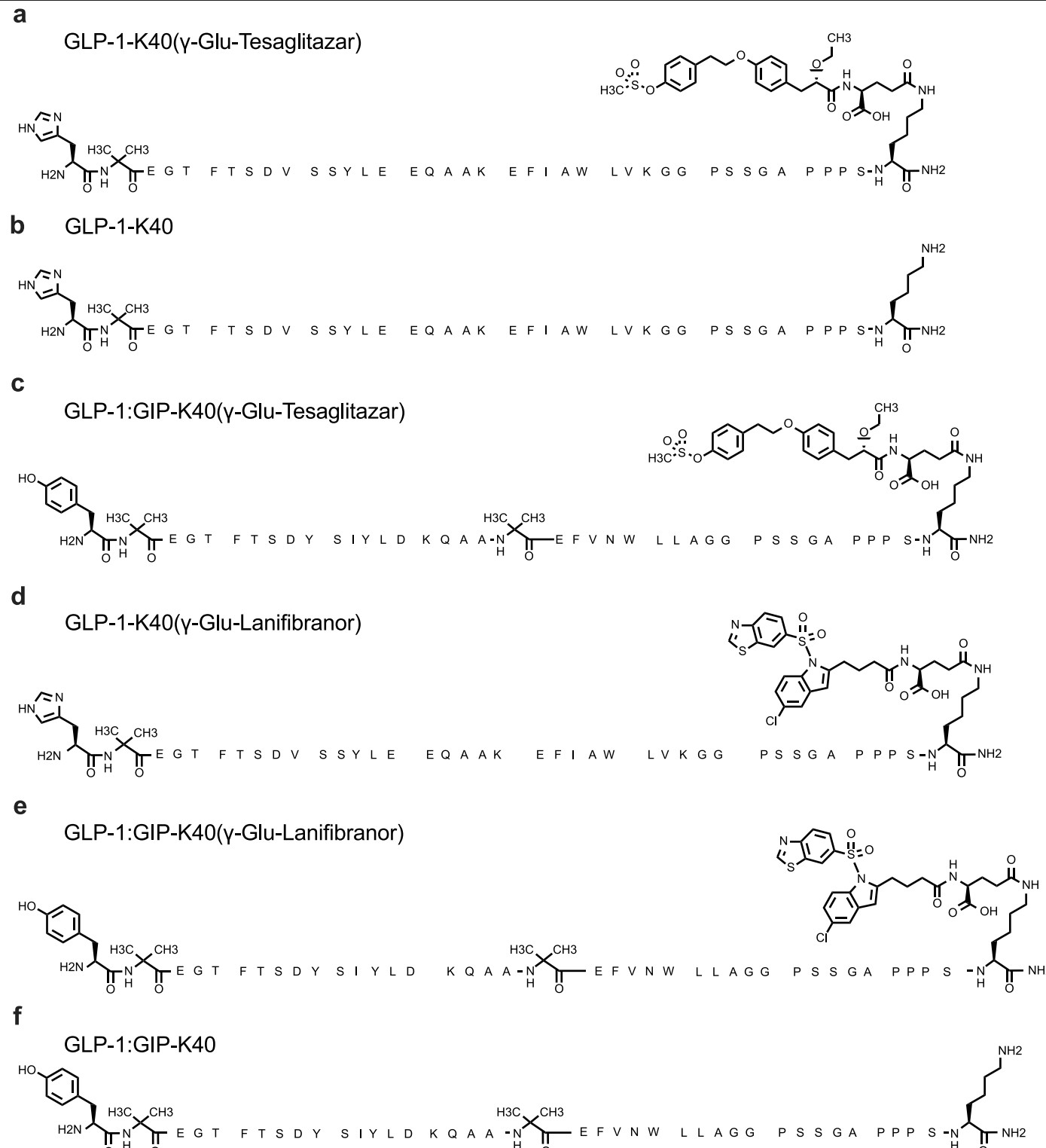

**Extended Data Fig. 1 | Structure of the used peptides. a–f**, Chemical structure of the used peptides. **a**, GLP-1:Tesa. **b**, the corresponding GLP-1 backbone. **c**, GLP-1: GIP:Tesa. **d**, GLP-1:Lani. **e**, GLP-1:GIP:Lani. **f**, and the corresponding GLP-1:GIP backbone.

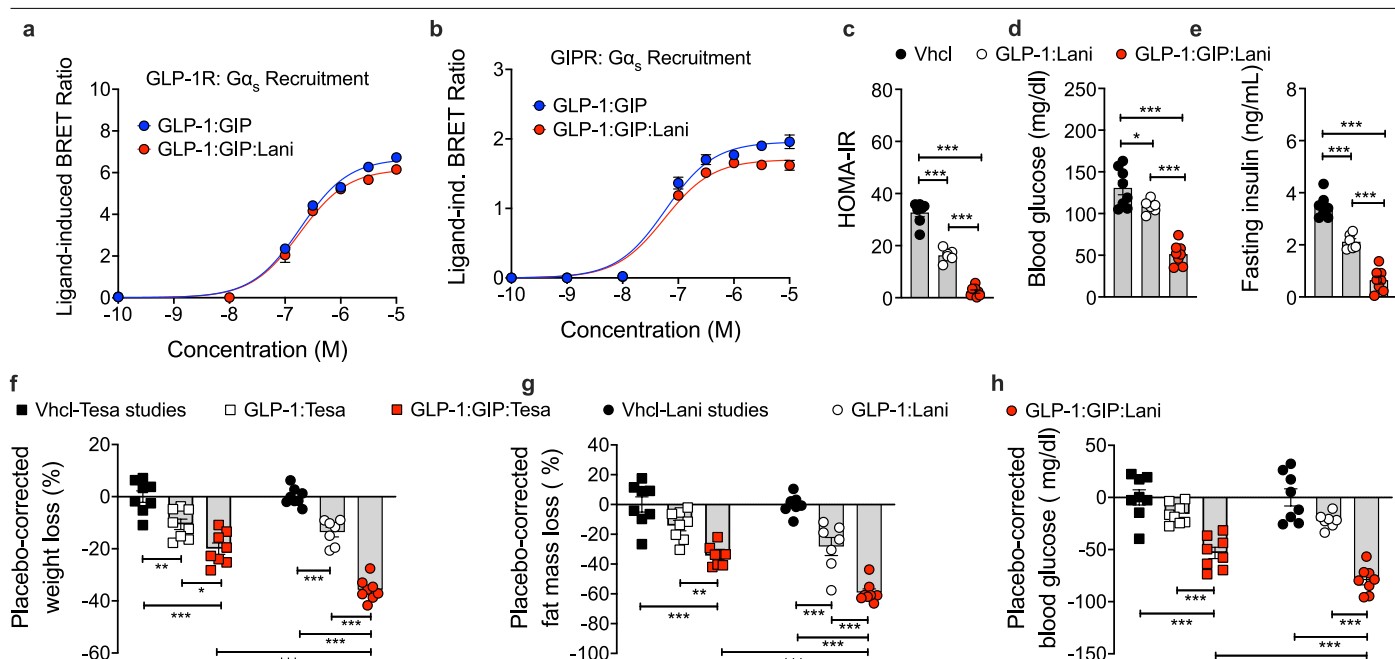

**Extended Data Fig. 2 | Comparison of GLP-1GIP:Tesa vs. GLP-1:GIP:Lani.**
**a**,**b**, Dose-response curves of ligand-induced BRET changes between Nluc-tagged G$\alpha_s$ recruitment to GFP-tagged GLP-1R or GIPR. **a**, HEK293T cells expressing GLP-1R (n = 3 independent biological replicates per group). **b**, HEK293T cells expressing GIPR (n = 6 independent biological replicates per group). **c**–**e**, DIO mice were treated once-daily via s.c. injections of GLP-1:Lani, GLP-1:GIP:Lani or vehicle for 14 days. 10 nmol kg⁻¹. **c**, Homeostatic model assessment of insulin resistance (HOMA-IR) (n = 7 Vhcl, n = 6 GLP-1:Lani, n = 8 GLP-1:GIP:Lani). **d**, Fasting blood glucose (n = 8 Vhcl, n = 7 GLP-1:Lani, n = 8 GLP-1:

GIP:Lani). **e**, Fasting plasma insulin (Vhcl, n = 7; GLP-1:Lani, n = 6; GLP-1:GIP:Lani, n = 8). **f**–**h**, DIO mice were treated once-daily via s.c. injections of GLP-1:Tesa, GLP-1:GIP:Tesa, GLP-1:Lani, GLP-1:GIP:Lani or vehicle for 14 days. GLP-1:Lani, n = 7; all other groups, n = 8. 10 nmol kg⁻¹. **f**, placebo-corrected effects on body weight. **g**, placebo-corrected effects on fat mass. **h**, placebo-corrected effects on fasting blood glucose. Data were analysed using one-way ANOVA with Bonferroni post hoc multiple-comparison test (**c**–**h**). Data are mean ± s.e.m. *P < 0.05, **P < 0.01, ***P < 0.001. Exact P-values, n-values and detailed statistics are provided in Supplementary Tables 1 and 3 and the Data Source File.

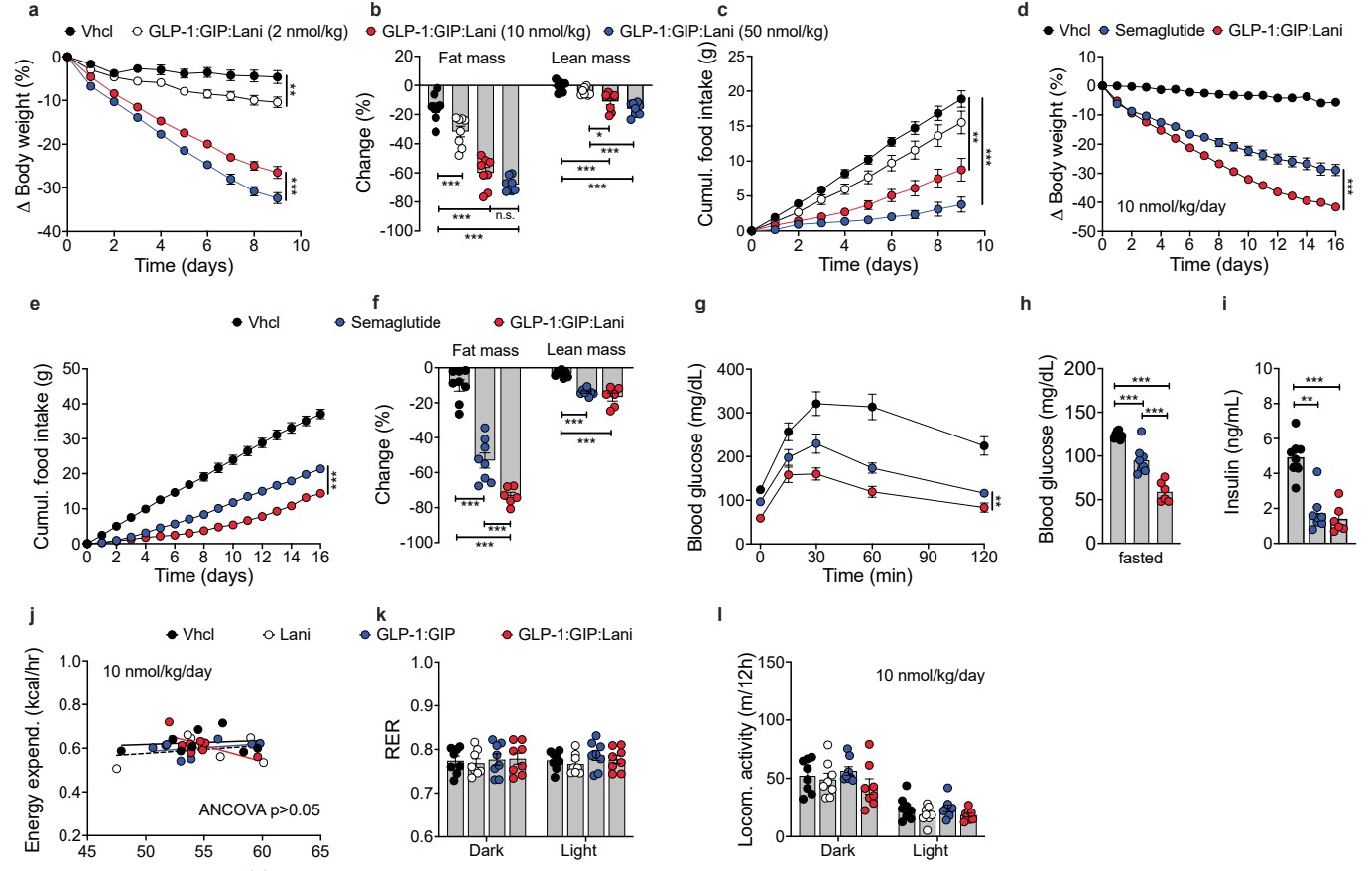

**Extended Data Fig. 3 | Comparison to semaglutide. a–c**, DIO mice were treated once-daily via s.c. injections of GLP-1:GIP:Lani or vehicle for 9 days. 2, 10 and 50 nmol kg⁻¹. **a**, Body weight (n = 8). **b**, Change in fat and lean mass (n = 8). **c**, Cumulative food intake (GLP-1:GIP:Lani 10 nmol/kg and 50 nmol/kg, n = 6; all other groups, n = 8). **d–i**, DIO mice were treated once-daily via s.c. injections of GLP-1:GIP:Lani, semaglutide or vehicle for 16 days. 10 nmol kg⁻¹. **d**, Body weight (GLP-1:GIP:Lani, n = 6; all other groups, n = 8). **e**, Cumulative food intake (GLP-1:GIP:Lani, n = 6; all other groups, n = 8). **f**, Change in fat and lean mass (GLP-1:GIP:Lani, n = 6; all other groups, n = 8). **g**, Glucose tolerance test at day 16 (Vhcl, n = 8; Semaglutide, n = 7; GLP-1:GIP:Lani, n = 6). **h**, Fasting blood glucose (GLP-1:GIP:Lani, n = 6; all other groups, n = 8). **i**, Fasting plasma

insulin (GLP-1:GIP:Lani, n = 6; all other groups, n = 8). **j–l**, DIO mice were treated once-daily via s.c. injections of Lani, GLP-1:GIP, GLP-1:GIP:Lani or vehicle for 14 days. n = 8 each group. 10 nmol kg⁻¹. **j**, Energy expenditure. **k**, Respiratory exchange ratio (RER). **l**, locomotor activity. Data were analysed using one-way ANOVA with Bonferroni post hoc multiple-comparison test (**b**,**f**,**h**) two-way repeated-measures ANOVA (**a**,**c**,**d**,**e**,**g**), Kruskall-Wallis test with uncorrected Dunns's test (**i**), or Analysis of Covariance (ANCOVA) with body weight as covariate (**j**). Cumulative food intake (**c**,**e**) was assessed per cage in single-or double-house mice. Data are mean ± s.e.m. *P < 0.05, **P < 0.01, ***P < 0.001. Exact P-values, n-values and detailed statistics are provided in Supplementary Tables 1 and 3 and the Data Source File.

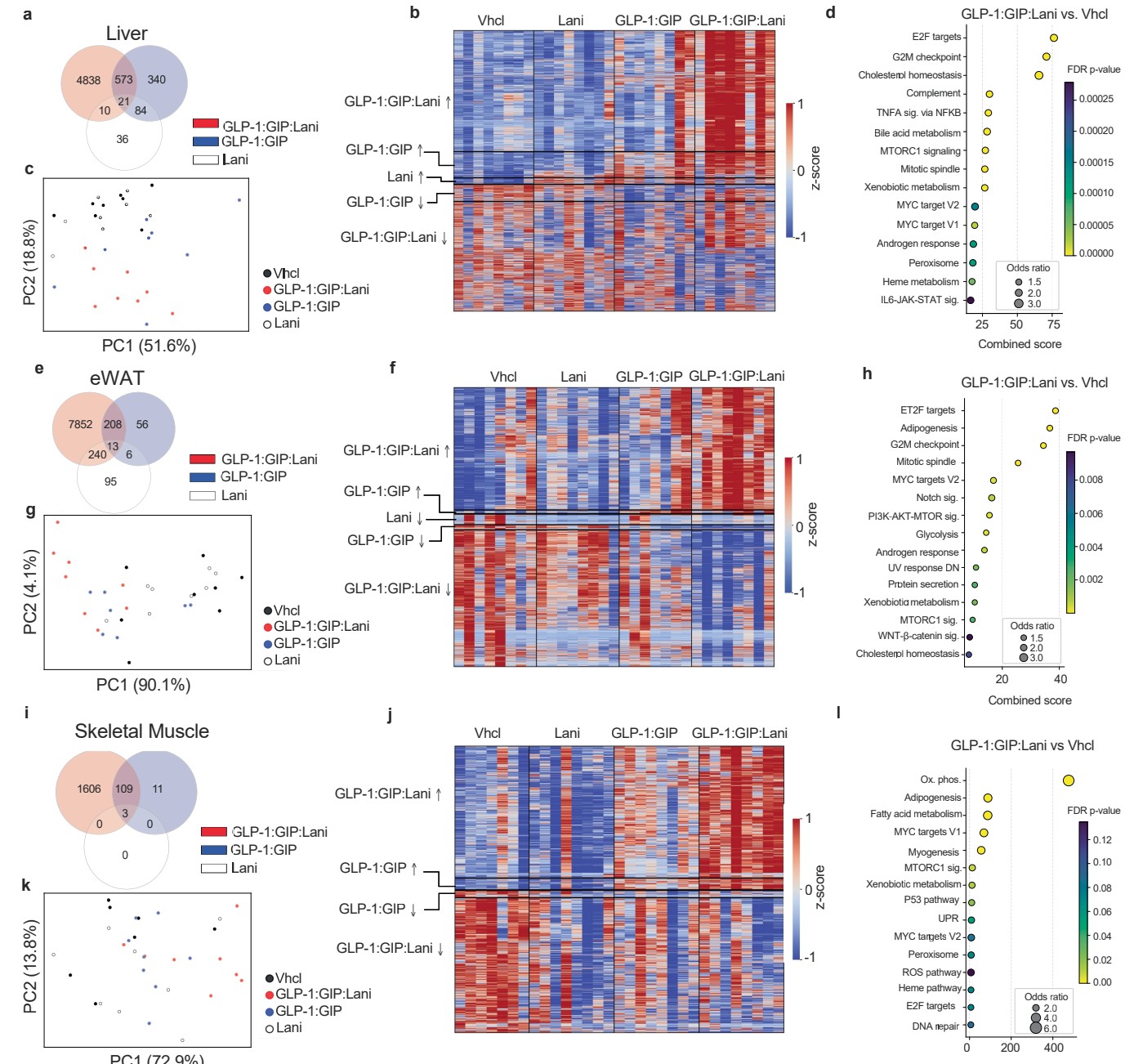

**Extended Data Fig. 4 | Transcriptional drug effects on the liver, eWAT and skeletal muscle. a–l**, DIO mice were treated once-daily via s.c. injections of GLP-1:GIP, GLP-1:GIP:Lani, Lani or vehicle for 12 days. 10 nmol kg⁻¹. **a,e,i**, Venn-diagram with number of differentially expressed genes (DEGs; padj <0.05) in the liver (**a**), eWAT (**e**) and skeletal muscle (**i**). **c,g,k**, Principal component analysis (PCA) showing the first two principal components of overall gene expression in liver (**c**), eWAT (**g**) and skeletal muscle (**k**). **b,f,j**, Heatmap showing gene expression z-scores of all significantly up- or down-regulated genes for each treatment relative to vehicle in the liver (**b**), eWAT (**f**) and skeletal muscle (**j**). **d,h,l**, Gene expression overrepresentation analysis (ORA) of hallmark pathways in GLP-1:GIP:Lani–treated mice vs. vehicle, showing odds ratio, false discovery rate (FDR)–adjusted p-value, and enrichment score for each pathway in the liver (**d**), eWAT (**h**) and skeletal muscle (**l**). Data were analysed using two-sided Wald test with Benjamin-Hochburg FDR adjustment (**a,b,e,f,h,i,j**) or one-sided Fischer's exact test with Benjamin-Hochburg FDR adjustment (**d,l**). Data in panel **a**–**h** comprise n = 8 mice each group; data in panel **j**–**k** comprise n = 7 mice for Vhcl and n = 8 for all other groups. Data are mean ± s.e.m. *P < 0.05, **P < 0.01, ***P < 0.001. Exact P-values, n-values and detailed statistics are provided in Supplementary Tables 1 and 3 and the Data Source File.

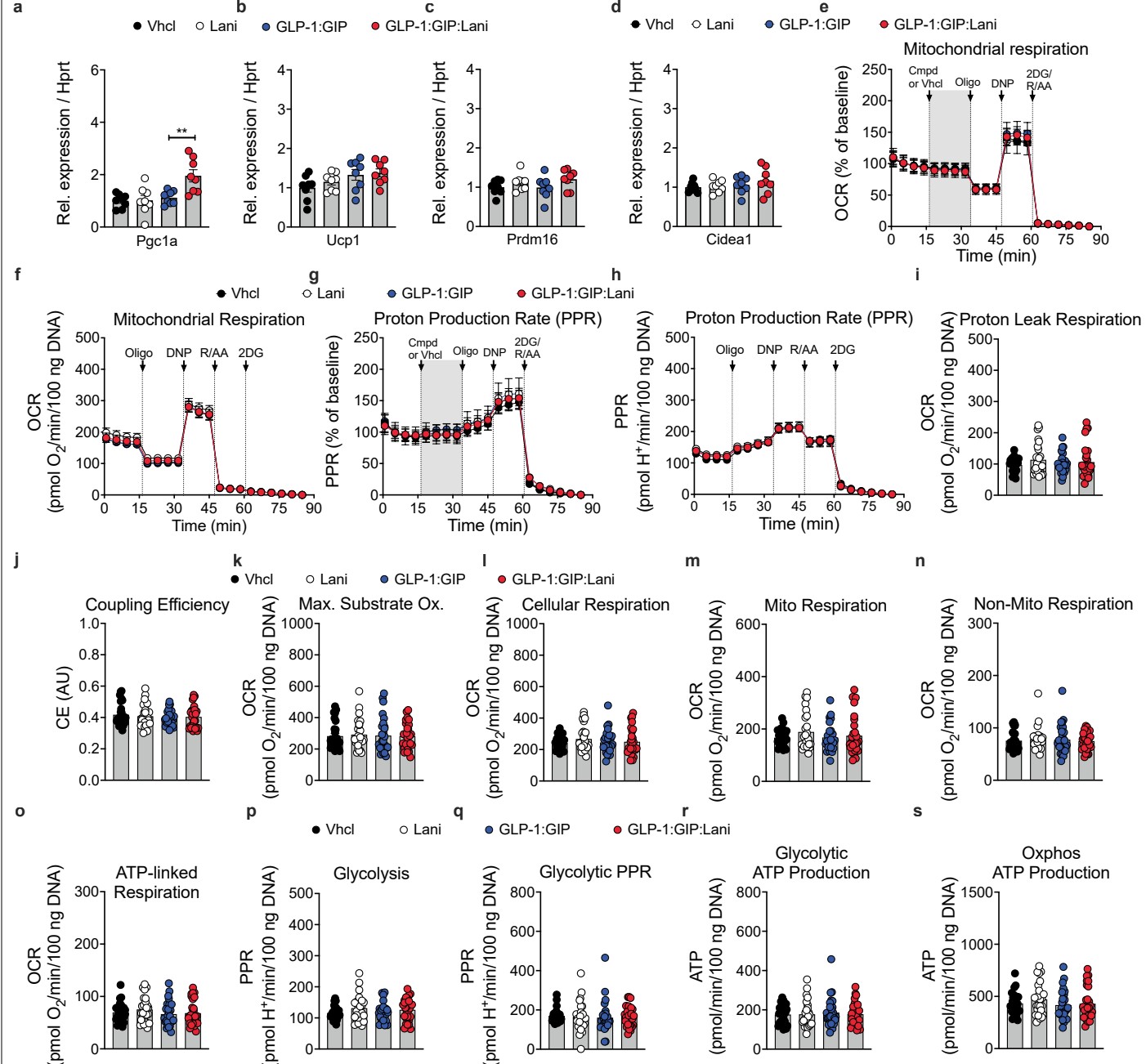

**Extended Data Fig. 5 | Drug effects on brown adipose tissue. a–d,** Relative expression of thermogenic genes in BAT of DIO mice treated once-daily via s.c. injections of GLP-1:GIP, GLP-1:GIP:Lani or vehicle for 12 days. n = 8 each group. 10 nmol kg⁻¹. **a,** Peroxisome proliferator-activated receptor gamma coactivator-1α (Pgc1α). **b,** Uncoupling protein 1 (Ucp1). **c,** PR domain containing 16 (Prdm16). **d,** Cell death-inducing DFFA-like effector a (Cidea1). **e–s,** Drug effects on mitochondrial function in differentiated immortalized BAT primary cells. 100 nM. **e,** Acute effects on mitochondrial respiration. **f,** Chronic effects on mitochondrial respiration. **g,** Acute effects on proton production rate. **h,** Chronic effects on proton production rate. **i,** proton leak respiration. **j,** Coupling efficiency. **k,** Maximal substrate oxidation. **l,** cellular respiration.

**m,** Mitochondrial respiration. **n,** Non-mitochondrial respiration. **o,** ATP-linked respiration. **p,** Glycolysis. **q,** Glycolytic proton production rate. **r,** Glycolytic ATP production. **s,** ATP production from oxidative phosphorylation. Data were analysed using one-way ANOVA with Bonferroni post hoc multiple-comparison test (**a–d,j,o,p,s**), two-way repeated-measures ANOVA (**e,f,g,h**), or Kruskall-Wallis test with uncorrected Dunns's test (**i,k–n,q,r**). Data in panel **e,g** comprise n = 11 independently differentiated wells per group; data in panel **f,h–s** comprise n = 26 independently differentiated wells for Lani and n = 27 independently differentiated wells for all other groups. Data are mean ± s.e.m. *P < 0.05, **P < 0.01, ***P < 0.001. Exact P-values, n-values and detailed statistics are provided in Supplementary Tables 1 and 3 and the Data Source File.

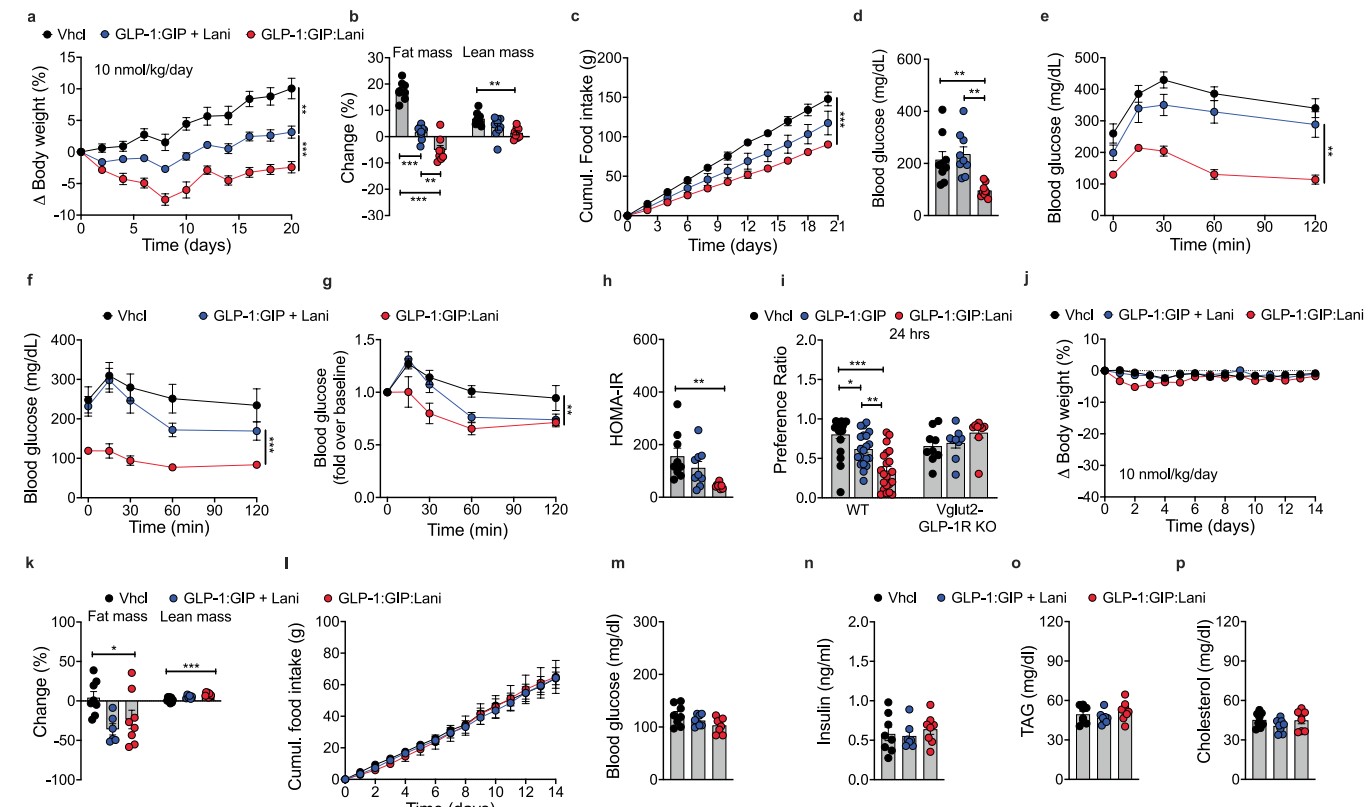

**Extended Data Fig. 6 | Drug effects in db/db mice and in lean WT and Vglut2-GLP-1R KO mice. a–h,** Leptin receptor deficient (db/db) mice were treated once-daily via s.c. injections of GLP-1:GIP + Lani (n = 9), GLP-1:GIP:Lani (n = 8) or vehicle (n = 9) for 20 days. 10 nmol kg⁻¹. **a,** Body weight. **b,** Change in fat and lean mass. **c,** Cumulative food intake. **d,** Fasting blood glucose. **e,** Glucose tolerance test. **f,** Insulin tolerance test. **g,** Baseline-corrected insulin tolerance test. **h,** Homeostatic model assessment of insulin resistance (HOMA-IR). **i,** Conditioned Taste Avoidance (CTA) in lean wildtype and Vglut2-GLP-1R KO mice 24 h after single s.c. administration of GLP-1:GIP (WT, n = 20; KO, n = 9), GLP-1:GIP:Lani (WT, n = 21; KO, n = 9) or vehicle (WT, n = 20; KO, n = 9). 10 nmol kg⁻¹. **j–p,** Lean wildtype mice were treated once-daily via s.c. injections

of GLP-1:GIP + Lani, GLP-1:GIP:Lani or vehicle for 14 days. 10 nmol kg⁻¹. **j,** Body weight (n = 8 each group). **k,** Change in fat and lean mass (Vhcl, n = 8; GLP-1:GIP: Lani, n = 8; GLP-1:GIP + Lani, n = 6). **l,** Cumulative food intake (n = 8 each group). **m,** Fasting blood glucose (n = 8 each group), **n,** Fasting plasma insulin (Vhcl, n = 8; GLP-1:GIP:Lani, n = 8; GLP-1:GIP + Lani, n = 7). **o,** Plasma triglycerides (n = 8 each group). **p,** Plasma cholesterol (n = 8 each group). Data were analysed using one-way ANOVA with Bonferroni post hoc multiple-comparison test (**b,d,h,k,m,o,p**), two-way repeated-measures ANOVA (**a,c,e–g,j,l**) or Kruskall-Wallis test with uncorrected Dunns's test (**i,n**). Data are mean ± s.e.m. *P < 0.05, **P < 0.01, ***P < 0.001. Exact P-values, n-values and detailed statistics are provided in Supplementary Tables 1 and 3 and the Data Source File.

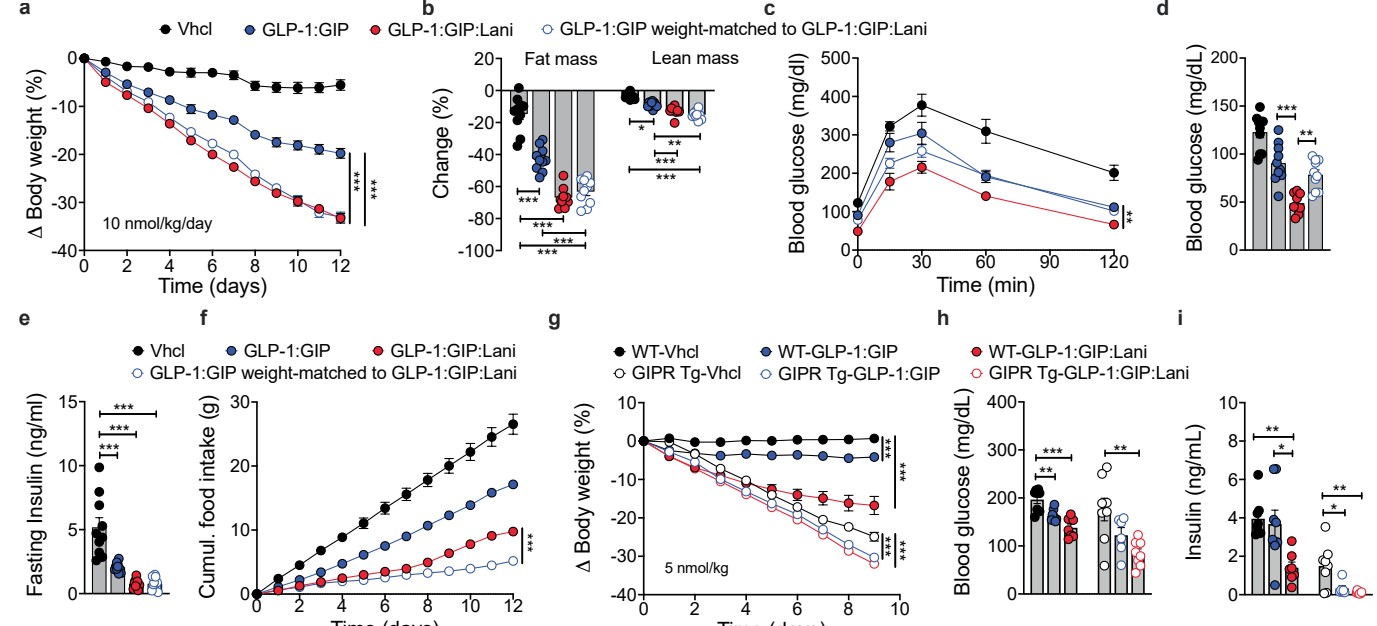

**Extended Data Fig. 7 | Drug effects in weight-matched DIO and adipose GIPR transgenic mice. a–f**, DIO mice were treated once-daily via s.c. injections of GLP-1:GIP (n = 10), GLP-1:GIP:Lani (n = 9) or vehicle (n = 10) for 12 days. An additional group was daily treated with GLP-1:GIP and additionally food restricted to match the body weight of mice treated with GLP-1:GIP:Lani (n = 10). 10 nmol kg⁻¹. **a**, Body weight. **b**, Change in fat and lean mass. **c**, Glucose tolerance test at day 12. **d**, Fasting blood glucose. **e**, Fasting plasma insulin. **f**, Cumulative food intake. **g–i**, DIO wildtype (WT) and GIPR transgenic (Tg) mice were treated daily with GLP-1:GIP, GLP-1:GIP:Lani or vehicle. 5 nmol kg⁻¹. **g**, Body weight (WT-GLP-1:GIP:Lani, n = 7; all other groups, n = 8). **h**, Fasting blood glucose. **i**, Fasting plasma insulin. Data in panel **h,i** comprise n = 8 for WT-Vhcl, WT-GLP-1:GIP, GIPRtg-Vhcl and GIPRtg-GLP-1:GIP:Lani, n = 7 for

WT-GLP-1:GIP:Lani and n = 6 for GIPRtg-GLP-1:GIP. Data were analysed using one-way ANOVA with Bonferroni post hoc multiple-comparison test (**d**,**e**,**h**), two-way repeated-measures ANOVA (**a**,**c**,**f**,**g**), or Kruskall-Wallis test with uncorrected Dunns's test (**i**). Data in **b** were analyzed using one-way analysis of variance (ANOVA) with Bonferroni post hoc multiple-comparison test in case of normal distribution (fat mass) or using the Kruskall-Wallis test with uncorrected Dunns's test in case of not normal distribution (lean mass). Cumulative food intake (**f**) was assessed per cage in single-or double-house mice. Data are mean ± s.e.m. *P < 0.05, **P < 0.01, ***P < 0.001. Exact P-values, n-values and detailed statistics are provided in Supplementary Tables 1 and 3 and the Data Source File.

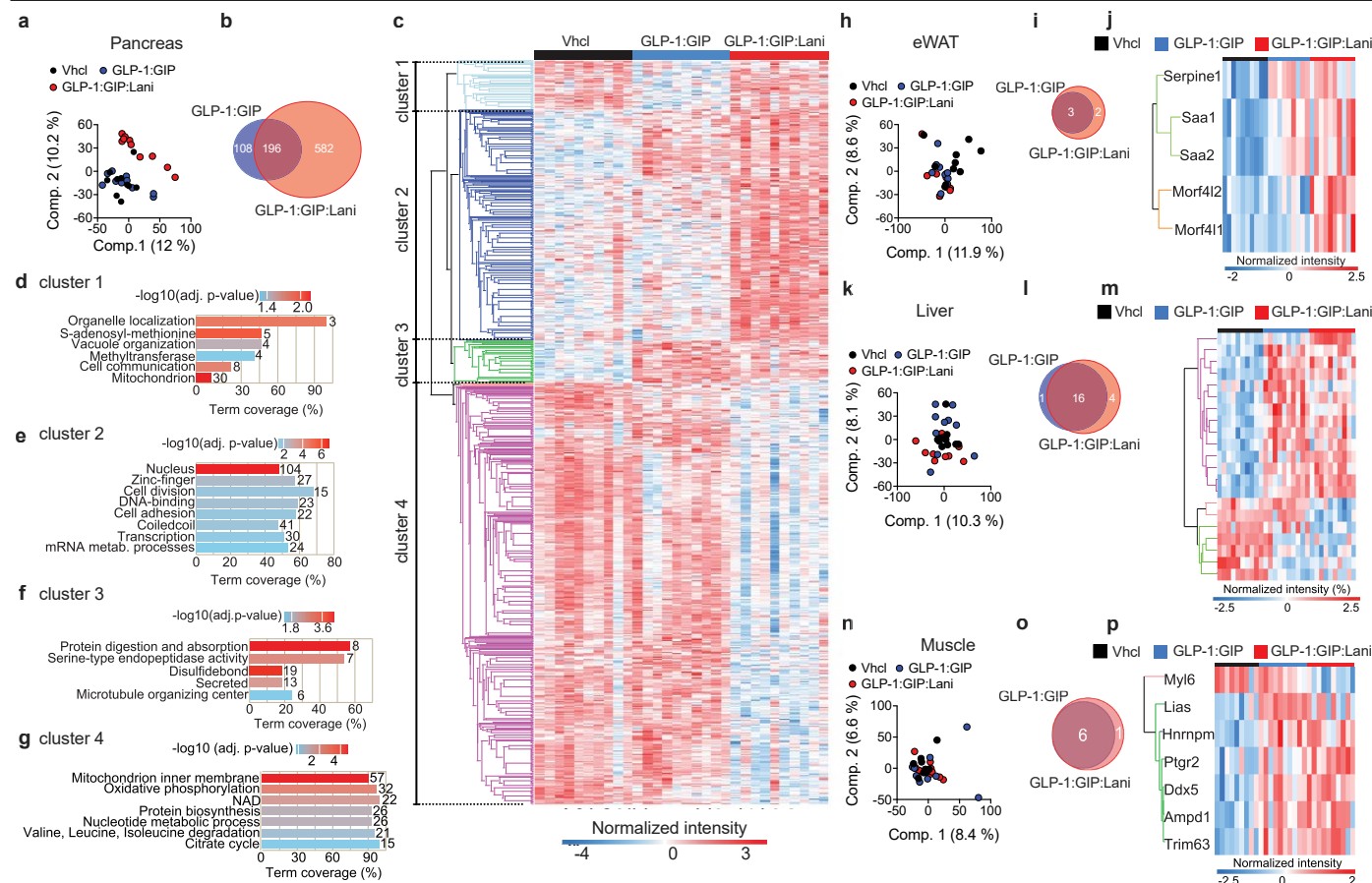

**Extended Data Fig. 8 | Peripheral drug effects. a–p**, DIO mice were treated with a singly s.c. injection of GLP-1:GIP, GLP-1:GIP:Lani or vehicle, followed by tissue harvesting 7 h post treatment. 100 nmol kg⁻¹. **a,h,k,n**, Principal component analysis (PCA) in pancreas (**a**), eWAT (**h**), liver (**k**) and skeletal muscle (**n**). **b,i,l,o**, Venn-diagram with differentially regulated proteins (DRPs) over vehicle in pancreas (**b**), eWAT (**i**), liver (**l**) and skeletal muscle (**o**). **c,j,m,p**, Heatmaps with supervised hierarchical clustering of z-scored intensities for DRPs in pancreas (**c**), eWAT (**j**), liver (**m**) and skeletal muscle (**p**). **d–g**, Selected enriched annotations. **d**, Cluster 1. **e**, Cluster 2. **f**, Cluster 3. **g**, Cluster 4.

Data were analysed using one-way ANOVA, FDR < 0.05 with Tukey's HSD post hoc test (**b,i,l,o**), Fisher's exact test with Benjamini–Hochberg FDR adjustment (**d–g**), or one-way ANOVA, FDR < 0.05 (**c,j,m,p**). Data are mean ± s.e.m. *P < 0.05, **P < 0.01, ***P < 0.001. Data in panel **a–g** comprise n = 10 per group. Data in panel **h–j**, **k–m** comprise n = 10 Vhcl, n = 9 GLP-1:GIP and n = 10 GLP-1:GIP:Lani. Data in panel **n–p** comprise n = 9 Vhcl, n = 10 GLP-1:GIP and n = 10 GLP-1:GIP:Lani. Exact P-values, n-values and detailed statistics are provided in Supplementary Tables 1 and 3 and the Data Source File.

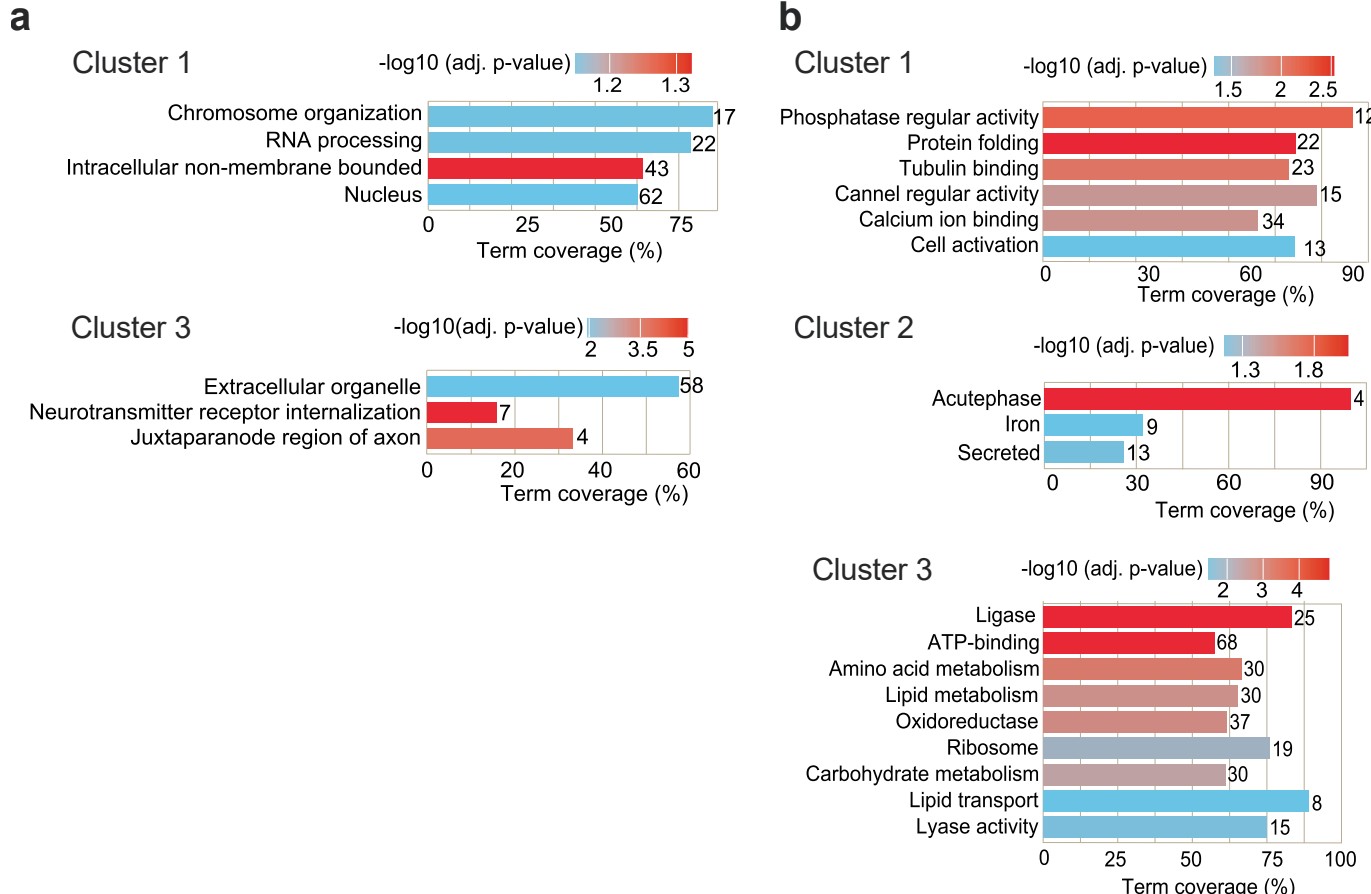

**Extended Data Fig. 9 | Selected enriched annotations for central drug effects. a,b,** Selected enriched annotations corresponding to Fig. 5j,k. **a,** Hindbrain. **b,** Hypothalamus. Data were analysed using Fisher's exact test with Benjamini–Hochberg FDR adjustment (**a,b**). Exact P-values, n-values and detailed statistics are provided in Supplementary Tables 1 and 3 and the Data Source File.

# Reporting Summary

## Statistics

For all statistical analyses, confirm that the following items are present in the figure legend, table legend, main text, or Methods section.

| n/a | Confirmed | |
|---|---|---|
| ☐ | ☒ | The exact sample size (*n*) for each experimental group/condition, given as a discrete number and unit of measurement |
| ☐ | ☒ | A statement on whether measurements were taken from distinct samples or whether the same sample was measured repeatedly |
| ☐ | ☒ | The statistical test(s) used AND whether they are one- or two-sided<br>*Only common tests should be described solely by name; describe more complex techniques in the Methods section.* |
| ☐ | ☒ | A description of all covariates tested |
| ☐ | ☒ | A description of any assumptions or corrections, such as tests of normality and adjustment for multiple comparisons |
| ☐ | ☒ | A full description of the statistical parameters including central tendency (e.g. means) or other basic estimates (e.g. regression coefficient) AND variation (e.g. standard deviation) or associated estimates of uncertainty (e.g. confidence intervals) |
| ☐ | ☒ | For null hypothesis testing, the test statistic (e.g. *F*, *t*, *r*) with confidence intervals, effect sizes, degrees of freedom and *P* value noted<br>*Give P values as exact values whenever suitable.* |
| ☒ | ☐ | For Bayesian analysis, information on the choice of priors and Markov chain Monte Carlo settings |
| ☒ | ☐ | For hierarchical and complex designs, identification of the appropriate level for tests and full reporting of outcomes |
| ☐ | ☒ | Estimates of effect sizes (e.g. Cohen's *d*, Pearson's *r*), indicating how they were calculated |

*Our web collection on statistics for biologists contains articles on many of the points above.*

## Software and code

Policy information about availability of computer code

Data collection    Indirect calorimetry data were collected using Promethion Live (version 24.9.1, Sable Systems International). qPCR data were acquired using QuantStudio (version 1.3, Thermo Fisher Scientific). BRET and ELISA data were acquired using PheraStar (version 4.00 R4, BMG Labtech). FASTQ files were generated using Illumina bcl2fastq Conversion Software (version 2.17.1.14). Brainstem cFOS images were acquired using LAS X (version 3.5.7.23225, Leica Microsystems); hypothalamic cFOS images were obtained by z-stack scanning with AxioScan 7 (ZEN Blue version 3.5, Zeiss). Proteomic data were processed using Spectronaut (v.19.5). For analysis of bulk RNA sequencing data, FASTQ files were generated from base calls using bcl2fastq (v. 2.17.1.14). Quality control for the raw transcriptomics sequencing reads was done using FastQC (v. 0.12.0). Transcript counts were imported into python using scanpy and decoupler (v. 2.14.0). Gene expression z-scores of differentially expressed genes were plotted using Matplotlib (doi.org/10.5281/zenodo.592536), ggplot2 (v.3.5.2) and seaborn (0.13.2). Fiber photometric data were aquired using Synapse (v. 102). Cardiac function was assessed using the Vevo 3100 High-Resolution Micro-Ultrasound Imaging System (FUJIFILM VisualSonics, Toronto, Canada). Blood pressure was measured using the CODA® Monitor tail-cuff system (AD Instruments, Oxford, United Kingdom). Calcium indicator signals from POMC neurons were measured using the LUX RZ10X processor (Tucker-Davis Technologies). Whole-cell current clamp was performed with a Multiclamp 700A amplifier (Molecular Devices, CA). Cultured adipocytes were imaged using Invitrogen™ EVOS™ XL Core Imaging System (Thermo Fisher Scientific). Immortalized mouse brown preadipocytes were analyzed using the XF96 Seahorse extracellular flux analyzer (Agilent Technologies).

Data analysis    Statistical analyses were performed using GraphPad Prism (v.10.03) and SPSS (v.31, IBM). cFOS data were analyzed using QuPath (version 0.4.4, University of Edinburgh). Raw proteomics data were processed with Spectronaut (version 19.5, Biognosys). Imputation and data analysis were performed in Perseus (version 1.6.15.0, Max Planck Institute of Biochemistry). Histological analyses were performed using Visiopharm (version 2018.9, Visiopharm A/S). FastQC (version 0.12.0) was used for quality control of sequencing reads. STAR (v. 2.7.10) was used for alignment to the GRCm38 reference genome. Transcript counts were analyzed in Python (version 3.13) using scanpy (version 1.11.1) and decoupler (version 2.14.0). Gene expression normalization was performed with pyDESeq2 (version 0.5.0). Differential expression analysis used

the Wald test. Data visualization was conducted with Matplotlib (version 3.10.1), ggplot2 (version 3.5.2), and seaborn (version 0.13.2). Functional enrichment analysis was performed using Over Representation Analysis (ORA) via decoupler. Ejection fraction (EF), fractional shortening (FS), cardiac output (CO), stroke volume (SV) and heart rate (HR) were quantified with VevoLab (v.5.10.0). Blood pressure was calculated with LabChart (v. 8.1.24). Electrophysiological data were sampled with an Apple Macintosh computer using AxoGraph X (v. 1.7.3). AP frequency was plotted with Igor Pro (v.9) and Prism (v. 10.0.3). Cultured adipocytes were analysed using ImageJ (v. 1.54)

For manuscripts utilizing custom algorithms or software that are central to the research but not yet described in published literature, software must be made available to editors and reviewers. We strongly encourage code deposition in a community repository (e.g. GitHub). See the Nature Portfolio guidelines for submitting code & software for further information.

## Data

Policy information about availability of data

All manuscripts must include a data availability statement. This statement should provide the following information, where applicable:
- Accession codes, unique identifiers, or web links for publicly available datasets
- A description of any restrictions on data availability
- For clinical datasets or third party data, please ensure that the statement adheres to our policy

Raw data for the proteomic and transcriptomic analysis are available via Pride (PXD062990). Raw data from the bulk RNAseq are available in the GEO under SuperSeries accession number GSE314029. All data used for the statistical analysis are available in the Data Source File, along with the GraphPad Prism-derived report on the statistical analysis. The statistical report contains the mean difference between the treatment groups, the 95% confidence intervals, the significance summary, and the exact p-values (unless p<0.0001). A summary of the statistical tests are along the p-values for the main treatment effects also shown in Supplementary Table 1. Transcripts were aligned using the GRCm38 reference genome (GenBank accession GCA_000001635.20).

## Research involving human participants, their data, or biological material

Policy information about studies with human participants or human data. See also policy information about sex, gender (identity/presentation), and sexual orientation and race, ethnicity and racism.

| | |
|---|---|
| Reporting on sex and gender | n/a |
| Reporting on race, ethnicity, or other socially relevant groupings | n/a |
| Population characteristics | n/a |
| Recruitment | n/a |
| Ethics oversight | n/a |

Note that full information on the approval of the study protocol must also be provided in the manuscript.

# Field-specific reporting

Please select the one below that is the best fit for your research. If you are not sure, read the appropriate sections before making your selection.

☒ Life sciences          ☐ Behavioural & social sciences          ☐ Ecological, evolutionary & environmental sciences

For a reference copy of the document with all sections, see nature.com/documents/nr-reporting-summary-flat.pdf

# Life sciences study design

All studies must disclose on these points even when the disclosure is negative.

| | |
|---|---|
| Sample size | For animal studies, sample sizes were calculated based on a power analysis assuming that a greater or equal (>/=) 5 g difference in body weight between genotypes can be assessed with a power of >/= 75% when using a 2-sided statistical test under the assumption of a standard deviation of 3.5 and an alpha level of 0.05. For in vitro and ex vivo experiments, sample sizes were estimated based on our experience using the same technologies in the same cell system (PMID: 37277609, PMID: 35995995, PMID: 33556643, PMID: 40204014) or animal model (PMID: 38418586, PMID: 37946085). |
| Data exclusions | No data were excluded from the analysis unless scientific (e.g. significant outlier identified by the Grubbs test for outlier) or animal welfare reasons (e.g. injury due to fighting) demanded exclusion. Outliers are stated in the data source file. Four samples were excluded from the Protemic analysis (one per tissue: Quad, brain stem, eWAT, and hypothalamus) as they did not pass quality control. |
| Replication | All reported in vivo, ex vivo and in vitro data correspond to independent biological replicates. The exact samples sizes are reported in the figure legends. |
| Randomization | Animals were either randomly assigned into treatment groups, or were grouped based on their genotype (WT or KO). At study start, only age-matched mice were included in the studies. There were no other covariats controlled. |

| Blinding | For in vivo studies, drugs were aliquoted by a lead scientist in number-coded vials and most, but not all, handling investigators were blinded to the treatment condition. Analyses of glucose and insulin tolerance were performed by experienced research assistants who did not know prior treatment conditions. Studies on drug effects on the cardiovascular system were performed blinded, with only the main PI but not the handling investigators being aware of the treatment groups. Ex vivo and in vitro studies were performed in ID coded vials, and with most, but not all investigators, being blinded to the underlying genotypes and treatment conditions. |

# Reporting for specific materials, systems and methods

We require information from authors about some types of materials, experimental systems and methods used in many studies. Here, indicate whether each material, system or method listed is relevant to your study. If you are not sure if a list item applies to your research, read the appropriate section before selecting a response.

## Materials & experimental systems

| n/a | Involved in the study |
|---|---|
| ☐ | ☒ Antibodies |
| ☐ | ☒ Eukaryotic cell lines |
| ☒ | ☐ Palaeontology and archaeology |
| ☐ | ☒ Animals and other organisms |
| ☒ | ☐ Clinical data |
| ☒ | ☐ Dual use research of concern |
| ☒ | ☐ Plants |

## Methods

| n/a | Involved in the study |
|---|---|
| ☒ | ☐ ChIP-seq |
| ☒ | ☐ Flow cytometry |
| ☒ | ☐ MRI-based neuroimaging |

## Antibodies

| Antibodies used | cFos (#MA5-15055, Invitrogen, 1:400), anti-rabbit Alexa546 (#A10040, Invitrogen, Karlsruhe, Germany, 1:2,000), rabbit anti-insulin (#3014, Cell Signaling Technology, Danvers, USA, 1:800), mouse anti glucagon (#G2654, Merck, Darmstadt, Germany, 1:1000,), goat anti-rabbit AF750 (#A21039, Invitrogen, Karlsruhe, Germany, 1:100), donkey anti-mouse AF555 (#A32773, Invitrogen, Karlsruhe, Germany, 1:200) |
|---|---|
| Validation | cFOS (Invitrogen, #MA5-15055): The cFOS monoclonal antibody Invitrogen #MA5-15055 was verified by Relative expression to ensure that the antibody binds to the antigen stated. The antibody shows reactivity in bovine, hamster, human, mouse, pig and rat. The antibody can be used for western blot, immunhistochemistry, immuncytochemistry, flow cytometry and ChIP assays. The antibody does not cross-react with other Fos proteins, including FosB, FRA1 and FRA2. Immunofluorescence analysis of c-Fos was performed using 70% confluent log phase HeLa cells treated with 200 ng/mL EGF for 30 min. The cells were fixed with 4% paraformaldehyde for 10 minutes, permeabilized with 0.1% Triton™ X-100 for 10 minutes, and blocked with 1% BSA for 1 hour at room temperature. The cells were labeled with c-Fos Monoclonal Antibody (T.142.5) (product # MA5-15055) at 1:250 dilution in 0.1% BSA, incubated overnight at 4 degree Celsius and then labeled with Goat anti-Rabbit IgG (H+L) Superclonal™

Anti-rabbit Alexa546 (#A10040, Invitrogen, Karlsruhe, Germany): The donkey anti-Rabbit IgG (H+L) Highly Cross-Adsorbed Secondary Antibody, Alexa Fluor™ 546 secondary antibody was verified using HepG2 cells stained with alpha-1 antitrypsin Rabbit Polyclonal Primary Antibody (Product # PA5-16661). The antibody was used at a concentration of 4 µg/mL in phosphate buffered saline containing 0.2 % BSA for 45 minutes at room temperature, for detection of alpha-1 antitrypsin in the cytoplasm. Nuclei were stained with DAPI in SlowFade® Gold Antifade Mountant. F-actin was stained with Alexa Fluor® 488 Phalloidin. No nonspecific staining was observed with the secondary antibody alone or with an isotype control. The highly specific antibody has been used in n=63 scientific publications.

Rabbit anti-insulin (#3014, Cell Signaling Technology, Danvers, USA): The Insulin (C27C9) Rabbit Monoclonal Antibody #3014 has been used in n=156 scientific publications. The specificity of this antibody has been verified using western blot and immunhistochemical analysis in paraffin-embedded mouse pancreata. Western blot analysis of lysates (1.0 mg/mL) from INS-1 cells using Insulin (C27C9) Rabbit mAb #3014 show a single band at the expected size. The virtual lane shows the target band at 1:10 and 1:50 dilutions of primary antibody. The corresponding electropherogram plots chemiluminescence by molecular weight along the capillary at 1:10 and 1:50 dilutions of primary antibody. The experiment was performed under reducing conditions on the JessTM Simple Western instrument from ProteinSimple, a BioTechne brand, using the 12-230 kDa separation module.

Mouse anti glucagon (#G2654, Merck, Darmstadt, Germany, 1:1000): This monoclonal Anti-Glucagon antibody produced in mouse has been verified for immunostaining of pancreatic tissue in flow cytometry and immunofluorescence imaging of pancreas cells for immunohistochemistry and morphology of pancreas. The antibody reacts specifically against pancreatic glucagon and exhibits only very weak cross-reaction with gut glucagon (enteroglucagon). The antibody may be used for the immunohistochemical staining of Bouin's-fixed, and formalin-fixed, paraffin-embedded pancreatic tissue sections. Binds to glucagon with an affinity constant of $6.1 \times 10^8$ M-1 in RIA. Monoclonal anti-Glucagon antibody can be used as an analytical tool for quantification of the hormone. It can also be used for immunocytochemical staining of formalin fixed and Bouin-fixed, paraffin-embedded pancreatic tissue sections. Mouse anti-Glucagon antibody reacts specifically with pancreatic glucagon. The product has also shown cross reactivity with glucagon-containing cells in fixed sections of pancreas from dog, mouse, rat, rabbit, porcine, guinea pig, cat and human and weak cross reactivity for gut glucagon (enteroglucagon). Monoclonal Anti-Glucagon reacts with pancreatic glucagon in RIA and immunocytochemistry. The affinity constant of $6.1 \times 10^{(8)}$ L/M in RIA. The antibody weakly cross-reacts with gut glucagon (enteroglucagon) in an immunohistological assay. Cross-reactivity has been observed with glucagon-containing cells in fixed sections of pancreas from human, porcine, dog, rabbit, mouse, rat, guinea pig, and cat. |

Goat anti-rabbit AF750 (#A21039, Invitrogen, Karlsruhe, Germany, 1:100): Goat anti-Rabbit IgG (H+L) Cross-Adsorbed Secondary Antibody, Alexa Fluor™ 750 has been used in 123 scientific manuscripts. Verification included immunofluorescence analysis of Goat anti-Rabbit IgG (H+L) Cross-Adsorbed Secondary Antibody, Alexa Fluor™ 750 (Product # A-21039) using MCF 10A (positive model) and T-47D (negative model) cells stained with Vimentin Polyclonal Antibody (Product # PA5-27231). The cells were fixed with 4% paraformaldehyde for 10 minutes, permeabilized with 0.1% Triton™ X-100 for 10 minutes, blocked with 1% BSA for 1 hour and labeled with 2 µg/mL primary antibody for 3 hours at room temperature. Goat anti-Rabbit IgG (H+L) Cross-Adsorbed Secondary Antibody, Alexa Fluor™ 750 (Product # A-21039, 1:2000 dilution) in 0.1% BSA in PBS for 45 minutes at room temperature, was used for detection of Vimentin in the cytoplasm. Nuclei were stained with Hoechst33342 (Product # H1399). F-actin was stained with Alexa Fluor® 488 Phalloidin (Product # A12379, 1:300). The specificity of the secondary antibody was proved by the absence of signal in T-47D (negative model for vimentin) due to no primary antibody binding. Nonspecific staining was not observed with secondary antibody alone. The images were captured at 40X magnification in CellInsight CX7 LZR High-Content Screening (HCS) Platform (Product # CX7A1110LZR) and externally deconvoluted.

Donkey anti-mouse AF555 (#A32773, Invitrogen, Karlsruhe, Germany, 1:200): Donkey anti-Mouse IgG (H+L) Highly Cross-Adsorbed Secondary Antibody, Alexa Fluor™ Plus 555 has been used in 182 scientific manuscripts. Verification of the antibody includes immunofluorescent analysis of tubulin in A549 cells. The cells were fixed with 4% formaldehyde for 20 mins, permeabilized with 0.5% Triton X-100 in PBS for 20 mins, washed 3X in PBS and blocked with 3% BSA in PBS for 30 mins at RT. Cells were stained with a tubulin antibody at a dilution of 1:2000 in 3% BSA in PBS for 1 hr at RT, washed 3X in PBS and then incubated with Invitrogen Alexa Fluor Plus 555 donkey anti-mouse IgG secondary antibody (Product # A32773) prepared in 3% BSA in PBS at a dilution of 1:1000 for 1 hr at RT in the presence of NucBlue Live ReadyProbes Reagent (Product # R37605). The image contains overlay of tubulin and nuclei. Images were taken on an EVOS FL Auto 2 Imaging System (Product # AMAFD2000) with an Olympus 40X Super Apochromat objective (Product # AMEP4754) at 40X magnification. Actin was stained using Alexa Fluor Plus Phalloidin (Product # A30105).

# Eukaryotic cell lines

Policy information about cell lines and Sex and Gender in Research

| Cell line source(s) | HEK293T cells (ATCC, USA) |
|---|---|
| Authentication | Authentication according to the manufacturer's website: The 293T cell line, originally referred as 293tsA1609neo, is a highly transfectable derivative of human embryonic kidney 293 cells, and contains the SV40 T-antigen.This cell line is competent to replicate vectors carrying the SV40 region of replication. It gives high titers when used to produce retroviruses. It has been widely used for retroviral production, gene expression and protein production. Product related references include DuBridge et al., Mol Cell Biol. 1987 Jan;7(1):379-87 and Pear et al., Proc Natl Acad Sci U S A. 1993 Sep 15;90(18):8392-6. https://www.lgcstandards-atcc.org/Products/All/CRL-3216.aspx?geo_country=de#generalinformation |
| Mycoplasma contamination | cell lines were free of mycoplasm contaminations |
| Commonly misidentified lines (See ICLAC register) | no misidentified cell lines were used in the manuscript |

# Animals and other research organisms

Policy information about studies involving animals; ARRIVE guidelines recommended for reporting animal research, and Sex and Gender in Research

| Laboratory animals | Figure 1a,b: 43 wk old male DIO wildtype C57BL6/J mice |
|---|---|
| | Figure 1c-e: 45 wk old male DIO wildtype C57BL6/J mice |
| | Figure 1h: 28-34 wk old male wildtype C57BL6/J mice |
| | Figure 1l: 43 wk old male DIO wildtype C57BL6/J mice |
| | Figure 1m: 45 wk old male DIO wildtype C57BL6/J mice |
| | Figure 1n: 43 wk old male DIO wildtype C57BL6/J mice |
| | Figure 1o: 45 wk old male DIO wildtype C57BL6/J mice |
| | Figure 2a: 33 wk old male DIO wildtype C57BL6/J mice |
| | Figure 2b: 35 wk old male DIO wildtype C57BL6/J mice |
| | Figure 2c: 33 wk old male DIO wildtype C57BL6/J mice |
| | Figure 2d-f: 35 wk old male DIO wildtype C57BL6/J mice |
| | Figure 2g-j: 28-30 wk old male DIO wildtype C57BL6/J mice |
| | Figure 2k-m: 55 wk old male DIO wildtype C57BL6/J mice |
| | Figure 2n,o: 35 wk old male DIO C57BL6/J mice |
| | Figure 2p,q: 8 wk old male chow fed C57BL6/J mice |
| | Figure 2r-v: 27wk old male DIO C57BL6/J mice |
| | Figure 3a: 40 wk old male DIO wildtype C57BL6/J mice |
| | Figure 3b: 45 wk old male DIO wildtype C57BL6/J mice |
| | Figure 3c: 40 wk old male DIO wildtype C57BL6/J mice |
| | Figure 3d: 42 wk old male DIO wildtype C57BL6/J mice |
| | Figure 3e: 45 wk old male DIO wildtype C57BL6/J mice |
| | Figure 3f,g: 43 wk old male DIO wildtype C57BL6/J mice |
| | Figure 3h-m: 45 wk old male DIO wildtype C57BL6/J mice |
| | Figure 3n,o: 51 wk old male DIO wildtype C57BL6/J mice |
| | Figure 3o: 47 wk (Day 0) and 51 (Day 26) wk old male DIO wildtype C57BL6/J mice |
| | Figure 3p: 51 wk old male DIO wildtype C57BL6/J mice |

Figure 3q-v: 20 wk old male DIO wildtype C57BL6/J mice
Figure 4a,b: 14-24 wk old male C57BL6/J mice wildtype or Vglut2 Cre GLP-1R KO mice
Figure 4c: 39 wk old male C57BL6/J DIO wildtype and global Gipr KO mice
Figure 4d,e: 68 wk old male DIO wildtype C57BL6/J mice
Figure 4f: 70 wk old male DIO wildtype C57BL6/J mice
Figure 4g: 56 wk old male DIO wildtype C57BL6/J mice
Figure 4h: 55 wk old male DIO wildtype C57BL6/J mice
Figure 4i,j: 32 wk old male DIO wildtype or double incretin receptor KO C57BL6/J mice
Figure 4k-m: 33 wk old male DIO wildtype or double incretin receptor KO C57BL6/J mice
Figure 5a-f: 45 wk old male DIO C57BL6/J mice
Figure 5h-m: 32 wk old male DIO C57BL6/J mice
Figure 5n,o: Male chow fed Pomc-cre)16Lowl/J mice (N=5, 10-17 wk old; N=1, 30 wk old)
Figure 5p: 8-10 wk old male chow fed C57BL6/J POMC-GFP mice
EDF2c-h: 45 wk old male DIO wildtype C57BL6/J mice
EDF3a: 27 wk old male DIO wildtype C57BL6/J mice
EDF3b: 28 wk old male DIO wildtype C57BL6/J mice
EDF3c: 27 wk old male DIO wildtype C57BL6/J mice
EDF3d,e: 47 wk old male DIO C57BL6/J mice
EDF3f-i: 49 wk old male DIO C57BL6/J mice
EDF3j-l: 33 wk old male DIO wildtype C57BL6/J mice
EDF4a-l: 35 wk old male DIO C57BL6/J mice
EDF5a-d: 35 wk old male DIO wildtype C57BL6/J mice
EDF5f-s: Immortalized BAT cells were obtained from 6-8 wk old chow-fed female C57BL6/J mice
EDF6a: 10 wk old male C57BL6/J db/db mice
EDF6b: 13 wk old male C57BL6/J db/db mice
EDF6c: 10 wk old male C57BL6/J db/db mice
EDF6d: 11 wk old male C57BL6/J db/db mice
EDF6e: 12 wk old male C57BL6/J db/db mice
EDF6f-h: 13 wk old male C57BL6/J db/db mice
EDF6i: 14-24 wk old male C57BL6/J wildtype and Vglut2-Cre GLP-1R flx/flx mice
EDF6j: 24 wk old male wildtype C57BL6/J mice
EDF6k: 26 wk old male wildtype C57BL6/J mice
EDF6j: 24 wk old male wildtype C57BL6/J mice
EDF6k: 26 wk old male wildtype C57BL6/J mice
EDF7a: 47 wk old male DIO wildtype C57BL6/J mice
EDF7b-e: 49 wk old male DIO wildtype C57BL6/J mice
EDF7c: 47 wk old male DIO wildtype C57BL6/J mice
EDF7g: 31 wk old male DIO wildtype C57BL6/J mice
EDF7e: 32 wk old male DIO wildtype C57BL6/J mice
EDF8a-p: 32 wk old male DIO C57BL6/J mice
EDF9a,b: 32 wk old male DIO C57BL6/J mice

| | |
|---|---|
| Wild animals | No wild animals were used in the study |
| Reporting on sex | All studies were performed in male mice, since female mice are largely resistant to the development of diet induced obesity and the development of high-fat diet induced insulin resistance. |
| Field-collected samples | No field collected samples were used in the study |
| Ethics oversight | Experiments were performed in accordance with the Animal Protection Law of the European Union after permission by the Governments of Upper Bavaria, Germany, or Copenhagen, Denmark, or by the Institutional Animal Care and Use Committees of the Universities of Texas Southwestern, Michigan, Duke or Yale, USA. |

Note that full information on the approval of the study protocol must also be provided in the manuscript.

# Plants

| | |
|---|---|
| Seed stocks | n/a |
| Novel plant genotypes | n/a |
| Authentication | n/a |

