## [Peer Review file · Nature]

GLP-1R–GIPR–pan-PPAR quintuple agonism corrects obesity and diabetes in mice

Corresponding Author: Professor Timo Müller

Reviewer #1 withdrew themselves in the second round of review. Their expertise is shared by the other two referees. Referee #3 felt that there was insufficient mechanistic insights, which were editorially overruled.

Version 0:

Reviewer comments:

Referee #1

(Remarks to the Author)

Overall Assessment

The manuscript from Liskiewicz et al. presents a multifunctional molecule targeting GLP-1R, GIPR, and PPAR $\alpha/\gamma/\delta$, with the aim of improving metabolic outcomes. While the concept of combining incretin signaling with PPAR modulation is interesting, the addition of PPAR δ to the GLP-1RA/tesaglitazar conjugate described in Nature Metabolism (2022) feels like a modest step rather than a major advance. The manuscript currently lacks a clear mechanistic or clinical rationale to justify the design. The broad experimental scope comes at the cost of clarity, making it hard to interpret the data or understand the main message.

Scientific Rationale and Innovation

The manuscript does not clearly explain why targeting all five pathways in a single molecule is necessary or beneficial. It's unclear how this design improves on existing drugs like tirzepatide or lanifibranor, or why approved agents used in combination wouldn't be sufficient. The novelty of the approach, both mechanistically and clinically, needs to be better defined. Safety considerations also need to be addressed more directly.

Experimental Design and Model Use

The use of multiple animal models is not well justified. Some models seem redundant, and it is not clear how each contributes uniquely to understanding the compound's effects. A more focused approach would help—fewer models, with each aligned to a specific mechanistic question, would strengthen the study.

Clarity and Structure

The manuscript is difficult to follow due to a lack of narrative flow. The transitions between sections are abrupt, and data from complex models are not well integrated or discussed. Reorganizing the manuscript to better connect rationale, results, and conclusions would improve readability and impact.

Recommendations for Improvement

1. Better explain the rationale for targeting all five pathways. What unique advantage does this provide?
2. Streamline the experimental plan and clearly justify each model.
3. Improve the structure and flow so the data better support the conclusions.
4. More explicitly discuss the clinical relevance—how this compares to existing treatments or combinations of approved drugs.

Specific Questions

Biodistribution: Is there direct evidence showing the biodistribution of the GLP-1:GIP:Lani molecule? The cited studies don't examine the full construct. Could this affect biodistribution or tissue targeting?

Line 162, Extended Data Fig. 2c–e:

Do the cAMP results suggest greater potency at GLP-1R? EC₅₀ values should be calculated and reported.

In Fig. 2e, is the experiment under high or low glucose? The legend is unclear—please clarify or add the missing data.

Line 175 (DIR⁻ cells): Why does GLP-1:GIP:Lani show no activity in DIR⁻ cells? Could conjugation affect receptor binding? If uptake doesn't depend on GLP-1R or GIPR, what explains the lack of effect?

Line 179: The reported 3-fold greater weight loss seems overstated—please check the data and adjust.
Fig. 2o (Weight loss vs. GTT): Weight loss is similar between GLP-1:GIP:Lani and GLP-1:GIP:Tesa, but GTT outcomes differ. Could this reflect differences in β -cell function or other mechanisms?
Line 252 (db/db model): What's the rationale for using GLP-1:GIP:Lani in db/db mice? TG and cholesterol levels are unchanged despite weight loss. Could this be due to short treatment? How does this relate to elevated liver FFA/cholesterol (Line 302)?
Line 330 (Adipose-specific GIPR): Again, what is the rationale for using this model? What data support the statement about greater weight loss in GIPR TG mice? Please provide or reference the results.
Line 378 (c-fos): If c-fos activation is similar, how does that relate to greater reduction in food intake with GLP-1:GIP:Lani versus GLP-1:GIP?
Line 486: GLP-1:GIP:Lani does not appear to improve liver fibrosis, which is a key benefit of Lanifibranor in Phase 3. This limits the value of combining the two and should be discussed more in the manuscript.
Line 496 (Brain penetration): Is there any evidence that Lani crosses the blood–brain barrier? Have c-fos effects been tested at clinically relevant doses?

Referee #2

(Remarks to the Author)

This manuscript presents results from studies which continue on a previously reported concept, ie the pharmacological development of unimolecular, multi-target pharmacological agents combining a peptide (eg a dual GIPR:GLP1R agonist) and a small molecule (ie the pan-PPAR agonist lanifibranor). The paper is well-written and the data are clearly presented. However, the conceptual novelty of this study is very limited and the mechanistic insights are lacking. Indeed, the clinical utility/superiority of a combined GLP1R:GIPR agonist (ie tirzepatide) has been amply demonstrated. Moreover, the interest of combining a peptide agonist with a small molecule PPAR agonist has already been demonstrated by the authors in their previous study (Nat. Metab., 2022, 4, 1071-1083). Unfortunately, the added benefit of the PPARdelta activity is unclear (as this is the case also in clinical studies) and has not been addressed. Altogether, the potential interest of this concept has been readily demonstrated in preclinical studies. Given the difficulties of translating such findings to the clinic, a POC clinical study would be of great interest now.

Other points:

- Semaglutide and tirzepatide are reported to be efficacious in the treatment of MASH. Lanifibranor is in Phase 3 clinical development for MASLD, but the rationale for selecting this compound in the present study is unclear, since there have been no studies using MASH-inducing diets. Moreover, the clinical lipid/glucose modifying effects of tesaglitazar have already been demonstrated and those of lanifibranor appear very similar, as for example recently reported in Barb et al. J.Hep.82,979-991. The CNS effect has also been shown with the tesaglitazar-GLP1 combination treatment. What is the added value of GIPR:PPARdelta agonism in the current study?
- Figure 2h: PPARg agonism rather increases food intake related to the down-regulation of leptin in adipose tissue (which may contribute to the observed effect in db/db mice in figure 4 and the higher food intake in figure 6d). What is the mechanism of the decrease in food intake by GLP1:GIP:Lani?
- Figure 4: why doesn't GLP1:GIP decrease blood glucose in db/db mice? Why doesn't GLP1:GIP:Lani increase liver FFA given that PPARg activation rather decreases FFA fluxes from adipose tissue.
- Mechanistic studies addressing tissue-specific effects of activation of the PPAR subtypes, admittedly very difficult, are lacking. Why is GLP1R:GIPR:Lani more efficient than the GLP1R:Lani + acylGIP combination on glucose (Figure 5C). Is this related to the tissue-distribution of lanifibranor? If so, which tissues/PPARs are involved?
- The well-known adverse effects of PPARg agonism (body weight increase, fluid retention and bone fractures) are likely also observed with lanifibranor in the clinic. It would be a real advance to see whether these effects are no longer observed with GLP1:GIP:Lani.
- Figure 2 and others: it is not a surprise that lanifibranor at 10nmol/kg/day is inefficacious. A proper control would be the murine-equivalent clinically used dose of lanifibranor and compare to this dose (also for side effects).
- Figure 7: via which PPAR? Difference with tesaglitazar?

Minor points:

- M&M, line 591 : single or double-housed mice: not a classical way of animal experimentation. Please clarify.
- lines 272, lines 285-287 + extended fig6: taste avoidance experiment is performed in lean mice. How can one be sure that the decreased body weight under other experimental conditions is not due to nausea? Avoidance, or preference, could be different in DIO mice.
- Figure 3n, Figure 4p, q, r: liver weight data should be expressed as % of body weight (as body weight changes with treatment).
- Fig 6 and line 355: why was glucagon measured in this experiment and not in others?
- M&M, line 593: subcutaneous administration of the compounds: volume?
- M&M, line 594: is vehicle composition identical for all compounds?
- Figure 3m, lines 253-254: why was spleen weight measured?

Referee #3

(Remarks to the Author)

The manuscript by Liskiewicz et al designed a unimolecular quintuple agonist (GLP-1:GIP:Lani) that has similar effects in

stimulating insulin in vitro, but far greater effects on food intake and body weight reduction and glucose regulation in vivo in diet-induced and genetic obesity. This is thought to be mediated by effects on gene programs in key tissues including brainstem. While this manuscript provides a very thorough phenotyping of GLP-1:GIP:Lani's impressive effects compared to many other drugs and conditions (GLP-1:GIP:Tesa, GLP-1:GIP, GLP-1:GIP + Lani, Sema, GLP-1:GIP pair fed, etc.), the lack of causal mechanistic insight into the potentiated effects of GLP-1:GIP:Lani diminish enthusiasm for the current manuscript. Nonetheless, this work as is would be appropriate for publication in a more specialized journal.

Specific comments:

- Information about the GLP-1:GIP agonists used to create the quintuple agonist should be provided in the results and methods sections.
- Figures 1-6 and associated supplements are very important data but all come to the same conclusion – that GLP-1:GIP:Lani is a superior drug for weight and food intake reduction and glycemic control. In this reviewer's opinion, these main figures could be condensed to display only key data in main figures: the comparison of either GLP-1:GIP to GLP-1:GIP:Lani or GLP-1:GIP:Lani to GLP-1:GIP + Lani co-treatment. Further, the in vitro data demonstrating effects in GIP or GLP-1 expressing cells but not in incretin KO cells is key and should be in the main figure. All other data could be supplemental.
- The exploration of potential side effects of this novel agonist is minimal compared to the thorough exploration of its effects on body weight and blood glucose regulation compared to nearly all possible comparison groups. For example, functional tests (beyond histology) on cardiovascular and renal function would be warranted, especially given known effects of PPAR agonists on these systems in clinical studies.
- The graphs depicting cumulative food intake are informative but may diminish important information. For example, it looks like food intake may be affected in short term but may normalize (equal slopes) after chronic intake? Therefore, daily food intake would also be informative here.
- Related, the authors suggest in Figure 6 that the effects on body weight may be independent of effects on food intake. How do the authors reconcile this with their data showing no significant effects on energy expenditure?
- It appears that most if not all studies were performed in male mice, however, it would be important to determine if there are potential sex differences in the ability for the quintuple agonist to reduce body weight and improve glycemia.
- All of the studies were performed with 10 nmol/kg doses of drugs. Why was this dose switched to 50 nmol/kg for the proteomics experiments? This is a potential confound when using these results as potential explanations for drug effects.
- While the authors used RNA sequencing and proteomics as an attempt to determine mechanisms for the synergistic effects of the multiagonist, all of this is observational with no causal mechanisms.

Referee #4

(Remarks to the Author)

I co-reviewed this manuscript with one of the reviewers who provided the listed reports.

Version 1:

Reviewer comments:

Referee #2

(Remarks to the Author)

None

Referee #3

(Remarks to the Author)

The authors added a substantial number of additional experiments in the revised manuscript; these addressed some but not all concerns. Most notably for this reviewer is the lack of mechanistic data explaining the large effect size on body weight. In my mind, the new knockout data are not particularly helpful – it is fully expected that knockouts for the incretin receptors should impair weight loss. Further, the manuscript implies that these effects are different than the knockout effects with dual agonism treatment, yet they are quite similar to me (the only difference is the effect size). Given that the body weight effects are likely mediated by CNS control, what is missing is a mechanistic investigation from this perspective. What was provided – fiber photometry data in hypothalamic POMC neurons – is not very satisfying. First, as the authors note, hindbrain changes in their studies were most robust, and this is consistent with the emerging literature that the hindbrain is the primary mediator at least of GLP-1 drugs, and so it is unclear why the authors chose to investigate the hypothalamus which is likely dispensable for the effect of these obesity drugs. Second, fiber photometry data show very short recordings – 400 s long – with activity traces equalizing between GLP-1:GIP and GLP-1:GIP:Lani by the end of this short recording. Therefore, I am very skeptical that this adds any mechanistic insight to the enhanced metabolic effects of GLP-1:GIP:Lani that occur over days/weeks.

Editor

Following a discussion of the editorial team, we are willing to wave all novelty concerns raised by the reviewers, given the dramatic effect of the compound. We ask that a revised study provide mechanistic explanation as to why 1) the addition of PPAR δ signaling has such a profound affect compared to the previous iteration of the drug with only PPAR α /b agonism and 2) the importance of GLP1R/GIPR targeting to specific cell types enhances the specificity/effectiveness of PPAR targeting.

We thank the editorial team for their fair and constructive evaluation of our work. In the revised manuscript, we now provide a series of mechanistic data that help to better understand the individual contribution of the targeted receptors, including PPAR δ . Accordingly, we now show that weight loss induced by GLP-1:GIP:Lani is, in contrast to GLP-1R:GIPR co-agonism, strikingly impaired in mice with *Vglut2-Cre*-mediated loss of GLP-1R in glutamatergic neurons (**Fig. 4a,b**). Furthermore, we now show that weight loss induced by GLP-1:GIP:Lani is diminished in DIO mice with global lack of GIPR (**Fig. 4c**), confirming that GLP-1:GIP:Lani decreases body weight via both incretin receptors.

Consistent with that GLP-1R agonists activate POMC neurons via hypothalamic glutamatergic GLP-1R signaling (PMID: 34626854), we further now show using fiber-photometry in *POMC-Cre* mice that GLP-1:GIP:Lani more robustly induces POMC neuronal activity compared to GLP-1R:GIPR co-agonism, and this is confirmed also *ex vivo* using electrophysiology in *POMC-GFP* mice (**Fig. 5n-p**).

Collectively, these data show that GLP-1:GIP:Lani reduces body weight via both incretin receptors and that this coincides with more robust activation of POMC neuronal activity in the hypothalamus. Notably, we also show that weight loss induced by GLP-1:GIP is, in contrast to GLP-1:GIP:Lani, only mildly diminished in the *Vglut2-GLP-1R* KO mice (see data above), indicating that the delivery of Lani into the glutamatergic GLP-1R neurons is crucial for the metabolic effects of the drug. Consistent with a role of GIPR, weight loss induced by GLP-1:GIP:Lani is comparable to co-therapy of GLP-1:Lani and a long-acting (acyl) GIP (**Fig 4d**). However, despite comparable weight loss, GLP-1:GIP:Lani outperforms the co-therapy to yield greater improvement of glucose tolerance (**Fig 4f**), indicating that enhanced weight loss by GLP-1:GIP:Lani originates from the action of Lani in *Vglut2* GLP-1R neurons, while delivery of Lani into GIPR cells further improves glucose control.

Confirming the contribution of PPAR δ for the glycemic effects of GLP-1:GIP:Lani, we now show that selective antagonization of PPAR δ using GSK3787 blocks the glucose lowering effect of GLP-1:GIP:Lani (Fig 4g), but interestingly without affecting the ability of the selective PPAR δ agonist GW501516 (or GLP-1:GIP:Lani) to decrease body weight (Fig 4h). These data hence show that GLP-1:GIP:Lani decreases body weight via both incretin receptors, while further improving glucose metabolism via PPAR δ . Moreover, these data indicate that the PPAR δ agonist GW501516, and likely also the PPAR moiety of GLP-1:GIP:Lani, also decrease body weight via mechanisms unrelated to PPAR δ .

The importance of PPAR α and γ is most notably demonstrated by their ability to improve GLP-1 effects on glucose metabolism. Consistent with this, we already showed previously that GLP-1:Tesa (Tesaglitazar is a PPAR α,γ co-agonist), despite having only modest effects on body weight, solidly enhances glucose metabolism over its pharmacokinetically-matched GLP-1 backbone (PMID: 35995995, see data below). Accordingly, the contribution of PPAR α and γ lie in their additional glycemic effects, without overt ability to drive greater weight loss.

Nonetheless, consistent with previous reports showing that all three PPAR isoforms improve glucose metabolism in obese rodents (PMID: 27636730), we show in our manuscript that GLP-1:GIP:Lani not only solidly outperforms GLP-1:GIP:Tesa to yield greater decreases in body weight and fat mass, but also to further decrease blood glucose (EDF2f-h). And these effects are now further corroborated by showing a direct role of PPAR δ for the blood glucose lowering effect of GLP-1:GIP:Lani (Fig 4g).

In our revised manuscript, we now also provide additional mechanistic insights into how GLP-1:GIP:Lani improves glucose metabolism. Consistent with the main action of PPAR agonism, we now show that GLP-1:GIP:Lani outperforms GLP-1:GIP to strongly upregulate anti-inflammatory gene programs in the liver and skeletal muscle (**Fig 2n,o**).

Consequently, GLP-1:GIP:Lani not only outperforms GLP-1:GIP to further improve insulin sensitivity (**Fig 2g-i**), but also to further suppress endogenous glucose production (**Fig 2j,k**), and this is mechanistically linked to decreased hepatic expression of *pyruvate carboxylase (Pc)* and *phosphoenolpyruvate carboxykinase (Pepck)*, the two master regulator of gluconeogenesis (**Fig 2i,m**).

Notably, PPAR γ agonists not only act on mature adipocytes to improve insulin sensitivity, but also on adipocyte precursors to induce adipocyte differentiation and body weight gain (**PMID: 27636730**). The latter is of particular importance since GIPR is expressed in only mature adipocytes but not preadipocytes (**PMID: 21245029**) and is therefore assumed to restrict the action of GLP-1:GIP:Lani on mature adipocytes while restraining its action on preadipocytes. Consistent with this, we found GLP-1:GIP:Lani, in contrast to Rosiglitazone, unable to induce adipocyte differentiation (**Figure 2p,q**), but with greater glucose uptake into the brown adipose tissue (**Figure 2r**) and comparable glucose uptake into eWAT, skeletal muscle, liver and heart (**Figure 2s-v**). These data hence show that GLP-1:GIP:Lani protects from the adverse effects of PPARs to increase body weight via stimulation of adipocyte differentiation, while at the same time preserving its ability to act on mature adipocytes to increase adipose tissue glucose uptake.

Our data are consistent with many reports showing that BAT glucose uptake is an indicator of insulin sensitivity and is not necessarily associated with changes in mitochondrial bioenergetics and energy expenditure (PMID: 28082439; PMID: 30135070, PMID: 33546400, PMID: 20606075, PMID: 23221344). In Agreement with this, we find no changes in expression of key thermogenic genes in BAT following treatment with either GLP-1:GIP:Lani or GLP-1:GIP (EDF5a-d).

We also now show that neither GLP-1:GIP nor GLP-1:GIP:Lani acutely or chronically affect whole-body energy expenditure (EDF3j) or mitochondrial function in differentiated BAT primary cells (EDF5e-s). Specifically, mitochondrial respiration, proton production rate, proton leak respiration, coupling efficiency, maximal substrate oxidation, total and non-mitochondrial respiration, ATP-linked respiration, glycolysis, and glycolytic ATP production all remain unchanged after treatment with GLP-1:GIP:Lani or GLP-1:GIP (EDF5e-s).

Collectively, and consistent with PPAR agonism, these data indicate that the superior glycemic benefits of GLP-1:GIP:Lani relative to GLP-1:GIP originate from its ability to further improve insulin sensitivity through decreased inflammation in key glucometabolic tissues, leading to enhanced suppression of hepatic glucose production and increased glucose uptake into key insulin sensitive and glucometabolic tissues.

Additionally, if there is any additional pre-clinical data in large mammals (pigs, non-human primates), or any preliminary human data, we would encourage you to include this in the revised study.

Unfortunately, the GIP system is, in contrast to the GLP-1 system, not well conserved between mice and humans (PMID: 40024571). The here used GLP:GIP:Lani is optimized for activating the mouse GIP receptor and requires structural optimization for studies in higher species. While drug optimization for clinical studies is ongoing, such data are unfortunately not available for this manuscript.

Reviewer 1

Overall Assessment

The manuscript from Liskiewicz et al. presents a multifunctional molecule targeting GLP-1R, GIPR, and PPAR $\alpha/\gamma/\delta$, with the aim of improving metabolic outcomes. While the concept of combining incretin signaling with PPAR modulation is interesting, the addition of PPAR δ to the GLP-1RA/tesaglitazar conjugate described in Nature Metabolism (2022) feels like a modest step rather than a major advance.

We appreciate the reviewer's comment. The advancement of GLP-1:GIP:Lani relative to the published GLP-1:Tesa lies in its markedly enhanced metabolic effects rather than in the concept of peptide-mediated nuclear hormone delivery itself. At a dose of 50 nmol/kg, GLP-1:Tesa decreased body weight in DIO mice by only -10% over Vhcl controls, while GLP-1:GIP:Lani achieved at the same dose placebo-corrected weight loss of -35.73% without even reaching a plateau (**Fig. 1a and I**).

Notably, this magnitude of weight loss exceeds every published report on tirzepatide (PMID: 34003802; 30473097; 39612941; 37840407; 35809773; 35921984) and is comparable to that of the GLP-1:GIP:Glucagon triagonist retatrutide (PMID: 35985340), which however has cardiovascular and acute hyperglycemic liabilities due to glucagon receptor agonism. Retatrutide increased heart rate by approximately +12 bpm in non-human primates (PMID: 35985340) and by +5.6–7.5 bpm in humans (PMID: 37366315, PMID: 4013943). Although Lanifibranor has shown cardiovascular benefits in patients with MASH (PMID: 38730247), it is used clinically at doses of 1.2 g/day (2.76 million nmol/day) and leads at this dose to body weight gain, anemia, leukopenia, fluid retention (peripheral oedema), and diarrhoea (PMID: 34670042).

GLP-1:GIP:Lani notably allows for the use of Lani at doses of as little as 10 nmol/kg. In our revised manuscript, we now show that GLP-1:GIP:Lani at this dose not only decreases body weight in a range previously reported for retatrutide, but also that it decreases HFD-induced heart hypertrophy without causing anemia, fluid retention, or changes in urinary/plasma creatinine (**Fig 3m-p**). Furthermore, we now show that GLP-1:GIP:Lani does not affect blood pressure but improves CV function with slight superiority to GLP-1R:GIPR co-agonism, including decreased heart rate and increased ejection fraction, fractional shortening, stroke volume and cardiac output (**Fig 3q-v**).

Together with the demonstration that GLP-1:GIP:Lani decreases marker for liver damage (AST and ALT) (**Fig 3i,j**), these data clearly show that GLP-1:GIP:Lani represents a significant advancement over existing incretin-based therapies. This is notably further supported by our demonstration that GLP-1:GIP:Lani completely prevents the development of obesity and diabetes in db/db mice (**EDF6a-h**), a model in which tirzepatide, semaglutide and retatrutide all failed to prevent the development of obesity (<https://doi.org/10.2337/db25-2169-LB> and PMID: 39344853).

The manuscript currently lacks a clear mechanistic or clinical rationale to justify the design. The broad experimental scope comes at the cost of clarity, making it hard to interpret the data or understand the main message.

We are sorry that this reviewer struggled to understand our data, and hope that the revised manuscript and this rebuttal letter will help to more clearly convey the main message of our work. The reviewers comment also stands in marked contrast to the feedback of the other reviewers. For instance, Reviewer 2 noted that “*the paper is well written, and the data are clearly presented,*” and Reviewer 3 stated that “*this manuscript provides a very thorough phenotyping of GLP-1:GIP:Lani’s impressive effects compared to many other drugs and conditions.*”

The rationale for developing GLP-1:GIP:Lani was to combine the insulin-sensitizing and anti-inflammatory properties of a PPAR agonist with the body weight and blood glucose lowering effects of a GLP-1R:GIPR co-agonist within a single, highly potent molecule. Starting with GLP-1:GIP:Tesa, we found this molecule to outperform GLP-1:GIP co-agonism to further improve glucose metabolism, but with only limited ability to further enhance weight loss. We then exchanged Tesa with the PPAR α,γ,δ triagonist Lanifibranor, since this molecule was shown preclinically to ameliorate HFD-induced body weight gain (PMID: 32360434, PMID: 29404476). The unimolecular design was employed to enable low-dose delivery of Lani specifically to cells expressing incretin receptors with the intention to preserve its anti-inflammatory and insulin-sensitizing actions while minimizing its clinical liability to cause weight gain, anemia and fluid retention, a point well appreciated by Reviewer 2.

The reviewer is certainly aware that many PPARs, despite being potent anti-inflammatory and insulin-sensitizing molecules, failed in the clinic due to adverse effects, most notably anemia and fluid retention (PMID: 17428730; PMID: 27636730). When used at clinical doses of 1.2 g per day (2.76 million nmol/day), Lanifibranor causes body weight gain, anemia, renal impairment (PMID: 34670042), which clearly discourages its use at clinical doses for the management of obesity and diabetes. To improve liver health, Lanifibranor requires preclinically doses of 30 mg/kg (68,981.4 nmol/kg) (PMID: 39805191), hence a dose 6,900-fold higher than the here used 10 nmol/kg.

As delineated above, GLP-1:GIP:Lani decreases at this dose body weight in the range previously reported for retatrutide while at the same time decreasing HFD-induced heart hypertrophy and improving cardiac performance without causing anemia, fluid retention or changes in urinary/plasma creatinine (**Fig. 3m-v**). Together with its ability to decrease marker for liver damage (indicated by lower AST and ALT, **Fig 3i,j**) and the robust effects to outperform GLP-1:GIP co-agonism and semaglutide to yield greater improvement in body weight and insulin sensitivity (**see Fig. 2a-m**), these findings strongly support its clinical development (which is currently ongoing). One could argue that the development of GLP-1:GIP:Lani is as much of an advance over the previously published GLP-1:Tesa molecule, as was the development of GLP-1:GIP:Glucagon triagonism relative to GLP-1:GIP co-agonism, both concepts notably first published by our group (PMID: 24174327, PMID: 25485909).

Scientific Rationale and Innovation

The manuscript does not clearly explain why targeting all five pathways in a single molecule is necessary or beneficial.

The importance of the unimolecular (and thus incretin receptor targeted) design is demonstrated in **Fig. 3a–m**, where we show that GLP-1:GIP:Lani yields markedly greater improvements in body weight and glycemic control compared to the physical co-therapy of GLP-1:GIP + Lani.

We also now provide more data to better understand the relevance of the different receptors. In **Fig. 4a,b**, we now show that weight loss induced by GLP-1:GIP:Lani is, in contrast to GLP-1R:GIPR co-agonism, strikingly impaired in mice with *Vglut-2* Cre-mediated loss of GLP-1R in glutamatergic neurons. Weight loss induced by GLP-1:GIP:Lani was further diminished in DIO mice with global lack of GIPR (**Fig. 4c**), confirming that GLP-1:GIP:Lani decreases body weight via both incretin receptors.

Consistent with that GLP-1R agonists activate POMC neurons via hypothalamic glutamatergic GLP-1R signaling (**PMID: 34626854**), we further now show using fiber-photometry that GLP-1:GIP:Lani more robustly increases POMC neuronal activity compared to GLP-1R:GIPR co-agonism in *POMC-Cre* mice, and this is confirmed also *ex vivo* using electrophysiology in *POMC-GFP* mice (**Fig. 5n-p**).

Collectively, these data show that GLP-1:GIP:Lani reduces body weight and food intake via both incretin receptors and that this is paralleled by more robust activation of POMC neuronal activity. Notably, weight loss by GLP-1:GIP is, in contrast to GLP-1:GIP:Lani, only mildly diminished in the *Vglut2-GLP-1R* KO mice (**see data above**), indicating that the delivery of Lani into the glutamatergic GLP-1R neurons is crucial for the metabolic effects of the drug. Consistent with a role of GIPR is also the observation that weight loss induced by GLP-1:GIP:Lani is identical to co-therapy of GLP-1:Lani with a long-acting (acyl) GIP (**Fig 4d**). However, despite yielding comparable weight loss, GLP-1:GIP:Lani outperforms the co-therapy to yield greater improvement of glucose tolerance (**Fig 4f**), indicating that enhanced weight loss by GLP-1:GIP:Lani originates from the delivery of Lani into *Vglut2* GLP-1R neurons, while the delivery of Lani into GIPR cells further improves glucose metabolism.

Confirming the contribution of PPAR δ for the glycemic effects of GLP-1:GIP:Lani, we now show that selective antagonization of PPAR δ using GSK3787 blocks the glucose lowering effect of GLP-1:GIP:Lani (**Fig 4g**), but interestingly without affecting the ability of the selective PPAR δ agonist GW501516 (or GLP-1:GIP:Lani) to decrease body weight (**Fig 4h**). Together, these data show that GLP-1:GIP:Lani decreases body weight via both incretin receptors, while further improving glucose metabolism via PPAR δ . Moreover, these data indicate that the PPAR δ agonist GW501516, and likely also the PPAR moiety of GLP-1:GIP:Lani, also decrease body weight via mechanisms unrelated to PPAR δ .

The importance of PPAR α and γ is most notably demonstrated by their ability to improve GLP-1 effects on glucose metabolism. Consistent with this, we already published that GLP-1:Tesa (Tesaglitazar is a PPAR α,γ co-agonist), despite having only modest effects on body weight, substantially enhances glucose metabolism compared to its pharmacokinetically-matched GLP-1R agonist backbone (**PMID: 35995995**). The contribution of PPAR α and γ hence lie in their additional glycemic effects, without major effects to further enhance weight loss.

Nonetheless, consistent with previous reports showing that all three PPAR isoforms improve glucose metabolism in obese rodents (**PMID: 27636730**), we show in our manuscript that GLP-1:GIP:Lani not only solidly outperforms GLP-1:GIP:Tesa to yield greater decreases in body weight and fat mass, but also to further decrease blood glucose (**EDF2f-h**). And these effects are now further corroborated by showing a direct role of PPAR δ for the blood glucose lowering effect of GLP-1:GIP:Lani (**Fig 4g**).

We now also provide more mechanistic data on how GLP-1:GIP:Lani improves glucose metabolism. Consistent with the main PPAR action, we show that GLP-1:GIP:Lani outperforms GLP-1:GIP to strongly upregulate anti-inflammatory gene programs in the liver and skeletal muscle (**Fig 2n,o**).

Consistent with this, we now show that GLP-1:GIP:Lani not only outperforms GLP-1:GIP co-agonism to further improve insulin sensitivity (**Fig 2g-i**), but also to further suppress endogenous glucose production (**Fig 2j,k**), and this is mechanistically linked to decreased hepatic expression of *pyruvate carboxylase (Pc)* and *phosphoenolpyruvate carboxykinase (Pepck)*, the two master regulator of gluconeogenesis (**Fig 2i,m**).

Notably, PPAR γ agonists not only act on mature adipocytes to improve insulin sensitivity, but also on adipocyte precursors to induce adipocyte differentiation and body weight gain (**PMID: 27636730**). The latter is of particular importance since GIPR is expressed in only mature adipocytes but not preadipocytes (**PMID: 21245029**) and is therefore assumed to restrict the action of GLP-1:GIP:Lani on mature adipocytes while restraining its action on preadipocytes. Consistent with this, we found GLP-1:GIP:Lani, in contrast to Rosiglitazone, unable to induce adipocyte differentiation (**Figure 2p,q**), but with greater glucose uptake into the brown adipose tissue (**Figure 2r**) and comparable glucose uptake into eWAT, skeletal muscle, liver and heart (**Figure 2s-v**). These data hence show that GLP-1:GIP:Lani protects from the adverse effects of PPARs to increase body weight via stimulation of adipocyte differentiation, while at the same time preserving its ability to act on mature adipocytes to increase adipose tissue glucose uptake.

These data are consistent with many reports showing that BAT glucose uptake is an indicator of insulin sensitivity and is not necessarily associated with changes in mitochondrial bioenergetics and energy expenditure (**PMID: 28082439**; **PMID: 30135070**, **PMID: 33546400**, **PMID: 20606075**, **PMID: 23221344**). Consistent with this, we find no changes in expression of key thermogenic genes in BAT following treatment with either GLP-1:GIP:Lani or GLP-1:GIP (**EDF5a-d**).

In line with these data, we now show that neither GLP-1:GIP nor GLP-1:GIP:Lani acutely or chronically affect whole-body energy expenditure (**EDF3j**) or mitochondrial function in differentiated BAT primary cells (**EDF5e-s**). Specifically, mitochondrial respiration, proton production rate, proton leak respiration, coupling efficiency, maximal substrate oxidation, total and non-mitochondrial respiration, ATP-linked respiration, glycolysis, and glycolytic ATP production all remain unchanged after treatment with GLP-1:GIP:Lani or GLP-1:GIP (**EDF5e-s**).

Collectively, and consistent with PPAR agonism, these data indicate that the superior glycemic benefits that were observed by GLP-1:GIP:Lani relative to GLP-1:GIP originate from its ability to further improve insulin sensitivity through decreased inflammation in key glucometabolic tissues, leading to enhanced suppression of hepatic glucose production and increased glucose uptake into key insulin sensitive glucometabolic tissues.

It's unclear how this design improves on existing drugs like tirzepatide or lanifibranor, or why approved agents used in combination wouldn't be sufficient.

In clinical Phase 2b studies, Lanifibranor was given at doses of 0.8–1.2 g/day (1.84 and 2.76 million nmol per day) and led at these doses to body weight gain, anemia, leukopenia, fluid retention (peripheral oedema) and diarrhoea (**PMID: 34670042**), which obviously discourages its combination at clinical doses with tirzepatide (a point well noticed by Reviewer 2). Preclinically, Lanifibranor requires doses of 30 mg/kg (68,981 nmol/kg) to improve liver health (**PMID: 39805191; 33278455**), which is 6,891-fold higher than the here used dose of 10 nmol/kg for GLP-1:GIP:Lani.

When administered at daily doses of 10–30 nmol/kg in DIO mice, tirzepatide-induced weight loss plateaus at –20% to –30% after 14–20 days of treatment (**PMID: 34003802; 30473097; 39612941; 37840407; 35809773; 35921984**). In contrast, we show here that GLP-1:GIP:Lani at a dose of 10 nmol/kg reduces body weight by –30% to –40% over the same period without reaching a plateau (**EDF4d**). This magnitude of weight loss clearly exceeds every published report on tirzepatide (**PMID: 34003802; 30473097; 39612941; 37840407; 35809773; 35921984**) and is comparable to that of the GLP-1:GIP:Glucagon triagonist retatrutide (**PMID: 35985340**), but without the cardiovascular and hyperglycaemic liabilities that reside in glucagon receptor agonism. In non-human primates, retatrutide increased heart rate by +12 bpm (**PMID: 35985340**), and by +5.6–7.5 bpm in humans (**PMID: 37366315; 4013943**). In our revised manuscript, we now show that GLP-1:GIP:Lani decreases heart rate and HFD-induced heart hypertrophy while increasing CV function, including ejection fraction, fractional shortening, stroke volume and cardiac output, notably without causing anemia, fluid retention or changes in urinary/plasma creatinine (**Fig. 3m-v**).

Moreover, we show in **EDF6-h** that treatment with GLP-1:GIP:Lani at a dose of 10 nmol/kg completely prevents body weight gain and development of obesity in leptin receptor deficient db/db mice. Importantly, and in contrast to GLP-1:GIP:Lani, previous studies have shown that tirzepatide, semaglutide and retatrutide are at the exact same dose all unable to prevent weight gain and development of obesity in db/db mice (<https://doi.org/10.2337/db25-2169-LB> and **PMID: 39344853**), which again supports the therapeutic advance of GLP-1:GIP:Lani over these existing therapies. It is very unfortunate that this reviewer missed to appreciate these data.

The novelty of the approach, both mechanistically and clinically, needs to be better defined.

We think to have adequately addressed this question in our answers above. In summary, the novelty of GLP-1:GIP:Lani lies in its metabolic effects, with superior (retatrutide-like) weight loss compared to GLP-1:GIP co-agonism and semaglutide, but with further improved cardiac performance without the adverse renal effects seen with Lanifibranor in clinical studies. As delineated in detail above, we now also provide more mechanistic details on how GLP-1:GIP:Lani affects energy and glucose metabolism. In brief, we now show that GLP-1:GIP:Lani decreases body weight via both incretin receptors, while further improving glucose metabolism via PPAR δ (please see our answers above for more details).

Safety considerations also need to be addressed more directly.

We now show that GLP-1:GIP:Lani decreases HFD-induced heart hypertrophy without causing anemia, fluid retention, or changes in plasma/urinary creatinine (**Fig 3m-p**). GLP-1:GIP:Lani does further not affect blood pressure, but decreases heart rate while increasing CV function, including ejection fraction, fractional shortening, stroke volume and cardiac output (**Fig. 3q-v**).

Experimental Design and Model Use

The use of multiple animal models is not well justified. Some models seem redundant, and it is not clear how each contributes uniquely to understanding the compound's effects. A more focused approach would help—fewer models, with each aligned to a specific mechanistic question, would strengthen the study.

It is unfortunate that the reviewer struggled to understand the rationale of the different animal models.

- 1) **Diet-induced obese (DIO) mice** are the most widely used model to study drug effects on energy and glucose metabolism.
- 2) **Leptin receptor deficient (db/db) mice** are the best model to study drug effects under extreme conditions of insulin resistance and under monogenetic obesity associated with loss of leptin action. The rationale for using this model was to assess whether GLP-1:GIP:Lani decreases body weight and hyperglycemia even under these extreme conditions (**EDF6a-h**). Notably, in contrast to GLP-1:GIP:Lani, previous studies have shown that tirzepatide, semaglutide and retatrutide are all unable to prevent weight gain and development of obesity in db/db mice when used at the exact same dose (<https://doi.org/10.2337/db25-2169-LB> and PMID: 39344853), which again supports the therapeutic advance of GLP-1:GIP:Lani over these existing therapies.
- 3) **DIO Double incretin receptor KO mice** were used to verify that GLP-1:GIP:Lani has no metabolic off-target effects in mice lacking both GLP-1R and GIPR.
- 4) **DIO adipose-specific GIPR overexpressing mice** were used based on the observation that the GIP moiety of the GLP-1:GIP:Lani conjugate significantly contributes to its body weight and glucose lowering effects (**Fig 4c,f**). The contribution of the GIP receptor is demonstrated by the greater metabolic effects of GLP-1:GIP:Lani compared to GLP-1:Lani (**Fig 1l-o**), and by demonstration that the metabolic effects of GLP-1:Lani are greatly enhanced when GLP-1:Lani is given together with acyl-GIP (**Fig 4d-f**). The contribution of GIPR is now further confirmed by diminished weight loss of GLP-1:GIP:Lani in DIO GIPR KO mice (**Fig 4c**). To decipher the role of GIPR agonism for the metabolic effects of the GLP-1:GIP:Lani conjugate, it was therefore reasonable to assume that GIP may act on the adipose GIPR to promote lipid utilization, as was previously shown by Regmi et al., *Cell Metabolism* 2024 (PMID: 38878772). Accordingly,

we here tested in adipose GIPR overexpressing mice whether GLP-1:GIP:Lani has greater effects on lipolysis in such mice with higher-than-normal adipose tissue GIPR expression.

- 5) **DIO glutamatergic GLP-1R KO mice** were used in the revised manuscript to decipher the contribution of glutamatergic neuronal GLP-1R signaling for the metabolic effects of GLP-1:GIP:Lani. GLP-1R agonists act via glutamatergic GLP-1R neurons in the hypothalamus to stimulate POMC neuronal activity (**PMID: 34626854**), and GLP-1R agonists fail to decrease body weight in glutamatergic GLP-1R KO mice (**PMID: 29776968**). Our new studies show that the body weight and food intake decreasing effects of GLP-1:GIP:Lani are strikingly diminished in glutamatergic GLP-1R KO mice (**Fig 4a,b**), and that GLP-1:GIP:Lani outperforms GLP-1:GIP co-agonism to activate POMC neuronal activity (**Fig 5n-p**), collectively indicating that GLP-1:GIP:Lani acts via glutamatergic GLP-1R neurons in the hypothalamus to enhance POMC neuronal activity.
- 6) **DIO GIPR KO mice** were used in the revised manuscript to assess the individual contribution of GIPR for the metabolic effects of GLP-1:GIP:Lani. These new studies show that weight loss induced by GLP-1:GIP:Lani is diminished in DIO GIPR KO mice (**Fig 4c**), indicating that weight loss induced by GLP-1:GIP:Lani originates from its action at both incretin receptors. This is also further confirmed by the demonstration that GLP-1:GIP:Lani decreases body weight with superiority to GLP-1:Lani (**Fig 1l-o**), and that weight loss induced by GLP-1:Lani is further enhanced by co-therapy with acyl-GIP (**Fig 4d-f**).

Clarity and Structure

The manuscript is difficult to follow due to a lack of narrative flow. The transitions between sections are abrupt, and data from complex models are not well integrated or discussed. Reorganizing the manuscript to better connect rationale, results, and conclusions would improve readability and impact.

The text has now been largely re-written to improve clarity, and we hope that the revised manuscript and this rebuttal letter will help to convey the main message of our work more clearly. The reviewers comment stands also in marked contrast to the feedback from the other reviewers. For instance, Reviewer 2 noted that *“the paper is well written, and the data are clearly presented,”* and Reviewer 3 stated that *“this manuscript provides a very thorough phenotyping of GLP-1:GIP:Lani’s impressive effects compared to many other drugs and conditions.”*

Recommendations for Improvement

1. Better explain the rationale for targeting all five pathways. What unique advantage does this provide?

We think we have adequately addressed this point our responses above. In summary, we now show that GLP-1:GIP:Lani exhibits retatrutide-like efficacy on body weight loss, but with cardiovascular benefits and without the classical adverse renal effects seen with Lanifibranor in clinical studies. Using various new mouse models, we now show that the body weight lowering effect of GLP-1:GIP:Lani is diminished in mice with genetic or pharmacological silencing of GIPR, GLP-1R, or PPAR δ .

2. Streamline the experimental plan and clearly justify each model.

The rationale for using the different models is explained in more detail in one of our answers above. Justification of the different models is now also discussed in more detail in the manuscript.

3. Improve the structure and flow so the data better support the conclusions.

The manuscript has now been restructured to better support the conclusions.

4. More explicitly discuss the clinical relevance—how this compares to existing treatments or combinations of approved drugs.

We believe to have have adequately addressed this point in our answers above. In summary, we show that GLP-1:GIP:Lani exhibits retatrutide-like efficacy on body weight loss, but with cardiovascular benefits and without the classical adverse renal effects seen with Lanifibranor in clinical studies. Using various new mouse models, we now show that the body weight lowering effect of GLP-1:GIP:Lani is diminished in mice with either genetic or pharmacological silencing of GIPR, GLP-1R, or PPAR δ . Further emphasizing the clinical relevance of our data, we show that GLP-1:GIP:Lani fully prevents weight gain and development of obesity in db/db mice, a model in which tirzepatide, semaglutide and retatrutide all failed to prevent weight gain and development of obesity when used at the exact same dose (<https://doi.org/10.2337/db25-2169-LB> and PMID: 39344853).

Specific Questions

Biodistribution: Is there direct evidence showing the biodistribution of the GLP-1:GIP:Lani molecule? The cited studies don't examine the full construct. Could this affect biodistribution or tissue targeting?

In **Fig 5a,b** we show comparable cFos neuronal activation by GLP-1:GIP:Lani and GLP-1:GIP in the brainstem and hypothalamus, indicating that both drugs equally reach their target receptors in the circumventricular organs. Liraglutide (PMID: 29985439), semaglutide (PMID: 32213703), acyl-GIP (PMID: 37946085), and GLP-1:GIP co-agonists (PMID: 40830598) all show no major ability to cross the blood-brain-barrier, and this is now also shown for the here used GLP-1R:GIPR co-agonist and GLP-1:GIP:Lani (**Fig 5g**). Our data are hence consistent with published reports on other incretin-based drugs and indicate that they primarily reach the circumventricular organs without crossing the BBB.

● Nafu ● GLP-1:GIP ● Sertraline
● GLP-1:GIP:Lani ○ Diazepam

Line 162, Extended Data Fig. 2c–e:

Do the cAMP results suggest greater potency at GLP-1R? EC₅₀ values should be calculated and reported.

EC₅₀ values are now mentioned (**Fig 1f,g**) and do not support a major difference in cAMP potency between GLP-1:GIP:Lani and GLP-1:GIP, which is also functionally verified by equal potentiation of glucose-stimulated insulin secretion in isolated murine islets (**Fig 1h**). Importantly, while incretin-induced cAMP potency is a good proxy for insulin secretion, it is not a predictor for incretin-induced weight loss efficacy (see **Fig 1C** in PMID: 40157531). Consistent with this, we recently reported that biochemical modification of semaglutide to yield lower cAMP potency (a molecule named NNC5840) increases body weight loss relative to the regular, and more cAMP potent, semaglutide (PMID: 40157531). Consistent with this, tirzepatide decreases body weight in DIO mice exclusively via GLP-1R (PMID: 34003802) but yields at equimolar comparison greater weight loss as semaglutide, which shows 20-fold greater cAMP potency than tirzepatide (EC₅₀ 0.36 nM vs. 6.54 nM) (PMID: 32730231).

In Fig. 2e, is the experiment under high or low glucose? The legend is unclear—please clarify or add the missing data.

We assume that the reviewer is referring here to **EDF2e (which is now Fig 1h)**. The shown data represent the so-called ‘insulin stimulation index’, which is defined as (drug-induced) fold-difference from low (2.8 mM) vs. high (20 mM) glucose. To briefly summarize the data, we here show that treatment with Vehicle induces a 1.58 fold (58%) increase in insulin secretion from low vs. high glucose, while treatment with GLP-1:GIP increases insulin secretion at doses of 1, 10 and 50 nM by 1,71-fold (71%), 2,32-fold (132%) and 3.27-fold (227%) and GLP-1:GIP:Lani by 1.64-fold (64.4%), 2.635-fold (163,5%) and 2.98-fold (198%) (**Fig 1h**).

Line 175 (DIR⁻ cells): Why does GLP-1:GIP:Lani show no activity in DIR⁻ cells? Could conjugation affect receptor binding? If uptake doesn’t depend on GLP-1R or GIPR, what explains the lack of effect?

As mentioned in the text and the figure legend, we are here referring to double-incretin receptor deficient (DIR⁻) cells. While GLP-1:GIP:Lani induces PPAR target gene expression in the presence of the incretin receptors (**Fig 1i,j**), it expectedly fails to do so in the absence of the incretin receptors (**Fig 1k**), confirming the incretin receptor-targeting nature of the molecule.

Line 179: The reported 3-fold greater weight loss seems overstated—please check the data and adjust.

Placebo-corrected weight loss is here -35.69% after treatment with GLP-1:GIP:Lani and -13.59% after treatment with GLP-1:Lani, so 2.63-fold greater. The text has been corrected accordingly.

Fig. 2o (Weight loss vs. GTT): Weight loss is similar between GLP-1:GIP:Lani and GLP-1:GIP:Tesa, but GTT outcomes differ. Could this reflect differences in β -cell function or other mechanisms?

We are confused about the reviewer’s comment. As mentioned in the text and the figure legend, Figure 2o (**now Fig 2f**) refers to an oral GTT for GLP-1:GIP:Lani. While we do not report oral glucose tolerance for GLP-1:GIP:Tesa, weight loss induced by GLP-1:GIP:Tesa is much lower, not similar, compared to GLP-1:GIP:Lani (**Fig. 1a and I**), with placebo-corrected weight loss of -19.97% vs. -35.73% ($p < 0.001$, **EDF 2f**).

Potentially, the reviewer rather wanted to point out the similarities in the ipGTT data reported for GLP-1:GIP:Tesa (**Fig. 1d**) and GLP-1:GIP:Lani (**Fig. 1o**), which indeed appear similar despite different degrees of weight loss (**Fig. 1a and I**). This point is important, since we show that great improvement of glucose tolerance is already achieved by GLP-1:GIP-mediated delivery of the PPAR α,γ co-agonist Tesaglitazar, while greater body weight loss requires incretin receptor-mediated delivery of the PPAR α,γ,δ triagonist Lanifibranor (**Fig 1a,d,o,I**). These data again support the value of targeting all five receptors for achieving the greatest metabolic outcome. While PPAR α,γ,δ triagonism is needed to maximize weight loss, PPAR α,γ co-agonism is already greatly improving glucose tolerance. The observation that glucose tolerance is similar between GLP-1:GIP:Tesa and GLP-1:GIP:Lani despite different degrees of weight loss is also not surprising, since GLP-1:GIP:Tesa already maximally improved glucose tolerance in this study, and additional weight loss induced by GLP-1:GIP:Lani cannot

further improve this already maximally improved glucose tolerance (you can't make glucose tolerance better than perfect; see data below) (Fig 1d,o).

Line 252 (db/db model): What's the rationale for using GLP-1:GIP:Lani in db/db mice? TG and cholesterol levels are unchanged despite weight loss. Could this be due to short treatment? How does this relate to elevated liver FFA/cholesterol (Line 302)?

As mentioned also above, leptin receptor deficient (db/db) mice are the best model to study drug effects under extreme conditions of insulin resistance and under monogenetic obesity associated with loss of leptin action. The rationale for using this model was to assess whether GLP-1:GIP:Lani even decreases body weight and hyperglycemia under these extreme conditions (Fig4a-h). In contrast to GLP-1:GIP:Lani, previous studies have shown that tirzepatide, semaglutide and retatrutide are all unable to prevent weight gain and development of obesity in db/db mice, which again supports the therapeutic advance of GLP-1:GIP:Lani over these existing therapies (<https://doi.org/10.2337/db25-2169-LB> and PMID: 39344853).

Related to the observed effects on lipids and cholesterol, it should be noted that db/db mice are extremely hyperphagic. In our study, the mice ate 7.5g/day, which is roughly 3-times more than what a regular DIO mouse eats per day. This extreme hyperphagia typically results in that anti-obesity drugs (including semaglutide, tirzepatide and retatrutide) work rather poor in this model and have very low to absent effects on glucose and lipid metabolism.

Line 330 (Adipose-specific GIPR): Again, what is the rationale for using this model? What data support the statement about greater weight loss in GIPR TG mice? Please provide or reference the results.

As mentioned also above, DIO adipose-specific GIPR overexpressing mice were used based on the observation that the GIP moiety of the GLP-1:GIP:Lani conjugate significantly contributes to its body weight and glucose lowering effect. The contribution of GIPR is demonstrated by greater metabolic effects of GLP-1:GIP:Lani compared to GLP-1:Lani (Fig 1l-o), and further by that the metabolic effects of GLP-1:Lani are greatly enhanced when GLP-1:Lani is given together with acyl-GIP (Fig. 4d-f). The contribution of GIPR is now further confirmed by diminished weight loss of GLP-1:GIP:Lani in DIO GIPR KO mice (Fig 4c). It was therefore reasonable to assume that GIP may act on the adipose GIPR to promote lipid utilization, as was previously shown by Regmi et al., *Cell Metabolism* 2024 (PMID: 38878772). Accordingly, we here tested in adipose GIPR overexpressing mice whether GLP-1:GIP:Lani has greater effects on lipolysis in such mice with higher-than-normal adipose tissue GIPR expression.

Related to question what data support the statement about greater weight loss in GIPR TG mice, we refer here Yu et al., *Cell Metabolism* 2025 (PMID: 39642881). The two senior authors of that paper (Scherer and Kusminski) are also co-authors of the current manuscript.

Line 378 (c-fos): If c-fos activation is similar, how does that relate to greater reduction in food intake with GLP-1:GIP:Lani versus GLP-1:GIP?

C-Fos is a marker for acute neuronal activation and only gives information as to whether a neuron is activated or not, but not on the signaling events within these activated neurons. Equal cFos induced by GLP-1:GIP and GLP-1:GIP:Lani only indicates that both drugs acutely turn on the same neurons, which is expected given that they both target the same (GIPR and GLP-1R expressing) neurons. But despite equal cFos activation, drug-induced gene programs and signaling can of course vary significantly in these neurons, and this is most impressively shown in **Fig 5j and k** where we show in the brainstem after single drug administration 296 differentially regulated proteins induced by GLP-1:GIP:Lani compared to only 40 by GLP-1:GIP. Equal cFos activation does not allow the assumption that food intake suppression has to be similar. Accordingly, we now show that GLP-1:GIP:Lani causes relative to GLP-1:GIP more robust induction of POMC neuronal activity (**Fig 5n-p**), which cannot be captured using cFos, since both drugs both turn on (induce cFos in) POMC neurons.

Line 486: GLP-1:GIP:Lani does not appear to improve liver fibrosis, which is a key benefit of Lanifibranor in Phase 3. This limits the value of combining the two and should be discussed more in the manuscript.

We urge the reviewer to be cautious here. Although GLP-1R and GIPR are not expressed in the liver, semaglutide and tirzepatide both improve liver fibrosis and MASH in clinical studies (**PMID: 40305708, PMID: 38856224**), and semaglutide is even already approved for the treatment of MASH. The ability of semaglutide and tirzepatide to improve liver fibrosis is notably not only attributed to their ability to decrease body weight and liver fat, but also to their ability to decrease systemic inflammation in peripheral tissues (**PMID: 40024571, PMID: 41212550, PMID: 40980163**). Given the here shown ability of GLP-1:GIP:Lani to outperform GLP-1:GIP co-agonism and semaglutide to further decrease body weight (**Fig 2a and EDF3d**) and hepatic inflammation (**Fig 2n**), it is also very likely that GLP-1:GIP:Lani will have positive effects on liver fibrosis. This is now also discussed in the manuscript.

Line 496 (Brain penetration): Is there any evidence that Lani crosses the blood–brain barrier? Have c-fos effects been tested at clinically relevant doses?

There is no information as to whether Lanifibranor crosses the BBB, but given its large molecular size, it is unlikely that it does. Previous studies have shown that liraglutide (**PMID: 29985439**), semaglutide (**PMID: 32213703**), acyl-GIP (**PMID: 37946085**), and GLP-1:GIP co-agonists (**PMID: 40830598**) are all unable cross the BBB, and this is now also shown for GLP-1:GIP:Lani and its GLP-1:GIP co-agonist backbone in our studies (**Fig 5g**). We are not aware of studies that assessed cFos effects of Lanifibranor at clinical doses, and we didn't test higher doses of Lanifibranor since it showed in clinical studies body weight gain, fluid retention, leukopenia and anemia (**PMID: 39824443**).

Referee #2:

This manuscript presents results from studies which continue a previously reported concept, ie the pharmacological development of unimolecular, multi-target pharmacological agents combining a peptide (eg a dual GIPR:GLP1R agonist) and a small molecule (ie the pan-PPAR agonist lanifibranor). The paper is well-written and the data are clearly presented.

We thank the reviewer for his/her acknowledgment of our work and think that this reviewer's comments and suggestions have significantly strengthened the manuscript.

However, the conceptual novelty of this study is very limited, and the mechanistic insights are lacking. Indeed, the clinical utility/superiority of a combined GLP1R:GIP agonist (ie tirzepatide) has been amply demonstrated. Moreover, the interest of combining a peptide agonist with a small molecule PPAR agonist has already been demonstrated by the authors in their previous study (Nat. Metab., 2022, 4, 1071-1083). Unfortunately, the added benefit of the PPARdelta activity is unclear (as this is the case also in clinical studies) and has not been addressed.

The advancement of GLP-1:GIP:Lani relative to the published GLP-1:Tesa lies in its markedly enhanced metabolic effects rather than the concept of peptide-mediated nuclear hormone delivery itself. At a dose of 50 nmol/kg, GLP-1:Tesa decreased body weight in DIO mice by only -10% over vehicle controls, while GLP-1:GIP:Lani achieved at the same dose weight loss of -35.73% without even reaching a plateau (**Fig. 1a and I**).

We now also provide a series of new data to demonstrate the relevance of the different receptors. In **Fig. 4a,b**, we now show that weight loss induced by GLP-1:GIP:Lani is, in contrast to GLP-1R:GIPR co-agonism, strikingly impaired in mice with *Vglut-2* Cre-mediated loss of GLP-1R in glutamatergic neurons. Weight loss induced by GLP-1:GIP:Lani was further diminished in DIO mice with global lack of GIPR (**Fig. 4c**), confirming that GLP-1:GIP:Lani decreases body weight via both incretin receptors.

Consistent with the ability of GLP-1R agonists to activate POMC neurons via hypothalamic glutamatergic GLP-1R (**PMID: 34626854**), we further now show that GLP-1:GIP:Lani more robustly increases POMC neuronal activity compared with GLP-1:GIP *in vivo* and *ex vivo* (**Fig. 5n-p**).

In summary, GLP-1:GIP:Lani reduces body weight and food intake via both incretin receptors and with more robust activation of POMC neuronal activity. Notably, weight loss induced by GLP-1:GIP is, in contrast to GLP-1:GIP:Lani, only mildly diminished in the Vglut2-GLP-1R KO mice, indicating that the delivery of Lani into the glutamatergic GLP-1R neurons is crucial for its metabolic effects. Consistent with the role of GIPR, weight loss induced by GLP-1:GIP:Lani is near identical to co-therapy of GLP-1:Lani with a long-acting (acyl) GIP (**Fig 4d**). However, despite comparable weight loss, GLP-1:GIP:Lani outperforms the co-therapy to yield greater improvement of glucose tolerance (**Fig 4f**), indicating that enhanced weight loss induced by GLP-1:GIP:Lani originates from the delivery of Lani into Vglut2 GLP-1R neurons, while the delivery of Lani into GIPR cells further improves glucose metabolism.

Confirming the contribution of PPAR δ for the glycemic effects of GLP-1:GIP:Lani, we now show that selective antagonization of PPAR δ using GSK3787 blocks the glucose lowering effect of GLP-1:GIP:Lani (**Fig 4g**), but interestingly without affecting the ability of the selective PPAR δ agonist GW501516 or GLP-1:GIP:Lani to decrease body weight (**Fig 4h**). Together, these data show that GLP-1:GIP:Lani decreases body weight via both incretin receptors, while further improving glucose metabolism via PPAR δ . However, these data also indicate that the PPAR δ agonist GW501516, and likely also the PPAR δ moiety of GLP-1:GIP:Lani, also decrease body weight via mechanisms unrelated to PPAR δ .

The importance of PPAR α and γ is most notably demonstrated by their ability to improve GLP-1 effects on glucose metabolism. Consistent with this, we already published that GLP-1:Tesa (Tesaglitazar is a PPAR α,γ co-agonist), despite having only modest effects on body weight, substantially enhances glucose metabolism compared to its pharmacokinetically-matched GLP-1 backbone (**PMID: 35995995, see data below**). The contribution of PPAR α and γ hence lie in their additional glycemic effects, without major ability to further drive weight loss.

Nonetheless, consistent with previous reports showing that all three PPAR isoforms improve glucose metabolism in obese rodents (**PMID: 27636730**), we show in our manuscript that GLP-1:GIP:Lani not only solidly outperforms GLP-1:GIP:Tesa to yield greater decreases in body weight and fat mass, but also to further decrease blood glucose (**EDF2f-h**). And these effects are now further corroborated by a direct role of PPAR δ for the blood glucose lowering effect of GLP-1:GIP:Lani (**Fig 4g**).

Altogether, the potential interest of this concept has been readily demonstrated in preclinical studies. Given the difficulties of translating such findings to the clinic, a POC clinical study would be of great interest now.

We agree that a POC study would be great. Unfortunately, the GIP system is, in contrast to the GLP-1 system, not well conserved between mice and humans (**PMID: 40024571**). Consequentially, the here used GLP:GIP:Lani is optimized for the mouse GIP receptor and requires structural optimization for studies in higher species. While drug optimization for human studies is ongoing, first data on monkeys and humans are unfortunately not available for the current manuscript.

Other points:

- Semaglutide and tirzepatide are reported to be efficacious in the treatment of MASH. Lanifibranor is in Phase 3 clinical development for MASLD, but the rationale for selecting this compound in the present study is unclear, since there have been no studies using MASH-inducing diets.

The rationale for developing GLP-1:GIP:Lani was to combine the insulin-sensitizing and anti-inflammatory effects of PPAR agonism with the body weight and blood glucose lowering effects of GLP-1R:GIPR co-agonism within a single, highly potent molecule. While GLP-1:GIP:Tesa outperformed GLP-1:GIP co-agonism to further improve glucose metabolism, it had only limited ability to further enhance weight loss. We then exchanged Tesa with the PPAR α,γ,δ triagonist Lanifibranor, since Lani was shown preclinically to ameliorate HFD-induced body weight gain (**PMID: 32360434**, **PMID: 29404476**). The unique unimolecular design was employed to enable low-dose delivery of Lani specifically to cells expressing the incretin receptors, thereby preserving its anti-inflammatory and insulin-sensitizing actions while minimizing its clinical liability to cause weight gain, anemia and fluid retention. We agree that studies on MASH would be interesting, since semaglutide and tirzepatide have already been shown to have beneficial effects in this regard (**PMID: 40305708**, **PMID: 38856224**). Given our new data showing that GLP-1:GIP:Lani not only solidly outperforms GLP-1:GIP co-agonism and semaglutide to further decrease body weight (**Fig 2a and EDF3d**), but also to upregulate anti-inflammatory gene programs in the liver and skeletal muscle (**Fig 2n,o; see data below**), these data collectively suggest that GLP:GIP:Lani will also have positive effects on liver fibrosis. However, given the number of studies required for the revision, we were unfortunately not able to assess this aspect in the current manuscript.

Moreover, the clinical lipid/glucose modifying effects of tesaglitazar have already been demonstrated and those of lanifibranor appear very similar, as for example recently reported in Barb et al. J.Hep.82,979-991. The CNS effect has also been shown with the tesaglitazar-GLP1 combination treatment. What is the added value of GIPR:PPARdelta agonism in the current study?

We now provide a series of mechanistic data that help to better understand the individual contribution of the different target receptors, including PPAR δ . In detail, we now show that weight loss induced by GLP-1:GIP:Lani is, in contrast to GLP-1R:GIPR co-agonism, strikingly impaired in mice with Vglut-2 Cre-mediated loss of GLP-1R in glutamatergic neurons (**Fig. 4a,b**). Weight loss induced by GLP-1:GIP:Lani was further solidly diminished in DIO mice with global lack of GIPR (**Fig. 4c**), confirming that GLP-1:GIP:Lani decreases body weight via both incretin receptors.

Consistent with the ability of GLP-1R agonists to activate POMC neurons via hypothalamic glutamatergic GLP-1R signaling (PMID: 34626854), we now show *in vivo* and *ex vivo* that GLP-1:GIP:Lani more robustly increases POMC neuronal activity compared with GLP-1:GIP (**Fig. 5n-p**).

In summary, GLP-1:GIP:Lani reduces body weight via both incretin receptors, and this effect is paralleled by more robust activation of POMC neuronal activity. It should also be noted that weight loss induced by GLP-1:GIP is, in contrast to GLP-1:GIP:Lani, only mildly diminished in the Vglut2-GLP-1R KO mice (**see data above**), indicating that the delivery of the PPAR agonist into the glutamatergic GLP-1R neurons is crucial for its metabolic effects. Consistent with a role of GIPR, weight loss induced by GLP-1:GIP:Lani is identical to co-therapy of GLP-1:Lani with a long-acting (acyl) GIP (**Fig 4d**). However, despite yielding comparable weight loss, GLP-1:GIP:Lani outperforms the co-therapy to yield greater improvement of glucose tolerance (**Fig 4f**), indicating that enhanced weight loss induced by GLP-1:GIP:Lani originates from the delivery of the PPAR into Vglut2 GLP-1R neurons, while its the delivery into GIPR cells further improves glucose metabolism.

Confirming the contribution of PPAR δ for the glycemic effects of GLP-1:GIP:Lani, we now show that selective antagonization of PPAR δ using GSK3787 blocks the glucose lowering effect of GLP-1:GIP:Lani (**Fig 4g**), but interestingly without affecting the ability of the selective PPAR δ agonist GW501516 or GLP-1:GIP:Lani to decrease body weight (**Fig 4h**). These data show that GLP-1:GIP:Lani decreases body weight via both incretin receptors, while offering additional glycemic benefits via PPAR δ . Moreover, these data indicate that the PPAR δ agonist GW501516 (and likely also the PPAR δ moiety of GLP-1:GIP:Lani) also decrease body weight via mechanisms unrelated to PPAR δ .

Corroborating a role of PPAR δ for the body weight and blood glucose lowering effects of GLP-1:GIP:Lani, we see GLP-1:GIP:Lani to outperform GLP-1:GIP:Tesa to further decrease body weight, fat mass and blood glucose (**EDF2f**).

We now also provide additional mechanistic insights into how GLP-1:GIP:Lani improves glucose metabolism. In detail, we now show that GLP-1:GIP:Lani outperforms GLP-1:GIP to strongly upregulate anti-inflammatory gene programs in the liver and skeletal muscle (**Fig 2n,o**).

Consistent with this, GLP-1:GIP:Lani not only outperforms GLP-1:GIP co-agonism to further improve insulin sensitivity (**Fig 2g-i**), but also to further suppress endogenous glucose production (**Fig 2j,k**), and this is mechanistically linked to decreased hepatic expression of *pyruvate carboxylase (Pc)* and *phosphoenolpyruvate carboxykinase (Pepck)*, the two master regulator of gluconeogenesis (**Fig 2i,m**).

Notably, PPAR γ agonists not only act on mature adipocytes to improve insulin sensitivity, but unfortunately also on adipocyte precursors to induce adipocyte differentiation and body weight gain (**PMID: 27636730**). The latter is of particular importance since GIPR is expressed in only mature adipocytes but not preadipocytes (**PMID: 21245029**). Consistent with this, we found GLP-1:GIP:Lani, in contrast to Rosiglitazone, unable to induce adipocyte differentiation (**Figure 2p,q**), but with greater glucose uptake into the brown adipose tissue (**Figure 2r**) and comparable glucose uptake into eWAT, skeletal muscle, liver and heart (**Figure 2s-v**). These data hence show that GLP-1:GIP:Lani protects from the adverse effects of PPARs to increase body weight via stimulation of adipocyte differentiation, while at the same time preserving its ability to act on mature adipocytes to increase adipose tissue glucose uptake.

These data are consistent with many reports showing that BAT glucose uptake is an indicator of insulin sensitivity and is not necessarily associated with changes in mitochondrial bioenergetics and energy expenditure (**PMID: 28082439**; **PMID: 30135070**, **PMID: 33546400**, **PMID: 20606075**, **PMID: 23221344**). Consistent with this, we find no changes in expression of key thermogenic genes in BAT following treatment with either GLP-1:GIP:Lani or GLP-1:GIP (**EDF5a-d**).

Further in line with these data, we now show that neither GLP-1:GIP nor GLP-1:GIP:Lani acutely or chronically effects whole-body energy expenditure (**EDF3j**) or mitochondrial function in differentiated BAT primary cells (**EDF5e-s**). Specifically, mitochondrial respiration, proton production rate, proton leak respiration, coupling efficiency, maximal substrate oxidation, total and non-mitochondrial respiration, ATP-linked respiration, glycolysis, and glycolytic ATP production all remain unchanged after treatment with GLP-1:GIP:Lani or GLP-1:GIP (**EDF5e-s**).

Collectively, and consistent with the main effects of PPAR agonism, these data indicate that the superior glycaemic benefits that were observed by GLP-1:GIP:Lani relative to GLP-1:GIP originate from its ability to further improve insulin sensitivity through decreased inflammation in key glucometabolic tissues, leading to enhanced suppression of hepatic glucose production and increased glucose uptake into key insulin sensitive glucometabolic tissues.

- Figure 2h: PPAR γ agonism rather increases food intake related to the down-regulation of leptin in adipose tissue (which may contribute to the observed effect in db/db mice in figure 4 and the higher food intake in figure 6d). What is the mechanism of the decrease in food intake by GLP1:GIP:Lani?

In **EDF6a-h (previously Fig 4)** we show that GLP-1:GIP:Lani decreases body weight and food intake in leptin receptor deficient db/db mice, hence excluding a role of leptin in the here reported suppression of food intake. However, we now show that food intake suppression by GLP-1:GIP:Lani is strikingly decreased in DIO mice with loss of GLP-1R in glutamatergic GLP-1R neurons (**Fig 4a,b**) and that GLP-1:GIP:Lani more solidly increases POMC neuronal activity relative to GLP-1:GIP co-agonism (**Fig 5n-p**). Since GLP-1R agonists increase POMC activity via glutamatergic GLP-1R neurons (**PMID: 34626854**), these data suggest that GLP-1:GIP:Lani decreases food intake via potentiation of GLP-1R-mediated induction of POMC neuronal activity. Furthermore, we now show that weight loss and food intake suppression induced by GLP-1:GIP:Lani is also diminished in DIO GIPR KO mice **Fig 4c**, collectively indicating that GLP-1:GIP:Lani decreases food intake via both incretin receptors.

Notably, PPAR γ agonists not only act on mature adipocytes to improve insulin sensitivity, but also on adipocyte precursors to induce adipocyte differentiation and body weight gain (PMID: 27636730). The latter is of particular importance since GIPR is expressed in only mature adipocytes but not preadipocytes (PMID: 21245029) and is therefore assumed to restrict the action of GLP-1:GIP:Lani on mature adipocytes while restraining its action on preadipocytes. Consistent with this, we found GLP-1:GIP:Lani, in contrast to Rosiglitazone, unable to induce adipocyte differentiation (Figure 2p,q), but with greater glucose uptake into the brown adipose tissue (Figure 2r) and comparable glucose uptake into eWAT, skeletal muscle, liver and heart (Figure 2s-v). Failure of GLP-1:GIP:Lani to act on preadipocytes hence protects from the body weight increasing effect of PPAR γ agonism.

- Figure 4: why doesn't GLP1:GIP decrease blood glucose in db/db mice? Why doesn't GLP1:GIP:Lani increase liver FFA given that PPAR γ activation rather decreases FFA fluxes from adipose tissue.

It should be noted that db/db mice are extremely hyperphagic. In our study, the mice ate 7.5g/day, which is roughly 3-times more than what a regular DIO mouse eats per day. This extreme hyperphagia typically results in extreme insulin resistance with the consequence of that that anti-obesity and anti-diabetes drugs work rather poor in this model and have very low to absent effects on glucose and lipid metabolism. The mentioned absence of glycemic effects with GLP-1:GIP co-agonism is in striking contrast to the profound effects of GLP-1:GIP:Lani (EDF6e), which again emphasizes the therapeutic value of this drug. It should also be noted that GLP-1:GIP:Lani completely prevented the development of obesity in db/db mice (EDF6a), a model in which tirzepatide, semaglutide and retratrutide have all been proven unable to prevent the development of obesity in db/db mice when used at the exact same doses (<https://doi.org/10.2337/db25-2169-LB> and PMID: 39344853). This again emphasizes the therapeutic advance of GLP-1:GIP:Lani over these existing therapies.

- Mechanistic studies addressing tissue-specific effects of activation of the PPAR subtypes, admittedly very difficult, are lacking. Why is GLP1R:GIPR:Lani more efficient than the GLP1R:Lani + acylGIP combination on glucose (Figure 5C). Is this related to the tissue-distribution of lanifibranor? If so, which tissues/PPARs are involved?

While fully agreeing with the reviewer, mice with tissue-selective loss of the individual PPAR isoforms are not available for us. The demonstration that GLP1R:GIPR:Lani improves glucose control with superiority to GLP1R:Lani + acylGIP suggests that that these additional glycemic effects originate from the delivery of Lani into GIPR cells. Consistent with this, we now show that selective antagonization of PPAR δ using GSK3787 blocks the glucose lowering effect of GLP-1:GIP:Lani (Fig 4g), but interestingly without affecting the ability of the selective PPAR δ agonist GW501516 (or GLP-1:GIP:Lani) to decrease body weight (Fig 4h). While these data show that GLP-1:GIP:Lani decreases blood glucose via PPAR δ , they also indicate that the PPAR δ agonist GW501516 (and likely also the PPAR δ moiety of GLP-1:GIP:Lani) also decrease body weight via mechanisms unrelated to PPAR δ .

The importance of PPAR α and γ is most notably demonstrated by their ability to improve GLP-1 effects on glucose metabolism. Consistent with this, we already showed previously that GLP-1:Tesa (Tesaglitazar is a PPAR α,γ co-agonist), despite having only modest effects on body weight, solidly enhances glucose metabolism over its pharmacokinetically-matched GLP-1 backbone (**PMID: 35995995, see data below**). Accordingly, the contribution of PPAR α and γ lie in their additional glycemic effects, without overt ability to drive greater weight loss.

Nonetheless, consistent with previous reports showing that all three PPAR isoforms improve glucose metabolism in obese rodents (**PMID: 27636730**), we show in our manuscript that GLP-1:GIP:Lani not only solidly outperforms GLP-1:GIP:Tesa to yield greater decreases in body weight and fat mass, but also to further decrease blood glucose (**EDF2f-h**). And these effects are now further corroborated by showing a direct role of PPAR δ for the blood glucose lowering effect of GLP-1:GIP:Lani (**Fig 4g**).

- The well-known adverse effects of PPAR γ agonism (body weight increase, fluid retention and bone fractures) are likely also observed with lanifibranor in the clinic. It would be a real advance to see whether these effects are no longer observed with GLP1:GIP:Lani.

The reviewer is spot on. Lanifibranor is used clinically at doses of 1.2g/day (2.76 million nmol/day) and causes at this dose body weight gain, anemia and fluid retention (**PMID: 34670042**). In our revised manuscript, we now show that treatment with GLP-1:GIP:Lani decreases HFD-induced heart hypertrophy without causing anemia, fluid retention or changes in urinary/plasma creatinine (**Fig 3m-p**). GLP-1:GIP:Lani further not only decreases liver inflammation (**Fig 2n,o**) and marker for liver damage (AST, ALT) (**Fig. 3i,j**), but also decreases heart rate while increasing ejection fraction, fractional shortening, stroke volume and cardiac output, without affecting blood pressure (**Fig 3q-v**).

- Figure 2 and others: it is not a surprise that lanifibranor at 10 nmol/kg/day is inefficacious. A proper control would be the murine-equivalent clinically used dose of lanifibranor and compare to this dose (also for side effects).

We agree with the reviewer that the here used dose of Lanifibranor (10 nmol/kg) is not expected to show adverse metabolic effects, which again underscores the value of GLP-1:GIP:Lani. For our pharmacological studies, it was however essential to compare drug effects at equimolar doses, hence at the here used dose of 10 nmol/kg. While it would be interesting to assess adverse effects of Lanifibranor at murine-equivalent clinically used dose (30 mg/kg; 68,000 nmol/kg), we were for animal ethics reasons not able to do this. Since we already showed that GLP-1:GIP:Lani does not cause side effects at the here used dose of 10 nmol/kg, our local authorities didn't allow to study a dose of Lanifibranor that would pharmacologically be incomparable to GLP-1:GIP:Lani.

- Figure 7: via which PPAR? Difference with tesaglitazar?

We appreciate the reviewer's comment. As also mentioned above, mice with conditional loss of the individual PPAR isoforms are not available for us. And we must note that the here reported studies in DIO mice require after cohort generation HFD-feeding for 25 wks prior of starting the pharmacological studies and the subsequent proteomic analysis. It would take nearly two years to provide such data, which was unfortunately not feasible for the current manuscript.

Minor points:

- M&M, line 591 : single or double-housed mice: not a classical way of animal experimentation. Please clarify.

Mice were typically double housed, unless animal welfare reasons (e.g. due to fighting) demanded separation of individual cages. Mice were also single housed during assessment of energy expenditure in the metabolic chambers.

- lines 272, lines 285-287 + extended fig6: taste avoidance experiment is performed in lean mice. How can one be sure that the decreased body weight under other experimental conditions is not due to nausea? Avoidance, or preference, could be different in DIO mice.

Conditioned taste avoidance (CTA) is routinely measured in lean mice (**PMID: 15459118, PMID: 13129831, PMID: 12954420, PMID: 12704398, PMID: 12084531, PMID: 22227019**), and underlining its validity, common aversive agents such as GLP-1R agonists clearly induce CTA in lean mice, as shown also in our studies (**PMID: 12451146, PMID: 15459118, PMID: 9124501**). In our revised manuscript, we now show that the observed drug-induced CTA vanishes in GLP-1R KO mice (**EDF6i**), indicating that we see here normal GLP-1-induced nausea. The shown CTA is hence consistent with those of other GLP-1-based drugs. Furthermore, for most patients on GLP-1R agonists, GI-adverse events become much less frequent with stable dosing while weight loss continues (**PMID: 40353578, PMID: 35658024, PMID: 39536238**), indicating that the weight loss is not secondary to nausea.

- Figure 3n, Figure 4p, q, r: liver weight data should be expressed as % of body weight (as body weight changes with treatment).

We appreciate the reviewer's comment but must note that this approach can lead to erroneous conclusions. As an example, we show in **Fig 3h** that GLP-1:GIP and GLP-1:GIP:Lani decrease liver mass in DIO mice and that this correlates with decreased hepatic lipid levels and plasma levels of AST and ALT (**Fig 3i-k**). While these data argue for improved liver health, which is further now supported by increased hepatic expression of anti-inflammatory gene programs (**Fig 2n**), expressing liver mass as % body weight will now erroneously show that treatment with GLP:GIP and GLP-1:GIP:Lani both have no effect on liver mass over vehicle (**see data below**). Such correction hence only states that liver mass is normal based in the given body weight but erroneously underemphasizes the real hepatic benefits of the drug. We therefore suggest to not correct the data by body mass.

- Fig 6 and line 355: why was glucagon measured in this experiment and not in others?

We also measured glucagon levels in other studies, but since the results were similar to the reported data, we did not show them in the other figures. Notably, we now removed glucagon values from the manuscript due to restrictions in Figure size and the need to show only the most essential data.

- M&M, line 593: subcutaneous administration of the compounds: volume?

All drugs were injected subcutaneously at a volume of 5 μ l/g body weight, with exception of the new data using the PPAR δ antagonist GSK3787 and the PPAR δ agonist GW501516, which were both given intraperitoneally at a volume of 5 μ l/g body weight. This is now also stated in the Methods section and each figure legend.

- M&M, line 594: is vehicle composition identical for all compounds?

We routinely used PBS for all drugs in the manuscript. For the new data on the PPAR δ antagonist GSK3787 (**Fig 4g,h**), we however needed a final concentration of 5% DMSO in the solution, which was of course then used also for all other groups in this particular study.

- Figure 3m, lines 253-254: why was spleen weight measured?

We often observe that drug-induced toxicity correlates with increased spleen mass, so we considered this a valuable safety measure. However, we now removed these data from the manuscript due to restrictions in figure size and the need to only report the most relevant data.

Referee #3 (Remarks to the Author):

The manuscript by Liskiewicz et al designed a unimolecular quintuple agonist (GLP-1:GIP:Lani) that has similar effects in stimulating insulin *in vitro*, but far greater effects on food intake and body weight reduction and glucose regulation *in vivo* in diet-induced and genetic obesity. This is thought to be mediated by effects on gene programs in key tissues including brainstem. While this manuscript provides a very thorough phenotyping of GLP-1:GIP:Lani's impressive effects compared to many other drugs and conditions (GLP-1:GIP:Tesa, GLP-1:GIP, GLP-1:GIP + Lani, Sema, GLP-1:GIP pair fed, etc.), the lack of causal mechanistic insight into the potentiated effects of GLP-1:GIP:Lani diminish enthusiasm for the current manuscript. Nonetheless, this work as is would be appropriate for publication in a more specialized journal.

We thank the reviewer for his/her fair and constructive comments. In the revised manuscript, we now performed a series of studies to better understand the relevance of the individual receptors. In **Fig. 4a,b**, we now show that weight loss induced by GLP-1:GIP:Lani is, in contrast to GLP-1R:GIPR co-agonism, strikingly impaired in mice with *Vglut-2* Cre-mediated loss of GLP-1R in glutamatergic neurons. Weight loss induced by GLP-1:GIP:Lani was further diminished in DIO mice with global lack of GIPR (**Fig. 4c**), confirming that GLP-1:GIP:Lani decreases body weight via both incretin receptors.

Consistent with the ability of GLP-1R agonists to activate POMC neurons via hypothalamic glutamatergic GLP-1R signaling (PMID: 34626854), we further now show *in vivo* and *ex vivo* that GLP-1:GIP:Lani more robustly increases POMC neuronal activity compared with GLP-1:GIP (**Fig. 4n-p**).

Collectively, these data show that GLP-1:GIP:Lani reduces body weight and food intake via both incretin receptors and that this is paralleled by more robust activation of POMC neuronal activity. It should also be noted that weight loss induced by GLP-1:GIP is, in contrast to GLP-1:GIP:Lani, only mildly diminished in the *Vglut2-GLP-1R* KO mice (**see data above**), indicating that the delivery of Lani into the glutamatergic GLP-1R neurons is crucial for its metabolic effects. Consistent with a role of GIPR, weight loss induced by GLP-1:GIP:Lani is near identical to co-therapy of GLP-1:Lani with a long-acting (acyl) GIP (**Fig 4d**).

However, despite yielding comparable weight loss, GLP-1:GIP:Lani outperforms the co-therapy to yield greater improvement of glucose tolerance (**Fig 4f**), indicating that enhanced weight loss induced by GLP-1:GIP:Lani originates from the delivery of Lani into Vglut2 GLP-1R neurons, while the delivery of Lani into GIPR cells further improves glucose metabolism.

Confirming the contribution of PPAR δ for the glycemic effects of GLP-1:GIP:Lani, we now show that selective antagonization of PPAR δ using GSK3787 blocks the glucose lowering effect of GLP-1:GIP:Lani (**Fig 4g**), but interestingly without affecting the ability of the selective PPAR δ agonist GW501516 or GLP-1:GIP:Lani to decrease body weight (**Fig 4h**). Together, these data show that GLP-1:GIP:Lani decreases body weight via both incretin receptors, while offering additional glycemic benefits via PPAR δ . However, these data also indicate that the PPAR δ agonist GW501516, and likely also the PPAR δ moiety of GLP-1:GIP:Lani, also decrease body weight via mechanisms unrelated to PPAR δ .

The importance of PPAR α and γ is most notably demonstrated by their ability to improve GLP-1 effects on glucose metabolism. Consistent with this, we already published that GLP-1:Tesa (Tesaglitazar is a PPAR α,γ co-agonist), despite having only modest effects on body weight, enhances glucose metabolism compared to its pharmacokinetically-matched GLP-1 backbone (**PMID: 35995995**). Accordingly, the contribution of PPAR α and γ lie in their additional glycemic effects, without major ability to further drive weight loss.

Nonetheless, consistent with previous reports showing that all three PPAR isoforms improve glucose metabolism in obese rodents (**PMID: 27636730**), we show in our manuscript that GLP-1:GIP:Lani not only solidly outperforms GLP-1:GIP:Tesa to yield greater decreases in body weight and fat mass, but also to further decrease blood glucose (**EDF2f-h**). And these effects are now further corroborated by showing a direct role of PPAR δ for the blood glucose lowering effect of GLP-1:GIP:Lani (see data above and **Fig 4g**).

We now also provide additional mechanistic insights into how GLP-1:GIP:Lani improves glucose metabolism. In detail, consistent with the main PPAR action, we now show that GLP-1:GIP:Lani outperforms GLP-1:GIP to strongly upregulate anti-inflammatory gene programs in the liver and skeletal muscle (**Fig 2n,o**).

Consistent with this, GLP-1:GIP:Lani not only outperforms GLP-1:GIP co-agonism to further improve insulin sensitivity (**Fig 2g-i**), but also to further suppress endogenous glucose production (**Fig 2j,k**), and this is mechanistically linked to decreased hepatic expression of *pyruvate carboxylase* (*Pc*) and *phosphoenolpyruvate carboxykinase* (*Pepck*), the two master regulator of gluconeogenesis (**Fig 2i,m**).

Notably, PPAR γ agonists not only act on mature adipocytes to improve insulin sensitivity, but also on adipocyte precursors to induce adipocyte differentiation and body weight gain (PMID: 27636730). The latter is of particular importance since GIPR is expressed in only mature adipocytes but not preadipocytes (PMID: 21245029) and is therefore assumed to restrict the action of GLP-1:GIP:Lani on mature adipocytes while preventing its adverse action on preadipocytes. Consistent with this, we found GLP-1:GIP:Lani, in contrast to Rosiglitazone, unable to induce adipocyte differentiation (Figure 2p,q), but with greater glucose uptake into the brown adipose tissue (Figure 2r) and comparable glucose uptake into eWAT, skeletal muscle, liver and heart (Figure 2s-v). These data hence show that GLP-1:GIP:Lani protects from the adverse effects of PPARs to increase body weight via stimulation of adipocyte differentiation, while at the same time preserving its ability to act on mature adipocytes to increase adipose tissue glucose uptake.

These data are consistent with many reports showing that BAT glucose uptake is an indicator of insulin sensitivity and is not necessarily associated with changes in mitochondrial bioenergetics and energy expenditure (PMID: 28082439; PMID: 30135070, PMID: 33546400, PMID: 20606075, PMID: 23221344). Consistent with this, we find no changes in expression of key thermogenic genes in BAT following treatment with either GLP-1:GIP:Lani or GLP-1:GIP (EDF5a-d). In line with these data, we now show that neither GLP-1:GIP nor GLP-1:GIP:Lani acutely or chronically affect whole-body energy expenditure (EDF3j) or mitochondrial function in differentiated BAT primary cells (EDF5e-s). Specifically, mitochondrial respiration, proton production rate, proton leak respiration, coupling efficiency, maximal substrate oxidation, total and non-mitochondrial respiration, ATP-linked respiration, glycolysis, and glycolytic ATP production all remain unchanged after treatment with GLP-1:GIP:Lani or GLP-1:GIP (EDF5e-s).

Collectively, and consistent with the main effects of PPAR agonism, these data indicate that the superior glycaemic benefits that were observed by GLP-1:GIP:Lani relative to GLP-1:GIP originate from its ability to further improve insulin sensitivity through decreased inflammation in key glucometabolic tissues, leading to enhanced suppression of hepatic glucose production and increased glucose uptake into key insulin sensitive glucometabolic tissues.

Specific comments:

- Information about the GLP-1:GIP agonists used to create the quintuple agonist should be provided in the results and methods sections.

The used GLP-1:GIP co-agonist backbone is the previously published MAR709 (PMID: 33571454, PMID: 40301583, PMID: 38492844, PMID: 37946085, PMID: 37592302, PMID: 24174327). The decision to use MAR709 rather than tirzepatide is based on the observation that MAR709 is strikingly more potent at the mouse GIP receptor relative to tirzepatide (TEXT REDACTED PMID: 37277609).

TABLE REDACTED

- Figures 1-6 and associated supplements are very important data but all come to the same conclusion – that GLP-1:GIP:Lani is a superior drug for weight and food intake reduction and glycemic control. In this reviewer's opinion, these main figures could be condensed to display only key data in main figures: the comparison of either GLP-1:GIP to GLP-1:GIP:Lani or GLP-1:GIP:Lani to GLP-1:GIP + Lani co-treatment. Further, the in vitro data demonstrating effects in GIP or GLP-1 expressing cells but not in incretin KO cells is key and should be in the main figure. All other data could be supplemental.

We fully agree with the reviewer and now changed the manuscript accordingly.

- The exploration of potential side effects of this novel agonist is minimal compared to the thorough exploration of its effects on body weight and blood glucose regulation compared to nearly all possible comparison groups. For example, functional tests (beyond histology) on cardiovascular and renal function would be warranted, especially given known effects of PPAR agonists on these systems in clinical studies.

We are fully aligned with the reviewer. In our revised manuscript, we now show that GLP-1:GIP:Lani decreases HFD-induced heart hypertrophy and heart rate and while increasing CV function, including ejection fraction, fractional shortening, stroke volume and cardiac output, without causing anemia, fluid retention or changes in urinary/plasma creatinine (Fig. 3m-v).

- The graphs depicting cumulative food intake are informative but may diminish important information. For example, it looks like food intake may be affected in short term but may normalize (equal slopes) after chronic intake? Therefore, daily food intake would also be informative here.

We thank the reviewer for this comment. Like previously shown for semaglutide, tirzepatide and retatrutide, we also see in our studies that food intake suppression is most pronounced within the first 7-14 days of the study to then slowly diminish. While this effect is well known for all food intake decreasing drugs, it must be noted that food intake suppression is never, at any point of the study, equal to the vehicle controls (so the slopes are never identical). In the longest study reported in our manuscript (**Fig 3a**), we treated DIO mice for more than 30 days and observed throughout the study that daily food intake (which was measured per cage) is decreased in mice treated with GLP-1:GIP:Lani compared to Vhcl controls (**see also bar graph below**).

- Related, the authors suggest in Figure 6 that the effects on body weight may be independent of effects on food intake. How do the authors reconcile this with their data showing no significant effects on energy expenditure?

This is an excellent question. In this study, we show that mice that are weight-matched to GLP-1:GIP:Lani require from day 8 onwards slightly lower food to match the body weight of the GLP-1:GIP:Lani treated mice. While these data intuitively suggest that non-food intake related effects on body weight kick-in over time, it must be noted that the slope of food intake in the GLP-1:GIP:Lani treated mice all of sudden started to increase after day 8 onwards (which is rather unusual).

Unfortunately, mice often tend to shred small amounts of food, and this often occurs particularly towards the end of a study, and in groups that experienced a lot of stress due to the weight loss. In this particular case, the GLP-1:GIP:Lani treated mice suddenly increased their daily food intake from day 8 onwards from ~0.7g/day to now ~1.2 g/day, whereas the body weight-matched mice continued to receive daily ~0.7 g/day. Measuring of food intake in this low range of differences (0.5g) is challenging, and we can't rule out the possibility that we see here a certain degree of micro-shredding that unfortunately remained unnoticed. In any case, our new data further support that GLP-1:GIP:Lani does not affect energy expenditure or mitochondrial function (**EDF3j and EDF51-s**).

- It appears that most if not all studies were performed in male mice, however, it would be important to determine if there are potential sex differences in the ability for the quintuple agonist to reduce body weight and improve glycemia.

While principally agreeing with the Reviewer, female mice do even after HFD exposure for >8 months not get adequately obese and glucose intolerant to study drug effects on energy and glucose metabolism (**PMID: 33001570, PMID: 28462078**). Since the degree of obesity was very important for our studies, we couldn't use female mice for our studies. Nonetheless, we previously tested the body weight lowering effect of GLP-1:GIP co-agonism and GLP-1:GIP:Glucagon triagonism in female mice kept on HFD for >8 months and observed no major difference in weight loss and improvement in glucose control compared to male DIO mice (**PMID: 33001570, PMID: 28462078**), which also aligns with studies in humans showing comparable weight loss in males and females after treatment with semaglutide and tirzepatide (**PMID: 40353578, PMID: 39536238, PMID: 35658024**). In conclusion, there is no reason to assume that the here reported effects are sex specific.

- All of the studies were performed with 10 nmol/kg doses of drugs. Why was this dose switched to 50 nmol/kg for the proteomics experiments? This is a potential cofound when using these results as potential explanations for drug effects.

We are also reporting drug effects on body weight and glucose control after treatment with 50 nmol/kg (**see Fig 1a-u**), so the dose used for the proteomic study was in the range also used for some of the *in vivo* studies. The reason for choosing a dose of 50 nmol/kg for the proteomic studies was to optimize data quality by increasing the signal-to-noise ratio, which is common for such studies and not trivial given that this study cost around 100,000€. For the same reason, higher doses are typically also used when assessing drug effects using RNAseq, cFos or brain imaging (**PMID: 29985439, PMID: 32213703, PMID: 37946085, PMID: 40830598**).

- While the authors used RNA sequencing and proteomics as an attempt to determine mechanisms for the synergistic effects of the multiagonist, all of this is observational with no causal mechanisms.

We can't agree more with the Reviewer. RNAseq and proteomic studies do just give a footprint of the actual situation in the tissue of interest, but they are quite descriptive with limited mechanistic insights. But as also explained in one of our answers above, we now provide a series of new mechanistic data that help to better understand how GLP-1:GIP:Lani affects energy and glucose metabolism (for details please see one of our answers above).

Rebuttal letter

We ask that you add the appropriate caveats to the discussion discussing the gaps in mechanistic understanding about how/where this drug is specifically acting and future directions in terms of mechanistic molecular understanding and human studies.

We now expanded the Discussion accordingly.

In addition to the general formatting requests in this email, we ask that you address the following formatting points below:

1. Please reduce the length of the title to 75 characters (with spaces) or less, so that it fits on two lines in the final layout.

The title has been shortened accordingly.

2. Please add references to the abstract (if applicable).

The abstract has been changed accordingly.

3. Please create a separate reference list for any methods references, making sure that the numbering continues from the main text references.

The methods have now a separate reference list with consecutive numbering. Note that most methods are in Supplementary Information 1, also with consecutive numbering from the main text. We are aware that these references will not account for PubMed citations, but it seems the best way forward.

4. Please reduce subheadings to 40 characters (with spaces) or less.

The subheadings have been shortened accordingly.

5. You have more than one account on EJP. Please contact Nature Manuscripts. (naturemanuscripts@nature.com) to confirm all accounts which belong to you, and your primary email address.

All EJP accounts have been merged.

6. Please provide a supplementary information guide as a separate word document.

Supplementary Information guide is now provided.

7. Please make sure to provide a third party rights table (more information below) when you resubmit. If Biorender or similar software has been used, please also ensure to provide relevant licenses.

We are now providing a third party rights table, but it only states that there are no third party rights affected.

8. Flagging that there are display items in the SI - please check if any of these materials should be extended data and notify us if they require further reproducibility checks.

The provided figures in supplementary figure 1 are the original files corresponding to Figure 1p, Figure 3l and Figure 5a,b. Since the main figures only contain representative examples out of several pictures per group, the supplementary figure 1 shows all samples from which the representative pictures for the main figures were taken from. There is no need to make the figures in supplementary figure 1 an extended data figure (we just follow the full transparency guideline here by showing all data).

9. Please ensure that the text size in all figures is at least 5 pt Arial.

The figures should now match these requirements. Only EDF8 needed an update here. It was a bit tricky for EDF8d,e,f,g. If the font size is still too low, then we may need to make a new and additional extended data figure out of EDF8d,e,f,g.

TRANSPARENT PEER REVIEW: Nature uses a transparent peer review system for original research manuscripts. We encourage increased transparency in peer review by publishing the reviewer comments, author rebuttal letters and editorial decision letters. Such peer review material is made available as a supplementary peer review file.

That's fine for us.

Please note: we allow redactions to authors' rebuttal and reviewer comments in the interest of confidentiality. If you are concerned about the release of confidential data, please let us know specifically what information you would like to have removed. Please note that we cannot incorporate redactions for any other reasons. Reviewer

names will be published in the peer review files if the reviewer signed the comments to authors, or if reviewers explicitly agree to release their name. For more information, please refer to our FAQ page.

There are no confidential information in the rebuttal letter

ORCID--IMPORTANT: All authors identified as 'corresponding author' on the manuscript must have an ORCID associated with their Nature account before submitting the final version of the manuscript.

My ORCID is associated with the manuscript.

In order to avoid delays with publication of your manuscript, please read the guidelines below carefully before resubmission of your manuscript.

STATISTICS: When revising your manuscript, you should ensure that any statistical analysis used is sound and that it conforms to our guidelines. A collection of articles explaining the basics of statistical analysis and advice on how to best present it can be found here.

Statistics are correctly reported. Please note that we provide a full statistical summary with the used tests and the individual p-values for each comparison in both the Supplementary Table 1 and the Data Source file. Given the amount of data and group comparisons, it was not possible giving the individual p-values within the figure legends while staying in the 300 word limit.

REPRODUCIBILITY: To ensure that the quality and transparency of methods and statistical reporting (as discussed here) are sound before the paper is published, we have reviewed your Reporting summary editorially. I have attached two documents: one listing specific issues related to your manuscript and one containing an annotated version of the Reporting summary. Please ensure that, as well as the more general points below, the points highlighted in the attached documents are addressed in full, both on these forms and within the manuscript. Both forms should be uploaded as a "Related Manuscript" file type. The Reporting summary will be published with your paper.

The points mentioned in the reporting summary have been addressed best possibly and both documents are now submitted. But there is one point we want to bring to your attention related to the request to give for each group the exact sample size rather than a range. While we managed to make these changes for EDF2-8, it was impossible to do so for the main Figures 1-5, since this would have increased the word count to up to 600 words. Just to give an example, In Fig 1i and j we have in each panel 4 treatment groups with 3 different conditions (so 12 groups each panel). These panels have group sizes of n=4-6 biological replicates for each of the three conditions. If we now state for each group and condition the exact sample size, then it would say only for Fig 1i the following:

GIPR+PPARa: Vhcl, n=6 biological replicates; GLP-1:GIP, n=4 biological replicates; Lani, n=5 biological replicates, GLP-1:GIP:Lani, n=4 biological replicates, GIPR+PPARg: Vhcl, n=6 biological replicates; GLP-1:GIP, n=4 biological replicates; Lani, n=4 biological replicates, GLP-1:GIP:Lani, n=4 biological replicates;
GIPR+PPARd: Vhcl, n=6 biological replicates; GLP-1:GIP, n=4 biological replicates; Lani, n=5 biological replicates; GLP-1:GIP:Lani, n=4 biological replicates. So this are 51 words only for the sample size in Figure 1i!

If we now do this for the entire Figure 1, then we expand the figure legend from currently 281 words to roughly 500 words. For this reason we can in Figures 1-5 unfortunately give only the range and otherwise refer to the Data Source Files, which show the exact sample size and the individual data corresponding to each sample.

Note that we have currently the following word count for the legends:

- Fig 1 - 281 words
- Fig 2 - 293 words
- Fig 3 - 290 words
- Fig 4 - 299 words
- Fig 5 - 301 words

There is really no possibility other than giving a sample size range for these figures 1-5, unless the journal is fine with an intense extension of the legends.

LENGTH: In print, biological sciences papers do not normally exceed 8 pages on average; the final print length, however, is at the editor's discretion. The typical length of an 8-page article with 5 modest (quarter-page) display items is 4300 words. If a composite figure (with multiple panels) must occupy at least half a page in order for all the elements to be visible, the text length may need to be reduced accordingly to accommodate such figures.

Essential but technical details can be moved into the Methods or Supplementary Information (see below).

In this case, we feel the current length of the paper is appropriate, so no further shortening is necessary.

That's nice. The manuscript has not been shortened accordingly.

TITLE: Titles cannot exceed 75 characters (including spaces); they must not contain punctuation.

The title has been shortened accordingly.

AUTHOR NAMES USING NON-ROMAN CHARACTERS: We can support presentation of author names using non-Roman characters in the HTML version of the published article. Currently supported scripts include Arabic, Chinese, Cyrillic, Devanagari, Greek, Hebrew, Hangul, Japanese and Persian. If this is relevant to your authors, please include the author names in parentheses after the Roman-character spelling (see an example here). You will be asked to verify the rendering is correct at the proof stage.

Author names appear correctly.

SUMMARY PARAGRAPH: Papers start with a fully referenced, bold paragraph, ideally of about 200 words, aimed at readers in other disciplines. Numbers, abbreviations, acronyms or measurements should be avoided unless essential. The summary paragraph consists of 2 to 3 sentences of basic-level introduction to the field; a brief account of the background and rationale of the work; a statement of the main conclusions (introduced by the phrase 'Here we show' or its equivalent); and a conclusion of 2 to 3 sentences putting the main findings into general context so it is clear how the results described in the paper have moved the field forward. A downloadable, annotated example is available here.

The summary paragraph is now referenced.

MAIN TEXT: If further introductory material is necessary, the main text can begin with up to 500 words of introduction expanding on the background to the work (some overlap with the summary is acceptable), before proceeding to a concise, focused account of the findings, and ending with 1 or 2 short paragraphs of discussion. Sections are separated with subheadings (up to 40 characters including spaces) to aid navigation.

REFERENCES: As a guideline, most papers should include no more than 50 main text references; all additional references can be cited in (and listed after) the Methods section, as detailed below.

FIGURE LEGENDS: These should be listed sequentially after the main text references and not in the figure files. Each legend should begin with a brief title for the whole figure and continue with a short description of each panel and the symbols used. Legends should not exceed 300 words each. Each figure legend should contain, for each panel where relevant, the following information:

* the exact sample size (n) for each experimental group/condition, given as a number, not a range;

Please see our answer above. While doable for EDF1-9, this is not possible for Figure 1-5 without significantly expanding the figure legend beyond 300 words (up to 500 words is necessary to include this request).

* a description of the sample collection allowing the reader to understand whether the samples represent technical or biological replicates (including how many animals, litters, cultures, etc);

All samples are biological replicates. The text has been modified accordingly.

* a statement of how many times the experiment shown was replicated;

We now added this information in the section "Statistics and Reproducibility". We also provide the original pictures from the independent in vivo validations as Supplementary Figure 1. The robustness of the in vivo data on the ability of GLP-1:GIP:Lani on body weight and glucose control is demonstrated in multiple independent studies across the manuscript (e.g. Fig 1l,o, Fig 2a,h-k, Fig 3a,d-e, Fig 4a,c,d,h,i,k, EDF2c-e, EDF3a,d,g,h). Notably, the shown metabolic effects of GLP-1:GIP:Lani originate from 5 different labs, located in Germany, Copenhagen, Michigan (US), and New Haven (US), which further underscores the robustness of the shown data. *In vitro* studies in Fig 1f,g and Extended Data Figure 2a,b represent 3-6 independent biological replicates, each obtained in an independent study and calculated based on the average of 2-6 technical replicates. *In vitro* studies in Fig 1i-k represent 4-6 independent biological replicates, each obtained in an independent study and calculated based on the average of 2 technical replicates. Histological data in Fig. 3l are representative examples out of n=8 mice each group. Microscopic images and cFos quantification in Fig. 5a-f are representative examples out of 3-4 mice each group. Electrophysiological recordings in Fig 5p are representative examples out of 6-7 mice each group. All *in vivo* data represent independent biological samples as indicated in the figure legends.

* definitions of statistical methods and measures:

* very common tests (e.g. t-test, simple Chi-square tests, Wilcoxon and Mann-Whitney tests) can be identified by

name only, but more complex techniques should be described in the Methods;

- * whether tests are one-sided or two-sided;
- * whether there are adjustments for multiple comparisons;
- * the statistical test results (e.g., P values);
- * the definition of 'center values' as median or average;
- * the definition of error bars as s.d. or s.e.m.

Descriptions that are too long for the figure legend should be included in the Methods section.

All these information are given in the figure legends. A detailed statistical summary with individual p-values for the different group comparisons is given in supplementary information 1 and the data source files. It was not possible to include the individual p-values in the figure legend without significantly exceeding the 300 word limit for the figure legends.

METHODS: The Methods section, which provides the full, step-by-step instructions that would allow other researchers to replicate the results, is included after the main text figure legends. The Methods section will not appear in print but will appear online in the full-text HTML and PDF versions. The Methods section should be written as concisely as possible but should contain all elements necessary to allow interpretation and reproduction of the results. If there are additional references (in the Methods section, Supplementary Information, etc), their numbering should continue from the last entry in the main text reference list, and they should be listed following the Methods section. Specialized methods that require chemical structures, figures, or tables cannot be accommodated in the Methods section of the main text file. If such information is part of the Methods, the entire Methods section must instead be included within a Supplementary Information text file.

Large methods are mentioned in supplementary information 1.

ETHICS STATEMENT: For research involving human research participants, the Methods section must include an ethics statement. This statement should provide the name of the committee that approved the study; confirm that the research was performed in accordance with all relevant guidelines and regulations; and confirm that informed consent was obtained from all participants. If the study was granted an exemption from requiring ethics approval, details of the committee granting the exemption must be included.

Ethics statement is provided in the manuscript.

Research involving human embryos or gametes, or human stem cells in contexts requiring ethical oversight, also must include an ethics statement in the Methods section. This statement should provide the name of the committee that approved the study and confirm that the research was performed in accordance with all relevant guidelines and regulations. We encourage authors to follow the principles laid out in the 2021 ISSCR Guidelines for Stem Cell Research and Clinical Translation. Where necessary, the ethics statement should also describe the conditions of donation of materials, such as human embryos or gametes, and confirm that informed consent was obtained from all donors of cells or tissues.

MAIN TEXT STATEMENTS: Several statements (which will not appear in print but will appear online in the full-text HTML and PDF) are required after the Methods (and additional references, if present). First, there should be an Acknowledgements section, listing grant/financial support. Next, we require a detailed Author Contribution statement; the specific contributions of each author, particularly in terms of which authors performed which specific experiments, must be listed. This is followed by a Competing Interest statement. Financial and non-financial interests should be noted here, as well as any patents; patent information should include at a minimum patent number, what is covered by the patent, and who submitted the patent application. Finally, an Additional Information statement should include information regarding reprints and permissions and name the author(s) to whom correspondence and requests for materials should be addressed. Formatting details and an example are available here.

All given in the manuscript.

DATA AND CODE AVAILABILITY STATEMENTS: Any manuscript reporting original research must include a Data Availability statement that makes transparent to the reader the conditions of access to the "minimum dataset" that is necessary to interpret, verify and extend the research in the article. This minimum dataset may be provided through deposition in public community/discipline-specific repositories, custom proprietary repositories (for certain types of datasets), or general repositories like Figshare, Zenodo and Dryad. We strongly discourage providing large datasets in Supplementary Information; the preferred approach is to make data available in repositories. More information on Nature Portfolio's reporting standards and guidance on preparing your Data Availability statement can be found here.

For all studies using custom code or mathematical algorithms that are deemed central to the conclusions, a Code

Availability statement must be included, indicating whether and how the code or algorithm can be accessed, including any restrictions to access. The Code Availability statement is listed as a separate section after the Data Availability statement but before any additional references. Code should be deposited in a DOI-minting repository such as Zenodo, Gigantum or Code Ocean and cited in the reference list. Authors are encouraged to manage subsequent code versions and to use a license approved by the open source initiative. Additional details can be found here.

We now included a code availability statement, but this only states that no custom code has been used in the manuscript.

DISPLAY ITEMS: We suggest that you take stock of all data that have been generated throughout the review process and ensure that only the data most central to the conclusions are presented in the main text figures. Any figures included within the main text file during the review process must be removed from the final main text file and uploaded as separate, individual files; they will be integrated into the main paper in print and online. An overview of the key features of this presentation may be found here.

Figures should be comprehensible to readers in other disciplines and assist in understanding of the paper. Main text figures (but **not** Extended Data) must be provided in production-quality versions in an editable format (i.e., .ai, .cmx, .cdr, .doc, .eps, .pdf, .ppt, .ps, .psd, .svg and .xls); we cannot accept figures in .cvs, .gif, .jpg, .png and .tif formats. We highly encourage you to consult our artwork guidelines. They should be as small and simple as is compatible with clarity. All panels of a figure should be logically connected and assembled on a single page in a rectangular shape; any essential alignments (parts horizontal, vertical, spacings, etc) should be indicated. Each panel of a multipart figure should be sized so that the whole figure can be proportionally reduced and reproduced on the printed page at the smallest size at which essential details are visible. Nature's standard figure sizes are either 9 or 18 cm wide; the maximum permitted height is 17 cm. Panels should be arranged to fit these widths while minimizing excess space around the panels. Tables should be prepared using the Table menu in Word. As we must be able to edit the figures so that they conform to our house style, the submission of files that are incorrectly formatted, flattened, or of insufficient resolution may delay final acceptance of your manuscript.

All main figures match the requirements and are provided as eps files.

THIRD PARTY RIGHTS: You must provide proof that you have secured permission to use any third party materials that appear in any part of your manuscript, including Extended Data and Supplementary Information. Please fill out a Third Party Rights Table, and upload this with the final version of your manuscript. Third party materials include any figures, tables, images, videos or text boxes that are reproductions or adaptations of items that have previously been published elsewhere and/or are owned by a third party. This includes pictures taken by professional photographers, maps and images downloaded from the internet. You will need to obtain the right to use each of these items before your paper can be accepted for publication. You will also need to give proper attribution to the copyright holders in your paper. Please ensure you upload any necessary grants of rights alongside the final version of your manuscript. More information is available on our Rights and permissions page. Failure to obtain the appropriate rights and to supply a completed third party rights table will delay the publication of your article. The editorial assistant (cc'd) can help with any questions.

We are including now a third party rights document, but it only states that there are no third party rights affected.

COVER ARTWORK: We welcome submissions of artwork for consideration for our cover. More information can be found in our guide for cover artwork. The file name(s) should include the manuscript reference number and be labelled as a cover suggestion; a short description is also preferred. Illustrations should be selected more for their aesthetic appeal than for their scientific content. We cannot promise that your suggestions will be selected for the cover, as competition is intense.

IMAGE INTEGRITY: We strongly advise that you go carefully through all the data (including Extended Data and Supplementary Information) to ensure there are no accidental image/data duplications, other image manipulations or data errors. Such issues generally require correction after publication. Any image provided for publication, either in print or online (including Extended Data and Supplemental Information), may be subject to a quality control process to check for image integrity and manipulation. A discussion of our standards regarding how images should be prepared and presented can be found here.

We checked the figures again and they all appear correct and solid.

EXTENDED DATA: Extended Data do not appear in print but are included online within the full-text HTML and integrated in the downloadable PDF. Extended Data are an integral part of the paper, and only data that directly contribute to the main message should be included. All Extended Data must be referred to in the main text, and their legends should be listed sequentially at the end of the main text file, not in the Extended Data files. Extended Data should be assembled into a maximum of 10 A4 size, multi-panelled display items. They must be supplied as

individual files in .jpg, .tif or .eps format **only**. They should be of the same quality as the main figures, but there are important differences in their formatting. More specific instructions are provided here. If you need to describe a complex process, we encourage you to add a schematic of the main finding as part of the Extended Data to aid readers unfamiliar with the immediate discipline.

We checked the extended data figures again and they all appear correct and solid.

SUPPLEMENTARY INFORMATION: Supplementary Information (SI) is online-only, peer-reviewed material that is essential background to the study (e.g., large data sets, more complex methods, and calculations), but which is too large or impractical, or of interest only to a few specialists, to justify inclusion in the print version of the paper (see here for further details). While SI should not typically contain data figures (any figures additional to those appearing in the main text should be formatted as Extended Data), we require that the raw, uncropped data for gels be presented as an SI figure (see below). Tables may be included in SI, but only if they are unsuitable for formatting as Extended Data (e.g., tables containing large data sets that cannot fit a single page or raw data tables that are best suited to Excel files). If a manuscript has SI, each discrete SI item (e.g., videos, tables) must be referred to at an appropriate point in the main text file. You must also provide a Word file entitled "SI Guide", containing a cover page with manuscript title and author information; a table of contents (preferably with page numbers); and then any SI text, notes, figures, and titles and legends for any separate SI files; for additional information see here.

A SIguide is now provided. Note that the provided figures in supplementary figure 1 are the original files corresponding to Figure 1p, Figure 3l and Figure 5a,b. Since the main figures only contain representative examples out of several pictures per group, the supplementary figure 1 shows all samples from which the representative pictures for the main figures were taken from. So we just follow the full transparency guideline here. The shown figures here do not add further value for the manuscript, since representative pictures from these data are already provided and discussed with Figure 1p, Figure 3l and Figure 5a,b in the manuscript.

We recommend that you pay careful attention to the formatting of the SI because it is not subedited. After the paper has been accepted, SI files can only be amended for critical changes to the scientific content, not for style.

Supplementary information are correct and double-checked.

CELL LINE IDENTIFICATION: To help curb the inadvertent use of cross-contaminated or misidentified cell lines, we ask that you check your reagents against the list of commonly misidentified cell lines maintained by the International Cell Line Authentication Committee, which is also accessible through the NCBI BioSample database. If you have used a cell line that is on this list, you must provide a scientific justification and state the identity issue in the Methods. The editors reserve the right to demand that the data be removed from the paper if the justification is deemed unsatisfactory. In addition, authors must identify the source of cell lines (with catalog number if obtained from a vendor or cell bank) and report whether the cell lines have been authenticated, including the method used, the results, and the date authentication testing was last performed for that cell line. You should be able to provide the test results upon request. Mycoplasma contamination testing status must also be reported. These requirements will be particularly scrutinized for cancer studies, where the issue of cell line misidentification has been well documented. Resources on cell line authentication are available here.

Proper cell line identification is provided.

SOURCE DATA (GRAPHS): To increase transparency, we strongly encourage you to provide, in spreadsheet form, the data underlying the graphical representations used in figures. For all experiments presenting data from animal models, this is a requirement and is not optional. This is in addition to our well-established data-deposition policy for specific types of experiments and large datasets. Online readers of the manuscript will be able to access the graphical source data directly from the figure legend. Spreadsheets must be submitted in .xls, .xlsx or .csv formats. One file per figure is permitted. If there is a multi-panelled figure, the source data for each panel should be clearly labeled in the file; alternatively the source data for a figure can be included in multiple, clearly labeled sheets within an Excel file. File sizes of up to 30 MB are permitted, but it is expected that the vast majority of graphical source data files will be considerably smaller than this. When submitting these files with your manuscript, you should select the "Source Data" file type and use the title field in the file description tab to indicate the figure(s) to which the source data pertain. Source data should not be provided as Extended Data.

All source data with detailed individual data are provided. Here we also include full statistical reports and individual p-values. A statistical summary with individual p-values for the main treatment effects is also provided in supplementary table 1.

RAW DATA (GELS): You must provide the original source images for all data obtained by electrophoretic separation (e.g., EMSA, northern/Southern/western blots, etc). The raw images must be assembled into a single .pdf or .tif file (multiple gels on a single page is encouraged). The file should be uploaded as Supplementary

Figure 1. The full scanned images must be in uncropped form and contain labeled size/molecular weight markers and loading controls. There should be an accurate indication of how the gels were cropped for the final figure. The figure legends and raw data files should indicate whether controls (such as beta-actin) were run on the same gel as loading controls, or on separate gels as sample processing controls (see here for guidance). While the data can be displayed in a relatively informal style, there must be a correspondence between each source data image and a specific main text or Extended Data figure. The main text or Extended Data figure legends should refer to the uncropped scans explicitly (e.g., "For gel source data, see Supplementary Figure 1."). For examples, see here or here.

The manuscript does not contain western blots, but we provide the uncropped full pictures for the microscopic, histological and immunofluorescent studies in Supplementary Figure 1.

DATA DEPOSITION: The following specific points may be relevant to your paper, so please ensure that you provide the following information:

* Sequences for any RNAi/small RNA constructs must be included.

All primer sequences are mentioned in Supplementary Table 2, which is part of Supplementary Information 1.

* Accession numbers for gene expression data or RNA sequencing data must be listed.

All required accession numbers are now stated.

* Papers reporting protein structures must conform to our standards listed in the Guide to Authors. The Data Availability statement must state that the X-ray crystallographic coordinates and structure factor files (or comparable NMR or cryoEM data) have been deposited in the appropriate, named, public database, along with all relevant accession number(s). You must use the standard Nature templates for structural data; there are separate links to tables for X-ray crystallographic, NMR and cryoEM structures. These tables must be presented as Extended Data; if the number of entries causes the table to exceed a page, it must be divided into two Extended Data items. The contour level of any electron density maps presented, as well as the type of map (i.e., Fo-Fc or 2Fo-Fc), should be explicitly stated in the figure legend.

* For every new chemical compound, a complete description of the synthesis and the physical characterization (i.e., NMR, MS, etc) must be included in the Supplementary Information (see here).

* Papers containing new or revised formal taxonomic nomenclature for animals, whether living or extinct, are accepted conditional on the provision of LSIDs (Life Science Identifiers) by means of registration of such nomenclature with ZooBank, the online registration system for the International Code of Zoological Nomenclature (ICZN). ZooBank LSIDs can be resolved and the associated information viewed through any standard web browser by appending the LSID to the prefix "<http://zoobank.org/>".

* We strongly encourage deposition of 3D morphological data in a suitable repository such as MorphoBank, MorphoSource or similar; the relevant accession numbers should be listed in the Data Availability statement.

* For animal experiments, you must confirm that all experiments were performed in accordance with relevant guidelines and regulations. There should be a statement identifying the institutional and/or licensing committee approving the experiments, including any relevant details. Sex and other characteristics of animals that may influence the results must be described. Details of housing and husbandry must be included if they are likely to influence experimental results. Further details can be found here.

This is stated in the manuscript.

* Human genotype data (e.g., SNP array data) should be deposited into a public database (dbGAP or EGA) with a controlled access policy.

* A full clinical and pathological characterization of patients/human subjects and samples should be provided in tabular format, including the magnitude of response for each patient (partial, complete, stable disease), the site of the biopsy, whether or not that lesion was progressing and mutational status if appropriate.

We will not send your revised paper for further review. If the revised paper is in our format (as detailed above), in accessible style and of appropriate length, we shall begin the acceptance process.

In order to accept your paper, we require the following electronic files:

* A cover letter describing your response to any editorial comments and detailing any format changes during revision, particularly if the overall length is affected.

A cover letter is provided

* A point-by-point response (preferably in Word) to any remaining issues raised by our referees.

A point-by-point letter is provided

* The final version of your text as a Word document. Word Equation Editor/MathType should be used only for formulae that cannot be produced using normal text or symbol font. If this is not possible, the manuscript can be supplied as a single plain vanilla TeX or LaTeX file that includes all references and abbreviations, with no special formatting, as well as a PDF version that is uploaded as a 'related manuscript file'.

Final version of the manuscript is provided

* Production-quality versions of all figures (see above).

All Figures are provided

* The final version of the Extended Data.

All extended data figures are provided

* The final version of any Supplementary Information, presented as one file (ideally a PDF) if feasible, as well as a separate SI Guide.

Supplementary files and SI Guide is provided.

* Source Data, if appropriate.

Source Data are provided.

* For optimal quality videos we encourage H.264 encoding and a standard aspect ratio of 16:9 (4:3 is second best), without compression.

* Completed and signed copies of the following **four (or five) forms**, uploaded as a "Related Manuscript File" file type:

1) Biology editorial checklist;

Provided

2) Manuscript checklist;

Provided

3) Reporting summary;

Provided

4) Third-party rights table;

Provided

5) Code and software submission checklist (if applicable).

No custom code has been used or generated in the manuscript. I guess there is no need to submit a blank code and software submission checklist.

Nature has now transitioned to a unified Rights Collection system which will allow our author services team to quickly and easily collect the rights and permissions required to publish your work. Once your paper is accepted, you will receive an email in approximately 10 business days providing you with a link to complete the grant of rights. If you choose to publish Open Access, our author services team will also be in touch at that time regarding any additional information that may be required to arrange payment for your article. If you have any questions please contact asjournals@springernature.com.

You may need to take specific actions to achieve compliance with funder and institutional open access mandates. If your research is supported by a funder that requires immediate open access (e.g. according to Plan S principles) then you should select the gold OA route, and we will direct you to the compliant route where possible. If you select the subscription publication route our standard licensing terms will need to be accepted, including our self-archiving policies. Those standard licensing terms will supersede any other terms that you or

any third party may assert apply to any version of the manuscript.

All of the files should be uploaded using the following link:

<https://mts-nature.nature.com/cgi-bin/main.plex?el=A2K7CKg4A7KNwc3J5A9fd1gVmmZmsZZXQfNILo0c5YAZ>

Referee #3 (Remarks to the Author):

The authors added a substantial number of additional experiments in the revised manuscript; these addressed some but not all concerns. Most notably for this reviewer is the lack of mechanistic data explaining the large effect size on body weight. In my mind, the new knockout data are not particularly helpful – it is fully expected that knockouts for the incretin receptors should impair weight loss. Further, the manuscript implies that these effects are different than the knockout effects with dual agonism treatment, yet they are quite similar to me (the only difference is the effect size). Given that the body weight effects are likely mediated by CNS control, what is missing is a mechanistic investigation from this perspective. What was provided – fiber photometry data in hypothalamic POMC neurons – is not very satisfying. First, as the authors note, hindbrain changes in their studies were most robust, and this is consistent with the emerging literature that the hindbrain is the primary mediator at least of GLP-1 drugs, and so it is unclear why the authors chose to investigate the hypothalamus which is likely dispensable for the effect of these obesity drugs. Second, fiber photometry data show very short recordings – 400 s long – with activity traces equalizing between GLP-1:GIP and GLP-1:GIP:Lani by the end of this short recording. Therefore, I am very skeptical that this adds any mechanistic insight to the enhanced metabolic effects of GLP-1:GIP:Lani that occur over days/weeks.

We have now expanded the discussion to further discuss the limitations that reside in assessing the mechanisms that underlie the metabolic action of the GLP-1:GIP:Lani conjugate.